# Neural Hamiltonian Diffusions for Modeling Structured Geometric Dynamics

**Sungwoo Park**
Department of Computer Science and Engineering
Korea University
sungwoo_park@korea.ac.kr

## Abstract

We propose *Neural Hamiltonian Diffusion* (NHD), a unified framework for learning stochastic Hamiltonian dynamics on differentiable manifolds. Unlike conventional Hamiltonian Neural Networks (HNNs), which assume noise-free dynamics in flat Euclidean spaces, our approach models stochastic differential equations (SDEs) on curved manifolds endowed with both a Riemannian metric and a Poisson structure. Specifically, we parameterize a neural Hamiltonian and define the dynamics via a Stratonovich SDE whose drift is the Poisson vector field lifted horizontally to the orthonormal frame bundle. This construction ensures coordinate-invariant, gauge-consistent dynamics across (pseudo-)Riemannian manifolds, enabling physically plausible modeling in systems with geometric constraints, periodicity, or relativistic structure. We establish generalization guarantees under curvature-dependent complexity and demonstrate applications across diverse scientific domains, including toroidal molecular dynamics, quantum spin systems, and relativistic $n$-body problems in Schwarzschild spacetime.

## 1 Introduction

Modeling physical dynamics from data is a fundamental challenge in machine learning, with applications ranging from molecular simulations and protein folding [Karplus and McCammon, 2002, Noé et al., 2020] to planetary motion and gravitational systems with curved spacetime [Rein and Liu, 2019, Pretorius, 2005]. A central goal in this context is to learn a dynamical model that not only predicts future states accurately but also reflects the underlying physical laws such as symplectic structure, and geometric invariance [Hairer et al., 2006]. However, many existing learning-based approaches focus on approximating state transitions in Euclidean spaces without explicitly encoding the physics or geometry of the system. In practice, physical systems often evolve on non-Euclidean domains. In molecular dynamics, for example, internal coordinates such as dihedral angles naturally reside on toroidal or pseudo-Riemannian manifolds [Zhou et al., 2020, Townsend et al., 2021]. Likewise, in modeling $N$-body interactions near massive celestial bodies, the trajectories of particles evolve in strongly curved spacetimes, where the underlying geometry plays a crucial role in determining the causal and dynamical structure [Rezzolla and Zanotti, 2013]. Quantum spin systems modeled on compact Lie groups also exhibit inherently curved dynamics due to their non-Euclidean group geometry [Sakurai and Napolitano, 2017]. These systems are inherently geometric, and their governing laws are often described by Hamiltonian mechanics on manifolds equipped with symplectic structures.

Recent advances in Hamiltonian learning including Hamiltonian Neural Networks (HNNs) [Greydanus et al., 2019] and their variants [Zhong et al., 2020, Chen et al., 2021, Cranmer et al., 2020, Wang et al., 2023, Dierkes et al., 2023, Khoo et al., 2023] have shown that incorporating symplectic structure can significantly improve generalization and long-term stability. Complementary approaches such as Symplectic ODE-Nets [Zhong et al., 2020], Symplectic Recurrent Neural Networks [Chen et al., 2021], and Symplectic Transformers [Finzi et al., 2020] embed symplectic constraints directly

39th Conference on Neural Information Processing Systems (NeurIPS 2025).

into the learning architecture. However, these models typically operate in flat Euclidean phase space and do not generalize to curved configuration spaces or non-canonical geometries. Orthogonally, a growing line of research on modeling learnable stochastic dynamics on Riemannian manifolds [De Bortoli et al., 2022, Huang et al., 2022, Park et al., 2022, Mathieu et al., 2023] has enabled geometry-aware stochastic modeling. Yet, these approaches are not physically grounded: they do not enforce Hamiltonian or symplectic structure to preserve physical fidelity. In this context, *our contribution lies in unifying these previously disconnected pillars.* We propose a stochastic Hamiltonian modeling framework that incorporates both the geometric complexity of differentiable manifolds and the structural constraints of Hamiltonian mechanics. We extend Hamiltonian learning to curved, periodic, and causally structured domains with stochasticity. We highlight the following contributions:

- **Neural Hamiltonian Diffusions on Curved Manifolds.** We propose a novel framework that unifies stochastic diffusion processes and Hamiltonian mechanics on general curved spaces. By incorporating gauge consistency into the modeling, we ensure that the learned dynamics remain physically meaningful and independent of local coordinate choices. Our approach respects both the symplectic structure and the intrinsic geometry of the system, enabling faithful simulation of stochastic physical processes beyond flat spaces.

- **Geometry-Consistent Learning via Frame Bundle Lifts.** Instead of learning dynamics directly on the base manifold, we lift the formulation to the frame bundle to handle curvature and coordinate-dependence explicitly. This allows the model to represent vector fields in a unified way across locally varying orthonormal frames, ensuring compatibility between overlapping charts. This geometric design provides consistency across varying local frames and improves the physical reliability of the learned vector fields.

- **Theoretical Guarantees and Empirical Superiority.** We establish theoretical generalization bounds that link curvature, network capacity, and frame symmetry, and prove that gauge consistency intrinsically reduces worst-case deviations. Our method achieves superior performance compared to existing approaches across various structured geometric systems, demonstrating the practical benefits of incorporating geometric and physical consistency into learning.

## 2 Neural Hamiltonian Diffusion

**Hamiltonian Dynamics.** Let $\mathbf{m}_t := (\mathbf{q}_t, \mathbf{p}_t) \in \mathbb{R}^{2d}$ denote the canonical position and momentum coordinates in phase space, and let $\mathrm{H} \in C^\infty(\mathbb{R}^{2d})$ be a smooth Hamiltonian function. We briefly recall the canonical formulation of Hamiltonian dynamics in Euclidean phase space, where the system evolves according to a smooth Hamiltonian function $\mathrm{H}$ via the associated Poisson bracket structure:

$$\frac{d}{dt}\begin{bmatrix}\mathbf{q}_t \\ \mathbf{p}_t\end{bmatrix} = \{\mathbf{m}, \mathrm{H}\} := J\nabla\mathrm{H}(\mathbf{q}_t, \mathbf{p}_t), \quad \text{where} \quad J := \begin{bmatrix}0 & I \\ -I & 0\end{bmatrix}. \tag{1}$$

The operator $\{f, g\} := \nabla f^\top J \nabla g$ defines the canonical Poisson bracket for any smooth functions $f, g \in C^\infty(\mathbb{R}^{2d})$, and $\{\cdot, H\}$ denotes the Hamiltonian vector field applied to observables. This formulation is analytically tractable and serves as the foundation of classical conservative dynamics.

However, such *Euclidean and deterministic* formulations may face fundamental limitations when applied to structured data domain. First, they assume a globally flat phase space, making them ill-suited for modeling systems with curved or topologically structured configuration spaces such as those encountered in relativistic, periodic, or molecular settings. Second, real-world physical dynamics are often inherently *stochastic*, due to latent variables, thermal fluctuations, or observational uncertainty, none of which are reflected in the deterministic formulation. To overcome these limitations, we move beyond the classical regime and explore a generalized class of Hamiltonian systems that operate over *differentiable manifolds* and evolve according to *stochastic dynamics*. Specifically, we adopt the framework of *Hamiltonian diffusion* Bismut [1981], which preserves the structural fidelity of Hamiltonian flows while incorporating both the intrinsic geometry of the underlying space and the probabilistic nature of physical systems. This generalized formulation enables the modeling of structured, curved, and noisy dynamics in a principled and physically consistent manner.

**Hamiltonian Diffusion on Manifolds[1].** Throughout, we work on $2d$-dimensional symplectic manifold $(\mathcal{M}, \omega)$ equipped with a Poisson structure *i.e.,* $\{\cdot, \cdot\}$ with local coordinates $\mathbf{m} := (\mathbf{q}, \mathbf{p}) \in$

---

[1]For a detailed explanation of the background, we refer the reader to the Appendix A.

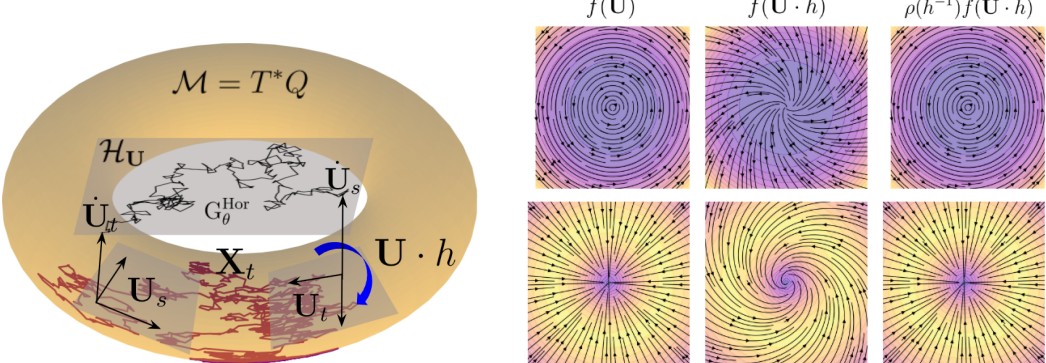

Figure 1: **Horizontally-lifted Hamiltonian Diffusion and Gauge Equivariance on Frame Bundle**. *(Left)* A red stochastic trajectory $\mathbf{X}_t$ evolves on $\mathcal{M} = T^*\mathcal{Q}$. The learned horizontal vector field $\mathbf{G}_\theta^{\mathrm{Hor}}$ transports an orthonormal frame $\mathbf{U}_t$ to $\mathbf{U}_s$ that spans $T\mathcal{M}$, $t \leq s$; its connection-driven rotation is highlighted by the blue arrow ($\mathbf{U} \cdot h$), realizing the symplectic structure in the principal $\mathrm{O}(d)$-bundle. *(Right)* We visualize a specific types of equivariant Hamiltonian vector fields along with their fiber rotations under $\mathrm{O}(d)$. The preserved structure under transformations highlights the gauge equivariance property of our model.

$T^*\mathcal{Q} := \mathcal{M}$, where the configuration space is set to Riemannian manifolds $(\mathcal{Q}, g)$. Now, we give a formal definition of Hamiltonian diffusion on manifolds:

> **Definition 2.1.** *Let $(\Omega, \mathcal{F}_t := \mathcal{F}_t^{\mathbf{B}}, \mathbb{P})$ be the augmented probability space with $2d$-dimensional Brownian motion $\mathbf{B}_t$. Given the standard non-degenerate symplectic $2$-form $\omega := \sum_{i=1}^{d} \mathrm{d}q^i \wedge \mathrm{d}p_i$, we define a $\mathcal{M}$-valued semi-martingale that solves the system of stochastic differential equations:*
>
> $$d\mathbf{X}_t = \{\mathbf{m}, \mathrm{H}_\theta\}(\mathbf{X}_t) \circ d\mathbf{B}_t, \quad \iota_{\{\cdot, \mathrm{H}_\theta\}}\omega = \mathfrak{d}\mathrm{H}_\theta, \quad \mathrm{H}_\theta := \mathrm{H}(\pi_{\mathbf{q}}(\cdot), \pi_{\mathbf{p}}(\cdot); \theta) \in C^\infty(T^*\mathcal{Q} \times \Theta), \quad (2)$$
>
> *where $\pi_{\mathbf{q}} : T^*\mathcal{Q} \to \mathcal{Q}$ and $\pi_{\mathbf{p}} : T^*\mathcal{Q} \to T_{\mathbf{q}}^*\mathcal{Q}$ are canonical projections onto configuration manifold and the fiber, and $\iota$ and $\mathfrak{d}$ denote the interior product and the exterior derivative on $\mathcal{M}$, respectively.*

A *neural Hamiltonian diffusion* (NHD) refers to a stochastic process $\mathbf{X}_t$ evolving on a manifold under the Hamiltonian flow, where the Hamiltonian $\mathrm{H}_\theta$ is modeled by neural networks with parameters $\theta \in \Theta$ so as to reflect the induced physical structure of the system. As can be seen, the Poisson bracket formulation naturally leads to geometry-aware dynamics, ensuring that the induced flow respects the curvature and structure of the underlying manifold. To be more specific, the vector field on the manifold takes the form

$$\{\mathbf{m}, \mathrm{H}_\theta\} := \mathbf{J}_g \nabla_{\mathbf{m}} \mathrm{H}_\theta(q, p) = \begin{bmatrix} 0 & \mathbf{G}^{-1}(q) \\ -\mathbf{G}^{-1}(q) & 0 \end{bmatrix} \nabla_{\mathbf{m}} \mathrm{H}_\theta(q, p), \quad (3)$$

where $\mathbf{m} = (q, p) \in T^*\mathcal{Q}$, and $\mathbf{G}^{-1}(q)$ is the inverse of the Riemannian metric on the configuration manifold $\mathcal{Q}$. The resulting vector field retains the skew-symmetric structure of Hamiltonian flows while encoding local geometric information through the metric $\mathbf{G} := [g_{ij}]$, and can be viewed as a geometry-aware generalization of Euclidean Hamiltonian vector fields in Eq (1).

In deterministic Hamiltonian systems in Eq (1), *energy conservation* is encoded by the identity $\dot{H} = 0$, which holds along every trajectory, ensuring exact invariance of the Hamiltonian over time. This reflects the fact that the energy function remains constant along deterministic flows. In contrast, our stochastic Hamiltonian framework characterizes energy conservation through the *stationarity* of the equilibrium distribution, given by $\mathcal{L}_\theta \pi = 0$, where $\pi \propto e^{-H(\mathbf{X}_\infty)}$. Rather than preserving energy along individual sample paths, the Hamiltonian in this case governs the long-term statistical behavior of the system via the generator $\mathcal{L}_\theta$. Table 3 summarizes the distinction between these two paradigms.

**Horizontal Lift of Hamiltonian Diffusion.** One major difficulty in the simulation of Eq. (2) lies in the absence of canonical coordinates, as well as the lack of a principled method to define stochasticity on manifolds. Recently, [De Bortoli et al., 2022] suggested geodesic random walks (GRWs) which harness the property of *extrinsic geometry* by using Riemannian exponential maps. Yet, there are open questions to respect Hamiltonian and symplectic structures by using GRWs. In contrast to extrinsic approaches, we formulate the *intrinsic geometry* by lifting the process to the frame bundle [Hsu,

2002], where geometry-consistent noise can be defined, naturally allowing for a principled realization of stochastic Hamiltonian dynamics on manifolds Lázaro-Camí and Ortega [2008].

Formally, we introduce a horizontal lift (*i.e.*, $\mathbf{U}_t$) of the diffusion process (*i.e.*, $\mathbf{X}_t$) to the frame bundle $\mathcal{O}(\mathcal{M})$, where the stochastic dynamics admit local frame coordinates adapted to the manifold:

**Proposition 2.2** (Horizontal Hamiltonian Diffusion). *Let $\mathbf{U}_t \in \mathcal{O}(\mathcal{M})$ be the horizontal lift of the diffusion process $\mathbf{X}_t = \pi(\mathbf{U}_t)$, where $\pi : \mathcal{O}(\mathcal{M}) \to \mathcal{M}$ is the canonical projection and $\mathbf{m}$ denotes a local coordinate function on $\mathcal{M}$. The lifted process $\mathbf{U}_t$ evolves according to the Stratonovich SDE:*

$$d\mathbf{U}_t := G_\theta^{\mathrm{Hor}}(\mathbf{U}_t) \circ d\mathbf{B}_t, \quad \pi(\mathbf{U}_t) = \mathbf{X}_t, \quad \mathbf{U}_t \in \mathcal{O}(\mathcal{M}), \tag{4}$$

$$\underbrace{G_\theta^{\mathrm{Hor}}(\mathbf{U}_t)}_{\in \mathrm{H}_\mathbf{U} \,:\, Horizontal} := \underbrace{(\{\mathbf{m}, \mathrm{H}_\theta\}(\mathbf{X}_t)) \nabla_x}_{:= \hat{G}_\theta} - \underbrace{\left[ \left( \mathbf{I}_d \otimes (\{\mathbf{m}, \mathrm{H}_\theta\}(\mathbf{X}_t))^\top \right) \cdot [\Gamma^\mathcal{M}]^\flat \mathrm{vec}(E) \right] \nabla_e}_{:= \omega^\# \hat{G}_\theta \,\in\, V_\mathbf{U} \,:\, Vertical\ Fiber}, \tag{5}$$

*where $[\Gamma^\mathcal{M}]^\flat \in \mathrm{M}(2d \times d^2)$ is the index-lowered connection tensor (i.e., Christoffel symbol), and $\mathrm{vec}(E) \in \mathrm{M}(d^2, 1)$ is the vectorized local orthonormal frame. $\nabla_x$ and $\nabla_e$ denote the vectorized gradients with respect to the configuration point and the frame coordinates, respectively.*

Here, the horizontal process $\mathbf{U}_t \in \mathcal{O}(M)$ augments the manifold trajectory with a local orthonormal frame, enabling noise to be defined in the tangent space and lifted via the horizontal distribution. In the formulation to ensure horizontal transport, we remove the vertical fiber part using the connection one-form $\omega^2$. The resulting vector field $G_\theta^{\mathrm{Hor}}(\mathbf{U}_t) \in \mathrm{H}_{\mathbf{U}_t}$ is obtained by projecting $\hat{G}_\theta$ onto the horizontal distribution as $G_\theta^{\mathrm{Hor}} = (\mathrm{Id} - \omega^\sharp) \hat{G}_\theta$, which ensures that the Stratonovich increment lies in $\mathcal{H}_\mathbf{U} = \ker \omega$. Figure 1 schematically illustrates this construction. Note that we lift the base metric $g$ to a Sasaki-type metric $g^\mathcal{M}$ to induce geometric structure on the total space $\mathcal{M} = T^*\mathcal{Q}$. The resulting connection tensor $\Gamma^\mathcal{M}$ serves as a geometric object defined on the total manifold of trajectory $\mathbf{X}_t$:

**Definition 2.3** (Lifted Metric on the Total Space). *Let $\mathcal{Q}$ be a configuration manifold equipped with Riemannian metric $g$ with its total space $\mathcal{M} := T^*\mathcal{Q}$. We define Sasaki-type metric on $\mathcal{M}$ as $g^\mathcal{M} := \pi_\mathbf{q}^* g \oplus g^{-1}$, where $\pi_\mathbf{q}^* g$ is a pull-back metric with respect to $\pi_\mathbf{q}$. Then, the connection tensor associated with $g^\mathcal{M}$ take the following block tensor form:*

$$[\Gamma^\mathcal{M}]_{bc}^a = \begin{bmatrix} \Gamma_{jk}^i & \frac{1}{2} \left( \partial_{q_j} g^{ik} + \partial_{q_k} g^{ij} \right) \\ -\frac{1}{2} \partial_{q_i} g^{jk} & \mathbf{0}_{d \times d} \end{bmatrix}, \quad \forall a, b, c \in \{1, \dots, 2d\}. \tag{6}$$

Having the complete definition of Hamiltonian diffusion on $\mathcal{M}$ by using horizontal lifts in Proposition 2.2, we now shift our focus from defining the model to learning it from physical data.

**Learning Geometric Dynamics.** Main feature of modeling physical systems is that the system is required to reconstruct both positional and complementary information such as velocity and momentum. This additional physical information is represented as trajectories evolving in the second-order tangent bundle of the phase space, formally modeled as $\gamma_t = (\mathbf{q}_t, \mathbf{p}_t, \dot{\mathbf{q}}_t, \dot{\mathbf{p}}_t) \in T(T^*\mathcal{Q})$. Given observational trajectory sampled from a data path distribution $\tilde{\gamma}_t \sim \mathbb{P}_{t,\mathrm{data}}$ on physical data space, we seek to recover the underlying Hamiltonian $\mathrm{H}_\theta$ that approximately generates the observed dynamics. The objective is to align the empirical drift of the process with the vector field induced by the learned Hamiltonian. To this end, we define the training loss over path distribution:

**Definition 2.4** (Hamiltonian Learning). *Let $\gamma_t = (q_t, p_t, \dot{q}_t, \dot{p}_t) \in T(T^*\mathcal{Q})$ be a sample trajectory on the physical data space, and let $\tilde{\gamma}_t \sim \mathbb{P}_{t,\mathrm{data}}$ denote the corresponding data distribution. Given a parameterized Hamiltonian $\mathrm{H}_\theta : T^*\mathcal{Q} \to \mathbb{R}$, the geometric learning objective is defined as follows:*

$$\mathcal{L}(\theta) := \mathbb{E}_{t \sim \mathbf{U}[0,T], \tilde{\gamma}_t} \left[ \left\| [\tilde{\mathbf{q}}_t, \tilde{\mathbf{p}}_t]^T - \{\mathbf{m}, \mathrm{H}_\theta\} \circ \pi(\mathbf{U}_t) \right\|_{g^\mathcal{M}}^2 + d_\mathcal{Q}(\tilde{\mathbf{q}}_t, \pi_\mathbf{q} \circ \pi(\mathbf{U}_t)|_\theta)^2 \Big| \mathbf{U}_0 \right], \tag{7}$$

*where $\|\cdot\|_{g^\mathcal{M}}$ is a Riemannian norm, and $d_Q$ denotes a distance function on configuration space.*

---

[2]Abusing notation, we denote both the connection one-form and the symplectic form by $\omega$ for simplicity (with $\omega^\#$ is its pull-back), as the meaning will be unambiguous from context.

Eq. (7) defines a hybrid loss that combines a vector field alignment term and a trajectory reconstruction term. The first term ensures that the learned Hamiltonian vector field aligns with the empirical time derivatives of the state to reflect the underlying physical dynamics. The second term penalizes deviations between the predicted and observed positions on the configuration manifold. The entire objective is conditioned on a fixed initial state $\mathbf{U}_0 = \pi^{-1}([\mathbf{q}_0, \mathbf{p}_0])$, reflecting the *initial value nature* of Hamiltonian systems where the full trajectory is determined by the initial condition. Together, these objectives encourage the model to learn a Hamiltonian function that faithfully captures both the evolution of the system and its observed behavior.

**Designing Neural Hamiltonian Functions.** To model physically consistent dynamics in curved or structured spaces, we aim to construct Hamiltonian functions that reflect the underlying geometry of the configuration space. As motivated from conventional force-field modeling Salomon-Ferrer et al. [2013], we begin by formulating $N$-body interaction systems on (pseudo-)Riemannian manifolds parameterized by neural networks.

$$
\mathrm{H}_\theta \begin{pmatrix} \mathbf{q}^N \\ \mathbf{p}^N \end{pmatrix} = \underbrace{\frac{1}{2} \sum_{i=1}^{N} [\mathbf{p}^i]^\top \mathbf{G}^{-1} \mathbf{p}^i}_{\text{(i) Kinetic Energy}} + \underbrace{\sum_k \mathcal{W}_\theta \begin{pmatrix} \mathbf{q}^k \\ \mathbf{p}^k \end{pmatrix}}_{\text{(ii) Single-particle Potential}} + \underbrace{\sum_{i<j} \mathcal{V}_\theta \begin{pmatrix} \mathbf{q}^i \\ \mathbf{q}^j \end{pmatrix}}_{\text{(iii) Pairwise Interaction}} . \tag{8}
$$

Eq. (8) defines a parameterized Hamiltonian function for a system of interacting particles, where the joint state $(\mathbf{q}^N := \{q_1, \cdots, q_N\}, \mathbf{p}^N := \{p_1, \cdots, p_N\}) \in \mathcal{M}^N$ evolves on the cotangent space of the joint configuration manifold $(\mathcal{M}^N, (g^{\mathcal{M}})^N)$. The proposed Hamiltonian consists of three distinct components: (i) the first kinetic energy term represents the kinetic energy of each particle and incorporates geometry-awareness by using the local inverse Riemannian metric, (ii) the second potential term captures single-particle effects through neural potentials that depend on individual states, including temporal, environmental, or local structural influences, and (iii) the third term accounts for pairwise interactions that model spatial dependencies and mutual influence across particles, which are essential for capturing correlated behavior on the manifold. Having established the parameterized Hamiltonian function $\mathrm{H}_\theta$, we now derive the associated Poisson bracket $\{\cdot, \mathrm{H}_\theta\}$ expressed explicitly in local coordinates on $\mathcal{M}$ as follows:

$$
\{\mathbf{m}, \mathrm{H}_\theta\} := \mathbf{J}_g \nabla_\mathbf{m} \mathrm{H}_\theta = \mathbf{J}_g \left[ \mathbf{G}^{-1} \mathbf{p}^N + \frac{\partial \mathcal{W}_\theta}{\partial \mathbf{p}}, -\frac{\partial \mathcal{W}_\theta}{\partial \mathbf{q}} + \sum_{i \le j} \frac{\partial \mathcal{V}_\theta(\mathbf{q}^i, \mathbf{q}^j)}{\partial \mathbf{q}} \right]^T \in \mathfrak{X}(T^* Q^N).
$$

In what follows, we impose a structural constraint on the neural Hamiltonian $\mathrm{H}_\theta$ that respects the gauge symmetry of the underlying frame bundle, leading to the formulation of gauge equivariance on $\mathcal{O}(\mathcal{M})$ to ensure the efficient learning of the parameterized Poisson bracket $\{\mathbf{m}, \mathrm{H}_\theta\}$.

**Gauge Equivariance on Frame Bundle.** While the choice of frame can be arbitrary, the proposed horizontal lift canonically projects dynamics from the base manifold to its frame bundle in a manner that is independent of specific frame parametrization. In this context, gauge equivariance is an essential property for ensuring that learned dynamics remain consistent across arbitrary frame choices and preserve geometric coherence during transport Cohen et al. [2019]. This principle can also formally be realized through the geometry of the orthonormal frame bundle in our framework.

Let $\mathbf{U} \in \mathcal{O}(\mathcal{M})$ and $h \in \mathrm{O}(d)$ be orthonormal frame and the rotation defined by $R_h(\mathbf{U}) := \mathbf{U} \cdot h$, then the orthonormal frame bundle $\mathcal{O}(\mathcal{M})$ admits a natural right action of the structure group $\mathrm{O}(d) := \{h \in \mathrm{GL}(d, \mathbb{R}) \mid h^\top h = I_d\}$. This action describes local gauge transformations within each fiber, and naturally lifts geometric quantities from base manifold $\mathcal{M}$ into an equivariant bundle.

**Definition 2.5** (Gauge Equivariance on the Frame Bundle). *Let $f : \mathcal{O}(\mathcal{M}) \to V$ be a map into a representation space $V \cong \mathbb{R}_q^d \oplus \mathbb{R}_p^d \cong \mathbb{R}^{2d}$, corresponding to position and momentum components in the cotangent bundle. We say that $f$ is* gauge equivariant *if, for every $h \in \mathrm{O}(d)$,*

$$
f(\mathbf{U} \cdot h) = \rho(h^{-1}) f(\mathbf{U}), \qquad \rho(h) = \begin{bmatrix} I_d & 0 \\ 0 & h \end{bmatrix} \in GL(2d), \tag{9}
$$

*where $\mathbf{U} \in \mathcal{O}(\mathcal{M})$ is an orthonormal frame, and $\mathbf{U} \cdot h$ denotes the right action of $h$ on $\mathbf{U}$.*

Figure 1 provides an illustrative example of gauge equivariance defined over the orthonormal frame bundle. The equivariance property reflects that under frame transformation, local momentum vectors

$p \in \mathbb{R}^d$ transform covariantly as $p \mapsto hp$, while position vectors $q \in \mathbb{R}^d$ remain invariant as base coordinates. In the context of Hamiltonian learning, the goal is then to construct the neural Hamiltonian $H_\theta$ such that the resulting geometry-induced drift term $G_\theta^{\mathrm{Hor}}(\mathbf{U})$ is gauge equivariant under the frame transformation $\mathbf{U} \mapsto \mathbf{U} \cdot h$, which holds if the following condition is satisfied:

$$\{m, H_\theta\}(\pi(\mathbf{U} \cdot h)) = \rho(h^{-1})\{m, H_\theta\}(\pi(\mathbf{U})), \qquad \forall h \in \mathrm{O}(d). \tag{10}$$

To realize equivariant learning within frame bundle coordinates, we propose the Frame Equivariant Transformer U-Net[3]. This architecture integrates canonicalization by transforming coordinates into the local orthonormal frame via $\mathbf{U}^\top p$, performing invariant computations, and reconstructing outputs via $\mathbf{U}\hat{p}$, thereby ensuring gauge-consistent predictions. In Appendix, we provide the algorithm and pseudo-code for sampling frame-equivariant and neural network architectures.

## 3 Theoretical Analysis

In this section, we present two theoretical results: a generalization bound linking curvature and model capacity, and a deviation bound showing how gauge equivariance improves stability across frames.

**Uniform-in-time Generalization of Hamiltonian Learning.** With the objective function posed earlier, a natural question arises: *If the model achieves near-perfect trajectory matching, why is Hamiltonian learning still necessary?* This question is fundamental, as trajectory matching alone does not guarantee physically meaningful generalization. To analyze this rigorously, we consider the neural network $\theta^\star$, which exactly reproduces the physical trajectories.

$$(\mathbf{C1}) \qquad \theta^\star = \arg\min_{\theta \in \Theta} d_\mathcal{Q}(\tilde{\mathbf{q}}_t, \pi(\mathbf{U}_t|_\theta)), \qquad \Theta := B_{W^{s,2}}(\theta^\star, R). \tag{11}$$

Here, the radius $R$ of the ball $\Theta$ reflects the capacity of neural networks. Unfortunately, although the neural network $\theta^*$ perfectly fits the physical trajectory in the training phase, it fails to capture the holistic physical information such as velocity and momentum. Proposition 3.1 demonstrates that our proposed geometric Hamiltonian learning significantly improves generalization.

**Proposition 3.1** (Informal). *Let $\mathbb{P}_t(\theta) := \mathrm{Law}(\gamma_t(\theta))$ be an associated probability measure of model trajectory, and assume the condition $(\mathbf{C1})$ in Eq. (11) holds, Under the mild regularity conditions of Hamiltonian function, the learned model distribution fails to remain close to the data distribution uniformly over time with high probability:*

$$\mathbb{P}\left[\sup_{t \in [0,T]} \sup_{\theta \in \Theta} \mathcal{W}\left(\mathbb{P}_t(\theta), \mathbb{P}_{t,\mathrm{data}}\right) \leq \delta\right] \lesssim \exp\left(-\Omega \cdot \delta^{1/2}\|\mathbf{\Gamma}\|_\infty^{3/2} R^{1/2}(\log R)^{1/4}\right),$$

*where $\mathcal{W} := \mathcal{W}_{T(T^*Q)}^{2,2}$ denotes the squared Wasserstein distance on the physical data space $T(T^*Q)$, and $\Omega$ is a constant depending on geometric and model-specific quantities.*

Generalization in geometric Hamiltonian learning hinges on two key factors: the curvature of the configuration manifold (*i.e.*, $\mathbf{\Gamma}$) and the network capacity (*i.e.*, $R$). High curvature ($\|\Gamma\|_\infty$) intensifies stochastic distortion, while large $R$ increases variance. This induces a trade-off—expressive models capture complex geometry but risk overfitting under curvature. Trajectory matching alone is insufficient, often neglecting velocity and momentum structure. Our method resolves this by enforcing physically consistent dynamics beyond position-level fitting.

**Gauge Equivariance Ensures Smaller Deviations.** In the second theoretical finding, Proposition 3.2 shows that enforcing gauge equivariance not only yields uniformly smaller worst-case Wasserstein deviations between the learned trajectories and the reference geodesic across all admissible frames, but also tightens the resulting generalization bounds by eliminating spurious frame-dependent variance.

**Proposition 3.2** (Informal). *Let $\gamma : [0, T] \to T^*Q$ be a reference physical data represented as a geodesic. For any Hamiltonian $H_\theta$ define the frame–rotated trajectory by $\mathbf{X}_t(h) := (q_t, \pi_\mathbf{p} \circ \pi(\mathbf{U}_t \cdot h))$ for $h \in \mathrm{O}(d)$. Then, for arbitrary $h' \in \mathrm{O}(d)$, there exists constants $\kappa, C > 0$ such that the following inequality holds:*

$$\mathcal{W}\left(\mathbb{P}_t^{\mathrm{eq}}(h'), \delta_{\gamma(t)}\right) = \sup_{h \in \mathrm{O}(d)} \mathcal{W}\left(\mathbb{P}_t^{\mathrm{eq}}(h), \delta_{\gamma(t)}\right) \leq \sup_{h \in \mathrm{O}(d)} \mathcal{W}\left(\mathbb{P}_t(h), \delta_{\gamma(t)}\right) \leq Ce^{\kappa t}, \tag{12}$$

*where $\mathbf{X}_t^{\mathrm{eq}}(h) \sim \mathbb{P}_t^{eq}(h)$ is generated by a gauge–equivariant Hamiltonian function $H_\theta$.*

---

[3]Appendix D contains in-detailed information of model architecture.

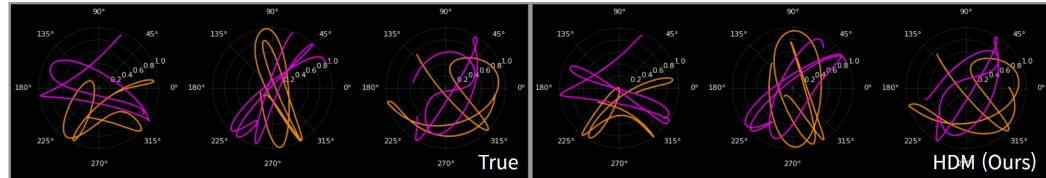

Figure 2: **Visualization of Three-body Quantum Spin Dynamics via Hopf Projection.** Each subplot shows the spin trajectory of a single body on the 3-sphere $\mathbb{S}^3$, projected to two orthogonal complex planes: $z_1 = x + iy$ (orange) and $z_2 = z + iw$ (magenta). *(Left)* Ground-truth trajectories reveal nonlinear yet phase-coherent dynamics. *(Right)* Our model (HDM) accurately reproduces the spin geometry across bodies.

## 4 Related Works

**Hamiltonian Neural Networks.** Hamiltonian Neural Networks (HNNs) [Greydanus et al., 2019] introduced the idea of learning a scalar energy function $H(q, p)$ whose gradients define conservative dynamics. Several extensions have since been proposed to improve generality, structure preservation, or application-specific modeling. [Cranmer et al., 2020] proposed learning Lagrangian dynamics as an alternative to the Hamiltonian formulation. [Chen et al., 2021] introduced symplectic recurrence for better long-term stability. [Wang et al., 2023] incorporated symplectic constraints into the learning process. [Simiao et al., 2023] adapted HNNs to rigid-body dynamics with energy-aware formulations. [Dierkes et al., 2023] focused on automatic symmetry detection and exploitation. [Khoo et al., 2023] proposed modeling separable Hamiltonians to reflect physical modularity. Unlike prior work constrained to Euclidean domains, our model learns Hamiltonian dynamics directly on manifolds, enabling faithful modeling of geometry-aware physical systems.

**Neural Diffusion on Manifolds.** Recent work has extended neural stochastic modeling to non-Euclidean spaces, particularly Riemannian manifolds, by incorporating geometric structure into diffusion or score-based generative models. [De Bortoli et al., 2022] proposed Riemannian score-based generative modeling on smooth manifolds, generalizing Langevin dynamics and score matching to curved spaces. [Huang et al., 2022] developed Riemannian diffusion models by extending continuous-time stochastic differential equations (SDEs) to arbitrary manifolds. [Park et al., 2022] introduced Riemannian Neural SDEs, enabling stochastic representation learning directly on manifolds using intrinsic geometry. [Lou et al., 2023] addressed the scalability of Riemannian diffusion models for high-dimensional and complex manifold settings. [Fishman and Cunningham, 2023] tackled constrained diffusion in non-Euclidean domains by incorporating boundary-aware mechanisms. These works lay the foundation for stochastic modeling on manifolds. Building on this line of research, our approach extends geometric diffusion models with a Hamiltonian perspective, enabling structured modeling of physical dynamics on curved spaces.

## 5 Experiments

**Problem Formulation.** In this section, we validate our proposed framework across three distinct physical scenarios that reflect a diverse range of geometric structures: (i) an interacting spin system evolving on the compact Lie group manifold $\mathrm{SU}(2) \cong \mathbb{S}^3$, (ii) relativistic $N$-body dynamics formulated on Lorentzian spacetimes such as the Schwarzschild manifolds and (iii) molecular dynamics of protein backbones represented on high-dimensional toroidal configuration spaces $\mathbb{T}^N$. Each setting highlights a unique combination of curvature, topology, and physical constraints, allowing us to assess the generality and fidelity of neural Hamiltonian diffusion on non-Euclidean domains. We compare our method with recent state-of-the art methodologies in geometric sequential modeling including GeoTDM Han et al. [2024], EqMotion Xu et al. [2023], EGNN Satorras et al. [2021], SE-3 transformer Fuchs et al. [2020]. Hamiltonian learning based such as HNN Greydanus et al. [2019], SympHNN David and Méhats [2023], Noether van der Ouderaa et al. [2024] are also considered.

In all scenarios, we formulate physical dynamics prediction as a sequence modeling problem on non-Euclidean manifolds. Let $\{\gamma_t\}_{t=1}^T$ denote a trajectory of geometric states $\gamma_t \in T(T^*\mathcal{Q})$ sampled from a Hamiltonian system. During training, each model is conditioned on a single initial state $\gamma_1$ and trained to autoregressively predict the subsequent states $\{\gamma_t\}_{t=2}^{T_{\mathrm{obs}}}$. At test time, the predicted state is re-fed into the model to generate the next one, allowing the model to learn long-range extrapolation dynamics at each step. This setup reflects realistic forecasting settings where long-term evolution must be inferred from geometric observations. A comprehensive summary of the experimental setups is included in the Appendix.

| Model | AD-3 | 2AA | 4AA | Spin | Schwarzschild |
|---|---|---|---|---|---|
| HNN | 0.413 / 0.779 | 0.612 / 0.859 | $\geq 1.0$ | 0.141 / 0.275 | 0.106 / 0.218 |
| Noether | 0.554 / 0.614 | 0.580 / 0.723 | $\geq 1.0$ | 0.077 / 0.162 | 0.053 / 0.116 |
| SympHNN | 0.596 / 0.717 | 0.519 / 0.736 | $\geq 1.0$ | 0.083 / 0.124 | 0.063 / 0.175 |
| SE(3)-Tr. | 0.312 / 0.513 | 0.445 / 0.596 | 0.649 / 0.830 | 0.384 / 0.665 | 0.338 / 0.437 |
| EGNN | 0.251 / 0.501 | 0.367 / 0.405 | 0.417 / 0.474 | 0.182 / 0.242 | 0.155 / 0.246 |
| EqMotion | 0.081 / 0.117 | 0.062 / 0.152 | 0.131 / 0.174 | 0.090 / 0.102 | 0.081 / 0.149 |
| GeoTDM | **0.045** / **0.102** | **0.079** / **0.145** | **0.093** / **0.179** | **0.037** / **0.085** | **0.026** / **0.046** |
| NHD (Ours) | **0.023** / **0.084** | **0.055** / **0.103** | *0.117* / *0.186* | **0.019** / **0.097** | **0.012** / **0.035** |

Table 1: **Comparison of toroidal protein trajectory prediction and curved-space $N$-body dynamics.** The first three columns (AD-3, 2AA, 4AA) report ADE/FDE on protein torsion angle trajectories. The last two columns (Spin and Schwarzschild) report ADE from $N$-body simulations of spin-based and Schwarzschild-metric particle systems with $N = 3$ and $N = 5$ particles. The first and second best is highlighted with **bold** and *blue*.

**Molecular Dynamics of Protein Backbones.** In modeling Hamiltonian formulations of protein molecular dynamics, we are motivated by classical force fields used in molecular modeling Cornell et al. [1995], Maier et al. [2015], Tian et al. [2019], which incorporate structured physical interactions such as bond stretching, angle bending, torsional rotations, and non-bonded forces. We reinterpret these force fields as learnable neural potentials while preserving underlying geometric and physical consistency. The configuration space is set to high-dimensional torus $\mathcal{Q} := \mathbb{T}^{N_{\text{angle}}}$, where $N_{\text{angle}}$ denotes the number of torsional degrees of freedom (*e.g.*, , backbone dihedral angles $\phi, \psi, \omega$ and side-chain angles $\chi_i$). To evaluate the proposed framework, we perform experiments on three representative peptide systems of increasing complexity: AD-3 Alanine dipeptide, which exhibits

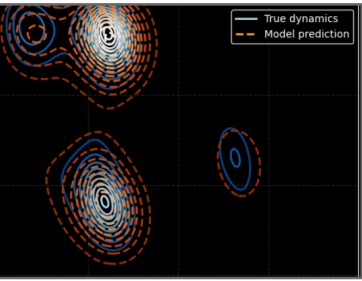

Figure 3: **Spatiotemporal Ramachandran Map**. Torsional state evolution over time compared between true and predicted trajectories.

simple dynamics on $\mathbb{T}^2$, 2AA dipeptides, where $\mathbb{T}^4$ arises from backbone and occasional side-chain torsions, and 4AA tetrapeptides, which form structured dynamics on $\mathbb{T}^{12}$ due to multiple interacting torsional modes. To extract geometric Hamiltonian states, we post-process time-aligned atomic trajectories from the Timewarp Klein et al. [2023] to compute angles $(\phi, \psi, \omega, \chi_i)$ as generalized coordinates, and approximate their corresponding momenta by estimating the reduced moment of inertia associated with each torsional mode. Time derivatives are computed via finite differences across consecutive frames. We evaluate trajectory quality using standard geodesic metrics on the torus manifold, including average displacement error (ADE) and final displacement error (FDE), where distances are measured along $\mathbb{T}^{N_{\text{angle}}} = \mathbb{S}^1 \times \cdots \times \mathbb{S}^1$. As summarized in Table 1, our method outperforms existing benchmarks by a significant margin across all evaluated metrics.

**Quantum Spin System.** In quantum physics, a spin system refers to a collection of particles, each possessing an intrinsic angular momentum (spin) that interacts with neighboring spins according to specified coupling rules. Mathematically, spin states are modeled as unit vectors on a sphere or, as elements of compact Lie groups. We model the dynamics of mutually interacting quantum spins on the unit 3-sphere $\mathbb{S}^3 \subset \mathbb{R}^4$, where each spin is represented as a unit quaternion that evolves under rigid-body dynamics. The system is equipped with anisotropic inertia and pairwise coupling Hamiltonians, giving rise to nonlinear, geometry-constrained motion. The total Hamiltonian of the system takes the following form: $H(\mathbf{q}^N, \mathbf{p}^N) = \frac{1}{2}\sum_{i=1}^{N}(\mathbf{p}_i)^\top \mathbf{I}^{-1}\mathbf{p}_i - \sum_{i<j} \lambda_{ij}\left(\langle \mathbf{q}_i, \mathbf{q}_j \rangle\right)^2$ where $\omega_i \in \mathbb{R}^3$ is the body angular velocity of the $i$-th spin, $I \in \mathbb{R}^{3\times3}$ is the moment of inertia tensor, and $\lambda_{ij}$ is the coupling strength promoting alignment between spins $q_i$ and $q_j$. The inner product $\langle q_i, q_j \rangle = x_i x_j + y_i y_j + z_i z_j + w_i w_j$ measures the similarity of unit quaternions on $\mathbb{S}^3$. The time evolution is governed by the Hamiltonian equations $\dot{q}_i = \frac{1}{2}\Omega(\omega_i)q_i$ and $\dot{\omega}_i = I^{-1}\tau_i$, where $\Omega(\omega)$ encodes angular velocity and $\tau_i$ is the coupling torque promoting spin alignment. The induced dynamics are thus constrained to a Riemannian manifold, specifically the 3-sphere endowed with its canonical metric. We visualize the resulting trajectories using Hopf projection $\pi : \mathbb{S}^3 \to \mathbb{S}^2$ in Figure 2, where each spin is mapped to complex plane components $(z_1, z_2) \in \mathbb{C}^2$ with $z_1 = x + iy$, $z_2 = z + iw$, $|z_1|^2 + |z_2|^2 = 1$. Both the qualitative trajectories in Figure 2 and the quantitative metrics in Table 1 demonstrate that our dynamics accurately capture the underlying spin system evolution.

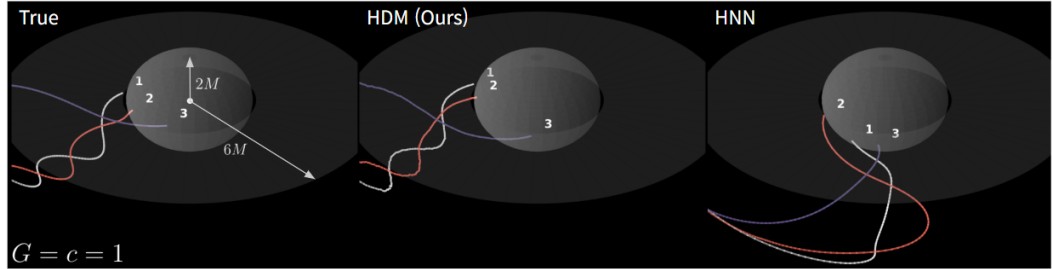

Figure 5: **Three-body Trajectories in the Spacetime of Schwarzschild Black Holes.** *Left*: The ground-truth simulation obtained by numerically integrating the exact relativistic dynamics, shows three mutually interacting bodies (labels 1–3) spiraling toward the event horizon (*i.e.*, Schwarzschild Radius = $2M$). *Center*: The proposed method accurately captures the relativistic deflection and inward inspiral of all three trajectories, remaining faithful to the ground-truth. *Right*: Existing Euclidean HNN trained without explicit geometric conditioning yields inconsistent trajectories that indicate an incorrect physical regime.

**Relativistic Particle Dynamics.** In the last experiment, we consider the dynamics $N$ interacting bodies in the curved spacetime surrounding compact astrophysical objects such as *Schwarzschild* black hole. In formulation, the background force field is derived from general relativity, encapsulating the relativistic geometry of spacetime. Meanwhile, the interaction forces between bodies follow classical modeling assumptions, *e.g.*, pairwise

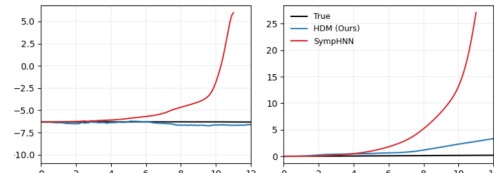

Figure 4: Comparison of total Hamiltonian $H(t)$ and cumulative relative drift $\mathbb{E}_t|\Delta H_t|/H_0$ across models.

Newtonian-like potentials. This setup allows us to generalize classical $N$-body systems to curved spacetime environments beyond the fat spaces Satorras et al. [2021]. The Hamiltonian consists of a kinetic term defined via the inverse Schwarzschild metric and a classical pairwise potential:
$H(\mathbf{q}^N, \mathbf{p}^N) = \frac{1}{2M} \sum_{i=1}^{N} \sum_{\mu,\nu=0}^{3} g^{\mu\nu}(\mathbf{q}^i) p_i^\mu p_i^\nu - \sum_{i<j} \frac{GM^2}{\sqrt{\|\bar{\mathbf{q}}^i - \bar{\mathbf{q}}^j\|_E^2 + \varepsilon^2}}$. The geodesic structure
of the spacetime introduces non-Euclidean curvature effects in the momentum transport, while inter-body forces remain Newtonian-like. We implement a symplectic leapfrog integrator adapted to relativistic Hamiltonian flow and simulate multi-body systems initialized near stable orbital radii. The results clearly indicate that Euclidean methods struggle to model particle behavior in curved geometry. In contrast, our proposed HDM achieves superior reconstruction accuracy and substantially lower energy drift in Table 1, reflecting improved alignment with the intrinsic geometry of the system.

**Ablation Study.** We assess the numerical stability and scalability of our model via two ablation criteria: (i) energy conservation, and (ii) robustness across varying system sizes. Figure 4 shows that our method yields significantly lower energy drift compared to Euclidean baselines (*i.e.*, SympHNN), indicating better consistency with the underlying geometric structure. In addition, Table 2 reports how performance degrades as the number of spin particles increases. While both models exhibit reduced accuracy for larger $N$, the proposed gauge-equivariant model remains consistently more stable. For instance, while both models experience increasing error as $N$ grows, the non-equivariant variant exhibits a sharp deterioration at $N = 10$, with over a sevenfold increase in ADE relative to $N = 3$. In contrast, the gauge-equivariant model maintains a more gradual degradation, reflecting improved scalability under growing system complexity.

| $N$ | $\mathcal{G}$-equiv | Non-equiv |
|---|---|---|
| 3 | 0.019 | 0.024 (+24%) |
| 5 | 0.097 | 0.103 (+6%) |
| 10 | 0.120 | 0.173 (+44%) |
| 20 | 0.148 | 0.169 (+14%) |

Table 2: Performance degradation as the number of spin particles increases.

## 6 Conclusion

This work presented *Neural Hamiltonian Diffusion* (NHD), a unified framework that integrates geometry-aware diffusion processes with structure-preserving Hamiltonian learning. We formulated a diffusion process lifted to the frame bundle and constructed neural Hamiltonian vector fields that are equivariant under frame transformations. We provided theoretical results characterizing the generalization properties of the proposed method, including uniform-in-time bounds and frame-wise deviation under gauge transformations. Experiments results across diverse scientific domains demonstrated that our NHD consistently improves physical fidelity and predictive stability compared to Euclidean or non-Hamiltonian baselines.

# Acknowledgments

This work was supported by ICT Creative Consilience Program through the Institute of Information & Communications Technology Planning & Evaluation (IITP) grant funded by the Korea government (MSIT) (IITP-2025-RS-2020-II201819)

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

# A Backgrounds

## A.1 Stochastic Riemannian Geometry, Hamiltonian Dynamics

Let $\mathcal{M}$ be a smooth $d$-dimensional Riemannian manifold with metric $g$. The *tangent bundle $T\mathcal{M}$* is the disjoint union of all tangent spaces $T_x\mathcal{M}$ for $x \in \mathcal{M}$. The *orthonormal frame bundle $\mathcal{O}(\mathcal{M})$* is a principal $O(n)$-bundle over $\mathcal{M}$, where each point $\mathbf{U} \in \mathcal{O}(\mathcal{M})$ is an ordered orthonormal basis $(e_1, \ldots, e_n)$ of $T_x\mathcal{M}$ at some $x \in \mathcal{M}$. The canonical projection $\pi : \mathcal{O}(\mathcal{M}) \to \mathcal{M}$ maps a frame to its base point. This allows lifting trajectories from $\mathcal{M}$ to $\mathcal{O}(\mathcal{M})$ in a geometrically structured way. A *connection* on $\mathcal{O}(\mathcal{M})$ decomposes the tangent space $T_U\mathcal{O}(\mathcal{M})$ into *vertical* and *horizontal* components:

$$T_U\mathcal{O}(\mathcal{M}) = \mathrm{H}_U \oplus V_U.$$

The *connection form* $\omega \in \Omega^1(\mathcal{O}(\mathcal{M}); \mathfrak{so}(n))$ is a Lie algebra-valued 1-form satisfying:

- $\omega(A^*) = A$ for all $A \in \mathfrak{so}(n)$, where $A^*$ is the fundamental vertical vector field,
- $R_g^*\omega = \mathrm{Ad}_{g^{-1}}\omega$ under the right action $R_g$ of $g \in O(n)$.

A vector field $X$ on $\mathcal{M}$ lifts *horizontally* to $\mathcal{O}(\mathcal{M})$ if $\omega(\tilde{X}) = 0$. This horizontal lift allows stochastic processes on $\mathcal{M}$ to be lifted into $\mathcal{O}(\mathcal{M})$ while preserving the connection structure.

Let $\mathbf{X}_t \in \mathcal{M}$ be a semimartingale on a smooth manifold $\mathcal{M}$. Let $\mathbf{U}_t \in \mathcal{O}(\mathcal{M})$ denote a process on the frame bundle such that $\pi(\mathbf{U}_t) = \mathbf{X}_t$. We say $\mathbf{U}_t$ is the *horizontal lift* of $\mathbf{X}_t$ if it satisfies the following condition from Hsu [2002]:

$$d\mathbf{U}_t = \sum_{i=1}^{n} \mathrm{H}_i(\mathbf{U}_t) \circ d\mathbf{B}_t^i, \quad U_0 = u,$$

where $\{\mathrm{H}_i\}$ is the canonical horizontal vector field associated with the standard basis vectors $e_i \in \mathbb{R}^n$, and $\mathbf{B}_t$ is an $\mathbb{R}^n$-valued Brownian motion. This construction allows noise to be defined canonically on $\mathbb{R}^n$, lifted horizontally via $\mathrm{H}_i$, and transported on the manifold through $\mathbf{U}_t$. The projected process $\mathbf{X}_t = \pi(\mathbf{U}_t)$ then inherits stochastic dynamics that respect the geometry induced by the connection.

An orthonormal frame $\mathbf{U} \in \mathcal{O}(\mathcal{M})$ at $x = \pi(\mathbf{U})$ is represented as an isometry $\mathbf{U} : \mathbb{R}^n \to T_x\mathcal{M}$, i.e., for a standard basis vector $e_i \in \mathbb{R}^n$, $\mathbf{U}e_i = e_i^x \in T_x\mathcal{M}$. Thus, the process $\mathbf{U}_t$ evolves according to the geometry of $\mathcal{M}$ with stochastic noise applied in the frame coordinates and mapped into the tangent bundle via horizontal transport. This perspective ensures that noise is not only intrinsic to the manifold but also compatible with the Levi-Civita connection and geometric constraints of $\mathcal{M}$.

**Hamiltonian Vector Fields on Manifolds.** Let $\mathrm{H} : T^*\mathcal{Q} \to \mathbb{R}$ be a Hamiltonian. The *Hamiltonian vector field $X_\mathrm{H}$* is defined implicitly via the symplectic form $\omega$ on $T^*\mathcal{Q}$, *i.e.*, $\iota_{X_\mathrm{H}}\omega = d\mathrm{H}$. In local Darboux coordinates $(q^i, p_i)$, $X_\mathrm{H}$ takes the standard form:

$$X_H = \sum_i \left( \frac{\partial H}{\partial p_i} \frac{\partial}{\partial q^i} - \frac{\partial H}{\partial q^i} \frac{\partial}{\partial p_i} \right).$$

This is the core generator of deterministic Hamiltonian evolution and provides the basis for its stochastic extension. For smooth functions $f, g : T^*\mathcal{Q} \to \mathbb{R}$, the *Poisson bracket* is defined as:

$$\{f, g\} := \omega(X_f, X_g) = \sum_i \left( \frac{\partial f}{\partial q^i} \frac{\partial g}{\partial p_i} - \frac{\partial f}{\partial p_i} \frac{\partial g}{\partial q^i} \right).$$

**Horizontal Lift and Connection form.** In Section 2 and Figure 1, we described the evolution of lifted Hamiltonian dynamics on the orthonormal frame bundle $\mathcal{O}(M)$. The key construction relies on decomposing a lifted vector field into horizontal and vertical components with respect to the principal connection. Let $\widehat{G}_\theta(\mathbf{U}_t) \in T_{\mathbf{U}_t}\mathcal{O}(M)$ denote the full lifted vector field constructed from the Hamiltonian flow $\{\mathbf{m}, \mathrm{H}_\theta\}$. This field is not guaranteed to lie in the horizontal distribution $\mathrm{H}_{\mathbf{U}_t} := \ker \omega_{\mathbf{U}_t}$ and must be projected to ensure that the resulting SDE respects the manifold connection structure. The canonical projection is defined via the connection 1-form $\omega \in \Omega^1(\mathcal{O}(M); \mathfrak{so}(d))$, which satisfies:

$$\omega_{\mathbf{U}_t}(V_{\mathbf{U}_t}) = \mathfrak{h} \in \mathfrak{so}(d), \qquad \omega_{\mathbf{U}_t}(\mathrm{H}_{\mathbf{U}_t}) = 0.$$

The vertical component is extracted using $\omega$, and the projection onto the horizontal space is given by:

$$G_\theta^{\mathrm{Hor}}(\mathbf{U}_t) = \widehat{G}_\theta(\mathbf{U}_t) - \omega_{\mathbf{U}_t}\left(\widehat{G}_\theta(\mathbf{U}_t)\right)^\sharp, \tag{13}$$

where $\omega_{\mathbf{U}_t}(\widehat{G}_\theta)^\sharp \in V_{\mathbf{U}_t}$ denotes the vertical lift of the Lie algebra element associated to the vertical component, and $\sharp$ maps Lie algebra elements to fundamental vector fields. This ensures that the resulting direction $G_\theta^{\mathrm{Hor}} \in \mathrm{H}_{\mathbf{U}_t}$ lies entirely in the horizontal distribution, satisfying the condition $\omega(G_\theta^{\mathrm{Hor}}) = 0$. In local coordinates used in Eq. (5), the vertical component explicitly appears as the second term involving the connection tensor $[\Gamma^{\mathcal{M}}]^\flat$ and the vectorized frame $\mathrm{vec}(E)$. The subtraction in Eq. (5) therefore realizes the above projection in local form, decomposing the lifted vector field into:

$$\widehat{G}_\theta = G_\theta^{\mathrm{Hor}} + G_\theta^{\mathrm{Ver}}, \qquad G_\theta^{\mathrm{Ver}} := \omega\left(\widehat{G}_\theta\right)^\sharp. \tag{14}$$

This decomposition plays a crucial role in ensuring that the Stratonovich increment $d\mathbf{U}_t = G_\theta^{\mathrm{Hor}}(\mathbf{U}_t) \circ d\mathbf{B}_t$ evolves along a direction consistent with the geometry of $T^*Q$. This geometric consistency is essential for transporting noise on the manifold without introducing spurious curvature-induced distortions, and forms the foundation of the gauge-consistent stochastic Hamiltonian dynamics defined in this paper. Note that the proof of Proposition 2.2 will deliver the detailed calculation of deriving the vanishing connection 1-form to show the validity.

**Infinitesimal Generator and Fokker-Planck Equation**. The stochastic process $\mathbf{X}_t \in \mathcal{M} := T^*Q$ governed by the Stratonovich SDE proposed in main manuscript

$$d\mathbf{X}_t = \{\mathbf{m}, \mathrm{H}_\theta\}(\mathbf{X}_t) \circ dB_t,$$

defines a diffusion process on the symplectic manifold $\mathcal{M}$ with Hamiltonian vector field $\{\mathbf{m}, \mathrm{H}_\theta\}$. The corresponding infinitesimal generator $\mathcal{L}_\theta$ acts on smooth test functions $f \in C^\infty(\mathcal{M})$ as:

$$\mathcal{L}_\theta f(x) = \frac{1}{2}\sum_{i=1}^{2d} \{\mathbf{m}, \mathrm{H}_\theta\}^i(x) \cdot \partial_i\left(\{\mathbf{m}, \mathrm{H}_\theta\}^i(x) \cdot \partial_i f(x)\right),$$

where the index $i$ runs over local coordinates $(q^1, \ldots, q^d, p^1, \ldots, p^d)$ on $T^*Q$, and $\{\mathbf{m}, \mathrm{H}_\theta\}^i$ denotes the $i$-th component of the Hamiltonian vector field. Let $\rho_t(x) \in \mathrm{Dens}(\mathcal{M})$ denote the time-dependent probability density function of $\mathbf{X}_t$. Then, the evolution of $\rho_t$ is governed by the Fokker–Planck equation associated with the generator $\mathcal{L}_\theta$:

$$\frac{\partial \rho_t(x)}{\partial t} = \mathcal{L}_\theta^* \rho_t(x), \tag{15}$$

where $\mathcal{L}_\theta^*$ is the formal adjoint of $\mathcal{L}_\theta$ in the $L^2(\mathcal{M})$ sense. Explicitly, using integration by parts, this yields:

$$\frac{\partial \rho_t(x)}{\partial t} = \frac{1}{2}\sum_{i=1}^{2d} \partial_i\left(\{\mathbf{m}, \mathrm{H}_\theta\}^i(x) \cdot \partial_i\left(\{\mathbf{m}, \mathrm{H}_\theta\}^i(x) \cdot \rho_t(x)\right)\right),$$

where the density $\rho_t$ allows the Radon-Nikodym derivative with respect to probability measure $\mathbb{P}_t$ by the formula $d\mathbb{P}_t = \rho_t d\mathbf{x}$ with Lebesgue measure $d\mathbf{x}$. This formulation describes how the probability mass of the stochastic process spreads over the symplectic manifold under the influence of the geometry-aware Hamiltonian noise.

Table 3: Comparison between Euclidean HNNs and Neural Hamiltonian Diffusions (Ours).

| Comparison Item | Euclidean HNNs | Neural Hamiltonian Diffusions (Ours) |
|---|---|---|
| Space | $\mathbb{R}^{2d}$ (flat) | General manifold $T^*\mathcal{Q}$ |
| Structure | Fixed symplectic form $J$ | Poisson structure from lifted geometry |
| Noise | None (deterministic) | Intrinsic stochasticity (via horizontal lift) |
| Energy Conservation | Pathwise $\dot{H} = 0$ | Statistical: $L_\theta \pi = 0$ |

# B  Experimental Setup

## B.1  Relativistic Dynamics

We consider a Hamiltonian system defined on full spacetime phase space $(\mathcal{M}, g)$, where $\mathcal{M}$ is a Lorentzian manifold with Schwarzschild metric $g$. The canonical geodesic Hamiltonian governing $N$ interacting particles takes the form:

$$H(\mathbf{x}, \mathbf{p}) = \sum_{i=1}^{N} \frac{1}{2M} g^{\mu\nu}(x_i) \, p_\mu^i p_\nu^i - \sum_{i<j} \frac{GM^2}{\sqrt{\|\vec{x}_i - \vec{x}_j\|^2 + \varepsilon^2}}, \tag{16}$$

where $g^{\mu\nu}(x)$ is the inverse Schwarzschild metric and $p_\mu^i$ is the four-momentum conjugate to the spacetime coordinate $x_i^\mu = (t_i, r_i, \theta_i, \phi_i)$. The second term encodes softened pairwise gravitational interactions.

The inverse Schwarzschild metric in spherical coordinates is given by:

$$g^{\mu\nu}(r, \theta) = \begin{pmatrix} -\left(1 - \frac{2GM}{r}\right)^{-1} & 0 & 0 & 0 \\ 0 & \left(1 - \frac{2GM}{r}\right) & 0 & 0 \\ 0 & 0 & \frac{1}{r^2} & 0 \\ 0 & 0 & 0 & \frac{1}{r^2 \sin^2\theta} \end{pmatrix}. \tag{17}$$

The dynamics follow the relativistic Hamilton equations:

$$\frac{dx_i^\mu}{dt} = \frac{\partial H}{\partial p_\mu^i}, \qquad \frac{dp_\mu^i}{dt} = -\frac{\partial H}{\partial x_i^\mu}. \tag{18}$$

We summarize the physical-to-code variable mapping as:

$$q = (r, \theta, \phi) \qquad \text{Position in Schwarzschild coordinates,}$$
$$\dot{q} = \frac{\partial H}{\partial p} \qquad \text{Relativistic velocity (spatial),}$$
$$p = (p_0, p_1, p_2, p_3) \qquad \text{Canonical four-momentum,}$$
$$\dot{p} = -\frac{\partial H}{\partial q} + \nabla V \qquad \text{Force from geometry and pairwise interaction.}$$

Each particle $i \in \{1, \ldots, N\}$ is initialized with a four-position $\mathbf{q}_i = (t_i, r_i, \theta_i, \phi_i)$ drawn from:

$$r_i \sim \mathcal{N}(8.0, 0.01^2),$$
$$\theta_i \sim \mathcal{N}(\pi/2, 0.005^2),$$
$$\phi_i^{(0)} \sim \mathcal{U}[\pi, 1.3\pi], \quad \phi_i \sim \mathcal{N}(\phi_i^{(0)}, 0.01^2),$$

centered around a stable orbital radius with small angular spread.

Initial four-momentum is sampled to induce slightly perturbed circular orbits:

$$p_i = m \cdot \begin{bmatrix} \dfrac{1}{1 - 2GM/r_i} \\ -0.1 \\ 0 \\ \varepsilon_\phi \end{bmatrix}, \quad \text{with } \varepsilon_\phi \sim \mathcal{N}(0, 0.01^2).$$

Table 4: Initial state (reproducible random seed 42).

| Body | $x$ | $y$ | $z$ | $w$ | $\omega_x$ | $\omega_y$ | $\omega_z$ |
|------|------|------|------|------|------|------|------|
| 1 | $-0.496$ | $0.647$ | $0.576$ | $-0.011$ | $0.228$ | $-0.238$ | $0.040$ |
| 2 | $0.261$ | $0.400$ | $0.812$ | $-0.314$ | $-0.141$ | $0.136$ | $-0.113$ |
| 3 | $-0.183$ | $0.861$ | $0.318$ | $0.351$ | $-0.119$ | $-0.438$ | $0.176$ |

**Frame Bundle Structure and Spatial Diffusion.** Let $(\mathcal{M}, g)$ be a pseudo-Riemannian manifold with signature $(-, +, +, +)$. The pseudo-orthonormal frame bundle $\mathcal{O}^{(1,3)}(\mathcal{M})$ is a principal $SO(1, 3)$-bundle. A point in this bundle is written as:

$$\mathbf{U}_t = (x^\mu, e^\mu{}_a), \quad \text{where } g_{\mu\nu}(x)\, e^\mu{}_a\, e^\nu{}_b = \eta_{ab},$$

and $\eta = \mathrm{diag}(-1, 1, 1, 1)$ is the Minkowski metric. The frame $e^\mu{}_a$ forms a local orthonormal basis of $T_x\mathcal{M}$. To respect relativistic causality, we restrict stochastic diffusion to spatial directions $a = 1, 2, 3$. Let $\mathbf{B}_t = (B_t^{(1)}, B_t^{(2)}, B_t^{(3)})$ be Brownian noise on the spatial frame. The spatially-restricted horizontal SDE on the frame bundle is then:

$$d\mathbf{U}_t = \mathrm{G}_\theta^{\mathrm{Hor}}(\mathbf{U}_t) \circ \begin{pmatrix} 0 \\ d\mathbf{B}_t \end{pmatrix}, \qquad \pi(\mathbf{U}_t) = \mathbf{X}_t, \;\; \mathbf{U}_t \in \mathcal{O}^{(1,3)}(\mathcal{M}). \tag{19}$$

No perturbation is applied to the temporal component ($a = 0$), and the drift term is entirely deterministic, maintaining consistency with the Lorentzian structure. The simulation integrates this Hamiltonian system using a symplectic leapfrog method, with symbolic metric evaluation via SymPy. The background mass $M = 1.0$ determines the Schwarzschild radius $r_s = 2GM = 2.0$.

## B.2 Spin Dynamics

Our goal in the expereiment is to model the time-evolution of three mutually–interacting rigid–body spins living on the unit 3-sphere $\mathbb{S}^3 \subset \mathbb{R}^4$, simulate three dynamical regimes, and render the results through the Hopf fibration. Let $q = (x, y, z, w) \in \mathbb{S}^3$ be a unit quaternion representing a rigid rotation. We split $q$ into the complex pair $(z_1, z_2) \in \mathbb{C}^2$, $z_1 = x + iy$, $z_2 = z + iw$, so that $|z_1|^2 + |z_2|^2 = 1$. We treat each spin as a point mass with principal inertia $(2, 1, 0.5)$ and equip the system with the *pair-exchange Hamiltonian* defined as follows:

$$H\big(\{q_i, \omega_i\}_{i=1}^N\big) = \sum_{i=1}^N \frac{1}{2}\omega_i^\top I \omega_i - \frac{J}{2} \sum_{i \neq j} (q_i^\top q_j),$$

where $\omega_i \in \mathbb{R}^3$ is the spatial angular velocity, $I = \mathrm{diag}(2, 1, 0.5)$, and $J > 0$ promotes alignment. Then, the Hamilton equations read $\dot{q}_i = \frac{1}{2}\,\Omega(\omega_i)\,q_i$, $\dot{\omega}_i = I^{-1}\tau_i$, with $\Omega(\omega)$ and $\tau_i = -J \sum_{j \neq i}(q_i - q_j)_{1:3}$ the coupling torque. To simulate the dynamics, we employ an explicit Euler step of size $\Delta t = 0.02\,\mathrm{s}$ and *renormalize* $q_i$ to unit length after each step to avoid drift off $\mathbb{S}^3$. All runs start from the randomized seed quaternion/velocity ensemble (Table 4 for seed 42). Each trajectory contains $T = 300$ frames. While a unit quaternion $(z_1, z_2) \in \mathbb{S}^3 \subset \mathbb{C}^2$ projects via the Hopf projection $\pi(z_1, z_2) = \big(2\Re(z_1\bar{z}_2), 2\Im(z_1\bar{z}_2), |z_1|^2 - |z_2|^2\big) \in \mathbb{S}^2$., we display each trajectory in *polar arg–magnitude* coordinates of $(z_1, z_2)$: $(\theta_k, r_k) = (\arg z_k, |z_k|)$, $k = 1, 2$.

## B.3 Toroidal Protein Sequence

We convert the raw Cartesian trajectory contained in `traj-arrays.npz` into generalized coordinates $(\theta, \dot{\theta}, p, \dot{p})$ on the dihedral-torus to support Hamiltonian learning. The `.npz` file provides positions $\mathbf{x}_t \in \mathbb{R}^{N \times 3}$, velocities $\mathbf{v}_t$, and time stamps $t \in \mathbb{R}^+$. Dihedral angles $\boldsymbol{\theta}_t = (\phi_t, \psi_t, \omega_t) \in (-\pi, \pi]^3$ are computed from atomic coordinates using standard torsion definitions applied to atom quadruplets $\mathcal{A}^\phi, \mathcal{A}^\psi, \mathcal{A}^\omega \subset \{1, \ldots, N\}^4$, which are in turn extracted from the molecular topology file `traj-state0.pdb` via MDTRAJ. This topology file also provides element-wise atomic masses $\{m_i\}_{i=1}^N$ for moment of inertia computation.

We first estimate a uniform time step $\Delta t = \mathrm{mean}_k(t_{k+1} - t_k)$ in femtoseconds from the raw time array. Angular velocities for each torsion angle $\theta \in \{\phi, \psi, \omega\}$ are then computed by finite differencing:

| Variable | Symbol | Shape |
|---|---|---|
| Angular velocities | $\dot{\boldsymbol{\theta}}$ | $(T-1) \times 3$ |
| Angular momenta | $\boldsymbol{p}$ | $(T-1) \times 3$ |
| Momentum derivatives | $\dot{\boldsymbol{p}}$ | $(T-2) \times 3$ |

Table 5: Dimensions of the augmented physical variables.

$\dot{\theta}_k = (\theta_{k+1} - \theta_k)/\Delta t$, followed by periodic unwrapping using $\dot{\theta}_k \leftarrow \text{wrap}_{(-\pi,\pi]}(\dot{\theta}_k)$ where the wrap function applies $\text{atan2}(\sin\cdot, \cos\cdot)$ elementwise. The resulting angular velocity matrix $\dot{\boldsymbol{\theta}} \in \mathbb{R}^{(T-1)\times 3}$ is stored under the key `torsion_dots`. To compute the scalar moment of inertia $I_k$ for a given torsion at time $t_k$, we approximate the rotation axis as the normalized vector $\mathbf{e} = (\mathbf{x}_{a_2} - \mathbf{x}_{a_1})/\|\cdot\|$ and use only the two terminal atoms to define transverse distances. Specifically,

$$I_k = m_{a_0}\, r_\perp^2(\mathbf{x}_{a_0}) + m_{a_3}\, r_\perp^2(\mathbf{x}_{a_3}), \qquad r_\perp(\mathbf{r}) = \|(\mathbf{r} - \mathbf{x}_{a_1}) - [(\mathbf{r} - \mathbf{x}_{a_1}) \cdot \mathbf{e}]\, \mathbf{e}\|.$$

This produces a single scalar inertia for each dihedral and time step. Using this, we compute the conjugate angular momentum as $p_k = I_k\, \dot{\theta}_k$ and aggregate the result as a matrix $\boldsymbol{p} \in \mathbb{R}^{(T-1)\times 3}$ stored as `torsion_momentum`. To obtain generalized forces, we compute the time derivatives of angular momentum by finite differencing:

$$\dot{p}_k = \frac{p_{k+1} - p_k}{\Delta t}, \qquad \dot{p}_k \leftarrow \text{wrap}_{(-\pi,\pi]}(\dot{p}_k),$$

where we again apply periodic unwrapping. This finally yields the data $\dot{\boldsymbol{p}} \in \mathbb{R}^{(T-2)\times 3}$. Table 5 summarizes the shape of the augmented tensors.

## B.4 Experimental Details

All experiments were conducted on a single NVIDIA RTX 5090 GPU using Python 3.11 and PyTorch 2.1.0 with CUDA 11.8 support. The proposed framework is evaluated on a pre-processed protein trajectory dataset embedded in a curved configuration space. We use an 80%/20% temporal split for training and testing. Each sub-trajectory consists of $0.8T$ frames: the initial frame $t_0$ serves as the input, and the following $0.8T$ frames $(t_{1:0.8T})$ are used for supervision. The model input and target sequences include generalized coordinates, velocities, momenta, and their time derivatives:

$$(q,\, \dot{q},\, p,\, \dot{p}) \in \mathbb{R}^{T \times 3},$$

where $T$ denotes the total number of time steps in each sequence.

The neural architecture in Algorithm 1 is a *Gauge-Equivariant Transformer UNet* designed to model Hamiltonian vector fields on curved manifolds by incorporating symmetry-preserving inductive biases. The input to the network is a concatenation of configuration and momentum coordinates $(q,p) \in \mathbb{R}^{2d}$, transformed into a canonical local gauge frame using a Cholesky-based projection with gauge matrix $G \in \mathbb{R}^{d \times d}$. This projected input is passed through a linear embedding layer and fused with a temporal encoding via sinusoidal or MLP-based `TimeEmbedding`.

The model consists of an encoder–decoder Transformer with $L = 16$ residual blocks, each using multi-head self-attention, GELU activations, and layer normalization. Skip connections and projection layers link encoder and decoder stages. Predictions for time derivatives are generated in a local gauge frame and mapped back to the global frame using a Cholesky-based projection, ensuring gauge equivariance. The network uses a hidden size of 128 and contains approximately 3M parameters. Training is performed for $10^5$ epochs using the Adam optimizer with learning rate $10^{-4}$ and batch size 128. The loss combines a local alignment term and a long-range reconstruction objective.

$$\mathcal{L}(\theta) = \mathbb{E}_{t \sim \mathcal{U}[0,T]}\Big[\underbrace{\lambda\,\big\|\,[\widehat{\dot{q}}_t, \widehat{\dot{p}}_t]^\top - [\dot{q}_t, \dot{p}_t]^\top\,\big\|_{g_{\mathcal{M}}}^2}_{\mathcal{L}_{\text{align}}} + \underbrace{\frac{1}{N_{\text{tar}}}\sum_{i=1}^{N_{\text{tar}}}\big\|\text{wrap}\big(\widehat{q}_{t,i} - q_{t,i}\big)\big\|_2^2}_{\mathcal{L}_{\text{recon}}}\Big], \qquad \lambda = 0.1.$$

To simulate future trajectories, we employ a geometry-aware simulator that integrates stochastic Hamiltonian dynamics via Stratonovich SDEs on the cotangent bundle. This simulator leverages

---

**Algorithm 1** GAUGE EQUIVARIANT TRANSFORMER UNET

---

1: **Input:** $x = [q^N, p^N] \in^{2dN}$, time $t$, bundle metric $G$
2: **Output:** $[\widehat{q}^N, \widehat{p}^N]$

3: *// Canonical gauge transform*
4: $q_{\text{can}} \leftarrow G^\top q^N$, $\quad x_{\text{can}} \leftarrow [q_{\text{can}}, p^N]$

5: *// Encoder–decoder trunk (shared context)*
6: $h \leftarrow \texttt{Linear}(x_{\text{can}}) + \texttt{TimeEmbed}(t)$
7: $h \leftarrow \texttt{Unsqueeze}(h, 1)$
8: **for** $i = 1$ **to** $L$ **do** $h \leftarrow \texttt{EncBlock}_i(h)$, $\texttt{enc}_i \leftarrow h$
9: **end for**
10: **for** $i = 1$ **to** $L$ **do** $h \leftarrow \texttt{Concat}(h, \texttt{enc}_{L-i+1})$; $h \leftarrow \texttt{DecProj}_i(h)$; $h \leftarrow \texttt{DecBlock}_i(h)$
11: **end for**
12: $c \leftarrow \texttt{GlobalPool}(h)$                            // context vector shared by all potentials

13: *// Shared trunk $\mathcal{T}_\phi$ and two heads $\mathcal{H}_{sp}$, $\mathcal{H}_{pair}$*
14: **for** $i = 1$ **to** $N$ **do**                                 ▷ single-particle branch
15:      $\phi_i \leftarrow \mathcal{T}_\phi([q^i, p^i], c)$                         // shared weights
16:      $E_{\text{sp}}^i \leftarrow \mathcal{H}_{\text{sp}}(\phi_i)$                           // head 1
17: **end for**
18: **for all** pairs $(i, j)$, $i < j$ **do**                       ▷ pairwise branch
19:      $\phi_{ij} \leftarrow \mathcal{T}_\phi([q^i, q^j], c)$                      // same trunk
20:      $E_{\text{pair}}^{ij} \leftarrow \mathcal{H}_{\text{pair}}(\phi_{ij})$                   // head 2
21: **end for**

22: $\text{KE} \leftarrow \frac{1}{2} \sum_i (p^i)^\top G^{-1} p^i$                        // kinetic
23: $\text{SP} \leftarrow \sum_i E_{\text{sp}}^i$, $\quad \text{PI} \leftarrow \sum_{i<j} E_{\text{pair}}^{ij}$
24: $H_\theta = \text{KE} + \text{SP} + \text{PI}$

25: $\widehat{q}^N = \nabla_{p^N} H_\theta$, $\widehat{p}^N = \nabla_{q^N} H_\theta$
26: $\widehat{q}^N \leftarrow G \widehat{q}^N$                                 // back to global frame
27: **return** $[\widehat{q}^N, \widehat{p}^N]$

---

gauge-equivariant drift fields and lifts the dynamics into a frame bundle, where the noise is transported horizontally. Chart transitions are handled using a manifold-aware update rule. The numerical integrator uses a fixed step size $\Delta t = 2.0$ fs and isotropic Gaussian noise of magnitude $10^{-2}\sqrt{\Delta t}$. Performance is evaluated using two common metrics: the *Average Displacement Error* (ADE) and the *Final Displacement Error* (FDE), defined as

$$\text{ADE} = \frac{1}{N\,T_{\text{tar}}} \sum_{n,t} \|\widehat{q}_{n,t} - q_{n,t}\|^2, \qquad \text{FDE} = \frac{1}{N} \sum_n \|\widehat{q}_{n,T_{\text{tar}}} - q_{n,T_{\text{tar}}}\|^2.$$

Random seeds for `torch`, `numpy`, and `random` are fixed to 42 for reproducibility. The codebase and configuration files will be made publicly available at `https://github.com/Anonymous/HDM`.

**Simulation of Neural Hamiltonian Diffusion.** Algorithm 2 outlines the simulation process of our Hamiltonian Diffusion Model (HDM) based on Proposition 2.2. Starting from an initial state $(q_0, p_0)$, we construct a lifted representation $U_0$ on the frame bundle using the inverse metric and its Cholesky decomposition. At each step, the model predicts a gauge-equivariant Hamiltonian drift, and isotropic noise is projected onto the horizontal space to ensure geometric consistency. The state is updated via a Stratonovich integrator that respects the manifold structure, and chart transitions are handled as needed. The algorithm outputs both the lifted trajectory on the frame bundle and its projection onto the base manifold, enabling structured simulation over curved geometric spaces.

**Initialisation of the Lifted State $U_0 = (x_0, E_0) \in \mathcal{O}(\mathcal{M})$.** The horizontal SDE of Proposition 2.2 requires an initial condition on the *orthonormal-frame bundle*. This amounts to choosing *(i)* a base point $x_0 \in \mathcal{M}$—which fixes the particle's initial configuration—and *(ii)* an orthonormal frame $E_0 \in \text{SO}(T_{x_0}\mathcal{M})$ that serves as the local gauge in which all subsequent tangent-space computations are expressed.

**Algorithm 2** SIMULATE NEURAL HAMILTONIAN DIFFUSION($q_0, p_0, t_0, T, n_f$; $H_\theta$)

---

**Require:** Initial state $(q_0, p_0) \in T^*Q$, start–end times $(t_0, T)$, # Stratonovich steps $n_f$, parameterized Hamiltonian $H_\theta$

**Ensure:** Lifted trajectory $\{\mathbf{U}_t\}_{t_0 \leq t \leq T}$ and its projection $\{\mathbf{X}_t = \pi(\mathbf{U}_t)\}$

1: **// *Initial frame lift***
2: $g_\sharp \leftarrow g_\sharp(q_0, p_0)$        // inverse bundle metric
3: $\nu_0 \leftarrow \text{chol}(g_\sharp)$        // local orthonormal frame
4: $U_0 \leftarrow (q_0, p_0, \text{vec}(\nu_0))$        // coordinates on $\mathcal{O}(T^*Q)$
5: $\mathcal{C}_0 \leftarrow \mathbf{1}_d$        // initial chart index
6: **// *Stratonovich SDE integration***
7: **for** $k = 0$ to $n_f - 1$ **do**
8:      $t \leftarrow t_0 + k\,\Delta t, \quad \Delta t \leftarrow \frac{T - t_0}{n_f}$
9:      $q_t, p_t, \mathbf{U}_t \leftarrow \text{unpack}(U_k)$
10:      *// Horizontal Hamiltonian drift*
11:      $[w_q, w_p] \leftarrow \text{MODELFORWARD}([q_t, p_t], t, \mathbf{U}_t)$
12:      $G_\theta^{\text{hor}}(U_k) \leftarrow [w_q, -w_p]$        // $\{\mathbf{m}, H_\theta\}$ part
13:      *// Horizontal diffusion term*
14:      $\mathcal{H}_t \leftarrow H_{\text{frame}}(q_t, \mathbf{U}_t)$        // horizontal projector
15:      $\xi \sim \mathcal{N}(0, I_{2d}), \quad \text{sto} = \sqrt{\Delta t}\,\mathcal{H}_t \xi$
16:      *// Stratonovich increment*
17:      $U_{k+1} \leftarrow U_k + G_\theta^{\text{hor}}(U_k)\,\Delta t + \text{sto}$
18:      *// (optional) Chart update on frame bundle*
19:      $\mathcal{C}_{k+1}, \mathbf{U}_{k+1} \leftarrow \text{CHARTUPDATE}(U_{k+1}, \mathcal{C}_k)$
20: **end for**
       **return** $\{U_k\}_{k=0}^{n_f}, \quad \{X_k = \pi(\mathbf{U}_k)\}$

---

**1. Base point $x_0$.** In practice $x_0$ is dictated by the task: for trajectory prediction one sets $x_0 = x_{\text{data}}$ (the observed configuration at time 0); for sampling or controlled experiments one may draw $x_0$ from a prescribed distribution on $\mathcal{M}$ (e.g. the uniform measure on $S^2$).

**2. Orthonormal frame $E_0$.** Given a coordinate chart $m = (q^1, \ldots, q^d)$ around $x_0$ with metric matrix $g(x_0)$, one constructs $E_0$ by orthonormalising the coordinate basis via the Gram–Schmidt (or Cholesky) procedure:

$$E_0 = \begin{bmatrix} e_1 & \cdots & e_d \end{bmatrix}, \qquad \langle e_i, e_j \rangle_{g(x_0)} = \delta_{ij}.$$

For the sphere example ($d = 2$) with $(\chi, \varphi)$ coordinates one obtains

$$e_1 = \partial_\chi\big|_{x_0}, \qquad e_2 = \frac{1}{\sin\chi_0} \partial_\varphi\big|_{x_0}, \qquad E_0 = \begin{bmatrix} 1 & 0 \\ 0 & 1/\sin\chi_0 \end{bmatrix}.$$

**3. Phase–space variables.** If the model evolves on $T^*\mathcal{M}$ one also specifies the initial momentum $p_0 \in T^*_{x_0}\mathcal{M}$. Typical choices are (a) the empirical momentum if one starts from real data, or (b) a draw from the canonical Gibbs distribution $p_0 \sim \mathcal{N}(0, g^{-1}(x_0))$, which is consistent with the kinetic term $\frac{1}{2} p^\top g^{-1} p$ in the Hamiltonian.

**4. Vectorised form for the SDE.** For implementation the frame is flattened, $\text{vec}(E_0) \in \mathbb{R}^{d^2}$, and concatenated with $(q_0, p_0)$ to produce the full initial vector fed into Algorithm **??**. Because $E_0$ is already orthonormal, the integrator starts in the horizontal sub-bundle, and the structural properties guaranteed by Proposition 2.2 are preserved from the first step onward without any corrective projection.

## C   Lemmas

**Lemma C.1.** *Let $\Gamma := \Gamma^i_{jk}$ be the connection (i.e., Christoffel symbols) of a smooth configura-tion manifold $(\mathcal{Q}, g)$, and let $\mathcal{R}^i{}_{jkl}$ be the components of the Riemann curvature tensor. Let us denote $\|\Gamma\|_\infty := \sup_{x \in \mathcal{Q}} \max_{i,j,k} |\Gamma^i_{jk}(x)|$, and $\|\partial\Gamma\|_\infty := \sup_{x \in \mathcal{Q}} \max_{i,j,k,l} |\partial_k \Gamma^i_{jl}(x)|$. Then the following sup-norm inequality holds:*

$$\|\mathcal{R}\|_\infty \leq 2\|\partial\Gamma\|_\infty + 2\|\Gamma\|_\infty^2, \qquad \|\Gamma\|_\infty \equiv 0 \longrightarrow \|\mathcal{R}\|_\infty \equiv 0.$$

*Proof of Lemma C.1.* Recall that, in a local coordinate chart on a smooth Riemannian manifold $(\mathcal{Q}, g)$, the components of the Riemann curvature tensor are

$$\mathcal{R}^i{}_{jkl} = \partial_k \Gamma^i_{jl} - \partial_l \Gamma^i_{jk} + \Gamma^i_{km} \Gamma^m_{jl} - \Gamma^i_{lm} \Gamma^m_{jk},$$

where $\Gamma^i_{jk}$ are the Christoffel symbols. Define the sup-norms

$$\|\Gamma\|_\infty := \sup_{x \in \mathcal{Q}} \max_{i,j,k} \left| \Gamma^i_{jk}(x) \right|, \qquad \|\partial\Gamma\|_\infty := \sup_{x \in \mathcal{Q}} \max_{i,j,k,l} \left| \partial_k \Gamma^i_{jl}(x) \right|,$$

and

$$\|\mathcal{R}\|_\infty := \sup_{x \in \mathcal{Q}} \max_{i,j,k,l} \left| \mathcal{R}^i{}_{jkl}(x) \right|.$$

For the derivative part of $\mathcal{R}^i{}_{jkl}$, we have

$$\left| \partial_k \Gamma^i_{jl} - \partial_l \Gamma^i_{jk} \right| \leq \left| \partial_k \Gamma^i_{jl} \right| + \left| \partial_l \Gamma^i_{jk} \right| \leq 2 \|\partial\Gamma\|_\infty.$$

For other quadratic parts, we control the terms by showing that

$$\left| \Gamma^i_{km} \Gamma^m_{jl} - \Gamma^i_{lm} \Gamma^m_{jk} \right| \leq \left| \Gamma^i_{km} \Gamma^m_{jl} \right| + \left| \Gamma^i_{lm} \Gamma^m_{jk} \right| \leq \|\Gamma\|_\infty^2 + \|\Gamma\|_\infty^2 = 2 \|\Gamma\|_\infty^2.$$

Applying above results and then we take the maximum over all indices at each point to have

$$|\mathcal{R}^i{}_{jkl}(x)| \leq 2 \|\partial\Gamma\|_\infty + 2 \|\Gamma\|_\infty^2, \quad \forall x \in \mathcal{Q}.$$

Finally, this gives

$$\|\mathcal{R}\|_\infty \leq 2 \|\partial\Gamma\|_\infty + 2 \|\Gamma\|_\infty^2.$$

If $\|\Gamma\|_\infty \equiv 0$ (so every $\Gamma^i_{jk}$ vanishes identically), then $\partial_k \Gamma^i_{jl} \equiv 0$ as well, and $\mathcal{R}^i{}_{jkl} \equiv 0$ by $(*)$. Consequently $\|\mathcal{R}\|_\infty \equiv 0$. This concludes the proof. $\qquad\square$

**Lemma C.2.** *Let $(M, g)$ be a Riemannian manifold. Fix a smooth reference curve $\gamma : [0, T] \to M$ and define*

$$F_t(y) := \exp^{-1}_{\gamma(t)}(y) \in T_{\gamma(t)}M, \qquad J_t := F_t(\mathbf{X}_t),$$

*where $\mathbf{X}_t$ is a solution to the proposed Stratonovich Hamiltonian Diffusion Model (HDM). Assume that $X_H(\gamma(t)) = 0$ for all $t \in [0, T]$. Then, to first order in the stochastic differential, we have*

$$D \exp^{-1}_{\gamma(t)}(d\mathbf{X}_t) \approx \nabla_{J_t} \{\mathbf{m}, \mathrm{H}_\theta\}(\gamma(t)) \circ dB_t. \tag{20}$$

*Proof.* We start by recalling the behavior of the exponential map and its derivative. For $v \in T_{\gamma(t)}M$, define the geodesic $\gamma_v(s) := \exp_{\gamma(t)}(sv)$. Given $\xi \in T_{\exp_{\gamma(t)}(v)}M$, the differential of the exponential map satisfies

$$D \exp_{\gamma(t)}(v)[\xi] = J_\xi(1), \tag{21}$$

where $J_\xi$ denotes the unique Jacobi field along $\gamma_v$ satisfying the initial conditions

$$J_\xi(0) = 0, \qquad \nabla_{\dot{\gamma}_v} J_\xi(0) = \xi.$$

At the origin $v = 0$, the exponential map behaves simply:

$$D \exp_{\gamma(t)}(0) = \mathrm{Id}, \qquad D^2 \exp_{\gamma(t)}(0) = 0.$$

This tells us that near $v = 0$, the differential $D\exp_{\gamma(t)}(v)$ is close to the identity up to second-order terms. Thus, for $y$ close to $\gamma(t)$ (writing $v = \exp_{\gamma(t)}^{-1}(y)$), the differential of the inverse exponential map satisfies

$$D\exp_{\gamma(t)}^{-1}(y) = \left(D\exp_{\gamma(t)}(v)\right)^{-1} = \mathrm{Id} + \mathcal{O}(\|v\|^2). \tag{22}$$

Next, we apply a Taylor expansion to the Hamiltonian vector field $X_H$ around $\gamma(t)$. Since $J_t = \exp_{\gamma(t)}^{-1}(\mathbf{X}_t)$ represents a small deviation, we have

$$X_H(\mathbf{X}_t) = X_H\big(\gamma(t)\big) + \nabla_{J_t} X_H\big(\gamma(t)\big) + \mathcal{O}(\|J_t\|^2). \tag{23}$$

By the assumption $X_H\big(\gamma(t)\big) = 0$, this simplifies to

$$X_H(\mathbf{X}_t) = \nabla_{J_t} X_H\big(\gamma(t)\big) + \mathcal{O}(\|J_t\|^2).$$

Thus, the stochastic differential $d\mathbf{X}_t$ is given by

$$d\mathbf{X}_t = \nabla_{J_t} X_H\big(\gamma(t)\big) \circ dB_t + \mathcal{O}(\|J_t\|^2) \circ dB_t.$$

Now, we apply the differential of the inverse exponential map to both sides. Using (22), we find

$$D\exp_{\gamma(t)}^{-1}\big(d\mathbf{X}_t\big) = \left[\mathrm{Id} + \mathcal{O}(\|J_t\|^2)\right] \left[\nabla_{J_t} X_H\big(\gamma(t)\big) \circ dB_t + \mathcal{O}(\|J_t\|^2) \circ dB_t\right]$$
$$= \nabla_{J_t} X_H\big(\gamma(t)\big) \circ dB_t + \mathcal{O}(\|J_t\|^3) \circ dB_t.$$

Finally, since $J_t = \mathcal{O}(|B_t|)$ under small-noise scaling, the remainder term $\mathcal{O}(\|J_t\|^3) \circ dB_t$ becomes negligible compared to $dB_t$ in the Stratonovich limit. Thus, we conclude the desired first-order approximation:

$$D\exp_{\gamma(t)}^{-1}\big(d\mathbf{X}_t\big) \approx \nabla_{J_t}\{\mathbf{m}, \mathrm{H}_\theta\}\big(\gamma(t)\big) \circ dB_t.$$

$\square$

**Lemma C.3.** *Let $(\mathcal{Q}, g)$ be a $d$-dimensional Riemannian manifold. Assume that the scalar curvature* $\mathrm{Scal}(x)$ *satisfies*

$$\sup_{x \in \mathcal{Q}} |\mathrm{Scal}(x)| \leq S \tag{24}$$

*for some constant $S \geq 0$. Then the operator sup-norm of the Riemann curvature tensor satisfies the estimate*

$$\|R\|_\infty \leq \frac{2S}{d(d-1)}. \tag{25}$$

*In particular, the curvature-induced deviation term $(B)$ in the stochastic Jacobi analysis can be uniformly bounded in terms of the scalar curvature bound $S$.*

*Proof.* Recall that for any point $x \in \mathcal{Q}$ and any orthonormal basis $\{e_i\}_{i=1}^d$ of $T_x\mathcal{Q}$, the scalar curvature is given by

$$\mathrm{Scal}(x) = \sum_{i<j} 2\, K(e_i, e_j), \tag{26}$$

where $K(e_i, e_j)$ denotes the sectional curvature of the 2-plane spanned by $e_i$ and $e_j$:

$$K(e_i, e_j) = \frac{g(R(e_i, e_j)e_j, e_i)}{\|e_i \wedge e_j\|^2} = g(R(e_i, e_j)e_j, e_i). \tag{27}$$

There are exactly $\binom{d}{2} = \frac{d(d-1)}{2}$ independent pairs $(i, j)$, and each sectional curvature contributes linearly to the scalar curvature. Therefore, taking absolute values and using the triangle inequality, we obtain

$$2 \max_{i<j} |K(e_i, e_j)| \cdot \binom{d}{2} \geq |\mathrm{Scal}(x)|, \tag{28}$$

which rearranges to

$$\max_{i<j} |K(e_i, e_j)| \geq \frac{|\mathrm{Scal}(x)|}{d(d-1)}.$$ (29)

Since by definition

$$\|R\|_\infty = \sup_{x \in \mathcal{Q}} \sup_{\|u\|=\|v\|=1} \|R(u,v)\|_g \quad \text{and} \quad \|R(u,v)\|_g \approx |K(u,v)|$$

up to constants depending on the wedge norm $\|u \wedge v\| = 1$ for orthonormal pairs, we obtain

$$\|R\|_\infty \leq 2 \sup_{x \in \mathcal{Q}} \max_{i<j} |K(e_i, e_j)| \leq \frac{2S}{d(d-1)},$$ (30)

where the factor 2 arises from symmetrization conventions in the definition of the Riemann tensor versus the sectional curvature. Thus, the curvature tensor's sup-norm is explicitly controlled by the scalar curvature bound $S$. ☐

**Lemma C.4.** *Øksendal and Øksendal [2003] Let $(B_t)_{t\in[0,T]}$ be a standard Brownian motion. Then, the mean-squared expectation of Stratonovich SDEs can be calculated as follows:*

$$\mathbb{E}\left\|\int_0^t Z_s \circ dB_s\right\|^2 = \int_0^t \mathbb{E}\|Z_s\|^2 \, ds + \frac{1}{4}\mathbb{E}\left\|\int_0^t \nabla Z_s \cdot Z_s \, ds\right\|^2$$

$$+ \mathbb{E}\left\langle \int_0^t Z_s \, dB_s, \int_0^t \nabla Z_s \cdot Z_s \, ds \right\rangle$$ (31)

# D  Proofs

This section serves to rigorously formalize all theoretical results that were informally stated in the main text. The goal is to provide complete proofs that fill in the technical gaps and support the conceptual developments discussed earlier.

## D.1  Proof of Proposition 2.2

**Proposition D.1** (Horizontal Hamiltonian Diffusion). *Let $\mathbf{U}_t \in \mathcal{O}(\mathcal{M})$ be the horizontal lift of the diffusion process $\mathbf{X}_t = \pi(\mathbf{U}_t)$, where $\pi : \mathcal{O}(\mathcal{M}) \to \mathcal{M}$ is the canonical projection and $\mathbf{m}$ denotes a local coordinate function on $\mathcal{M}$. The lifted process $\mathbf{U}_t$ evolves according to the Stratonovich SDE:*

$$d\mathbf{U}_t := G_\theta^{\mathrm{Hor}}(\mathbf{U}_t) \circ d\mathbf{B}_t, \quad \pi(\mathbf{U}_t) = \mathbf{X}_t, \quad \mathbf{U}_t \in \mathcal{O}(\mathcal{M}), \tag{32}$$

$$G_\theta^{\mathrm{Hor}}(\mathbf{U}_t) := (\{\mathbf{m}, H_\theta\}(\mathbf{X}_t)) \nabla_x - \left[ \left( \mathbf{I}_d \otimes (\{\mathbf{m}, H_\theta\}(\mathbf{X}_t))^\top \right) \cdot [\Gamma^{\mathcal{M}}]^\flat \mathrm{vec}(E) \right] \nabla_e, \tag{33}$$

*where $[\Gamma^{\mathcal{M}}]^\flat \in \mathrm{M}(2d \times d^2)$ is the index-lowered connection tensor (i.e., Christoffel symbol), and $\mathrm{vec}(E) \in \mathrm{M}(d^2, 1)$ is the vectorized local orthonormal frame. $\nabla_x$ and $\nabla_e$ denote the vectorized gradients with respect to the configuration point and the frame coordinates, respectively.*

*Proof.* We begin by deriving the horizontal lift of stochastic Hamiltonian dynamics in local coordinates. This involves expressing the dynamics $\{\mathbf{X}_t\}$ on the cotangent bundle $T^*\mathcal{Q}$ under a local trivialization of the frame bundle $\mathcal{O}\mathcal{M}$, equipped with a moving frame $E_t$. For explicit analytical and numerical handling, we represent the Stratonovich SDEs in terms of Euclidean coordinates via the horizontal lift operator. In this coordinate system, the horizontal lift of the Stratonovich-type stochastic Hamiltonian dynamics is written as:

$$\dot{q}_t^i = \left( \frac{\partial H_\theta}{\partial p_i} \right) (\mathbf{X}_t) - \sum_{j=1}^n \sum_{\alpha=1}^n \left( \sum_{k=1}^n \Gamma_{jk}^i(q_t) e_t^{\alpha,k} \right) \left( \frac{\partial H_\theta}{\partial p_j} \right) (\mathbf{X}_t) e_t^{\alpha,i} \circ d\mathbf{B}_t,$$

$$\dot{p}_t^i = -\left( \frac{\partial H_\theta}{\partial q^i} \right) (\mathbf{X}_t) - \sum_{j=1}^n \sum_{\alpha=1}^n \left( \sum_{k=1}^n \Gamma_{jk}^i(q_t) e_t^{\alpha,k} \right) \left( \frac{\partial H_\theta}{\partial q^j} \right) (\mathbf{X}_t) e_t^{\alpha,i} \circ d\mathbf{B}_t. \tag{34}$$

The above system describes the horizontal lift of Hamiltonian dynamics where the curvature of the base manifold $Q$ encoded by the Christoffel symbols $\Gamma_{jk}^i$ and the stochastic transport along local orthonormal frames $E_t$ jointly modulate the diffusion. The geometric structure is embedded via the lifted noise term on $T^*\mathcal{Q}$, ensuring that Brownian motion remains horizontal with respect to the Levi-Civita connection.

To enable stochastic calculus, we now transform the Stratonovich integrals in (34) into Itô form. This allows the introduction of correction terms due to the nonlinear dependence of the coefficients on the stochastic process. The position dynamics in Itô form become:

$$\dot{q}_t^i = \left[ \left( \frac{\partial H_\theta}{\partial p_i} \right) - \sum_{j=1}^n \sum_{\alpha=1}^n \left( \sum_{k=1}^n \Gamma_{jk}^i e_t^{\alpha,k} \right) \left( \frac{\partial H_\theta}{\partial p_j} \right) e_t^{\alpha,i} \right] d\mathbf{B}_t$$

$$+ \frac{1}{2} \sum_{j=1}^{2n} \left\{ \frac{\partial^2 H_\theta}{\partial p_i \partial p_j} - \sum_{r=1}^n \sum_{\alpha=1}^n \sum_{k=1}^n \left[ \frac{\partial \Gamma_{rk}^i}{\partial p_j} e_t^{\alpha,k} \left( \frac{\partial H_\theta}{\partial p_r} \right) e_t^{\alpha,i} + \Gamma_{rk}^i \frac{\partial E_t^{\alpha,k}}{\partial p_j} \left( \frac{\partial H_\theta}{\partial p_r} \right) e_t^{\alpha,i} \right. \right. \tag{35}$$

$$\left. \left. + \Gamma_{rk}^i e_t^{\alpha,k} \frac{\partial^2 H_\theta}{\partial p_j \partial p_r} E_t^{\alpha,i} + \Gamma_{rk}^i e_t^{\alpha,k} \left( \frac{\partial H_\theta}{\partial p_r} \right) \frac{\partial e_t^{\alpha,i}}{\partial p_j} \right] \right\} \cdot \mathrm{V}^j \, dt.$$

Similarly, the momentum equation is converted as:

$$\dot{p}_t^i = \left[ -\left(\frac{\partial H_\theta}{\partial q^i}\right) - \sum_{j=1}^{n}\sum_{\alpha=1}^{n}\left(\sum_{k=1}^{n}\Gamma_{jk}^i E_t^{\alpha,k}\right)\left(\frac{\partial H_\theta}{\partial q^j}\right) E_t^{\alpha,i}\right] d\mathbf{B}_t$$

$$+ \frac{1}{2}\sum_{j=1}^{2n}\left\{ -\frac{\partial^2 H_\theta}{\partial q^i \partial q^j} - \sum_{r=1}^{n}\sum_{\alpha=1}^{n}\sum_{k=1}^{n}\left[\frac{\partial \Gamma_{rk}^i}{\partial q^j} E_t^{\alpha,k}\left(\frac{\partial H_\theta}{\partial q_r}\right) E_t^{\alpha,i} + \Gamma_{rk}^i \frac{\partial E_t^{\alpha,k}}{\partial q^j}\left(\frac{\partial H_\theta}{\partial q_r}\right) E_t^{\alpha,i}\right.\right.$$

$$\left.\left. + \Gamma_{rk}^i E_t^{\alpha,k}\frac{\partial^2 H_\theta}{\partial q^j \partial q_r} E_t^{\alpha,i} + \Gamma_{rk}^i E_t^{\alpha,k}\left(\frac{\partial H_\theta}{\partial q_r}\right)\frac{\partial E_t^{\alpha,i}}{\partial q^j}\right]\right\} \cdot V^j \, dt. \tag{36}$$

Here, the additional drift induced by curvature and moving frames is absorbed into the auxiliary term $V^j$, defined as:

$$V^j(X) = \left(\mathbf{1}_{j\leq d}\cdot\frac{\partial H_\theta}{\partial p_j}(X) - \mathbf{1}_{j>d}\cdot\frac{\partial H_\theta}{\partial q^{j-d}}(X)\right) - \sum_{\alpha,r,k}^{n}\Gamma_{rk}^{j'}(q)\,E_t^{\alpha,k}\left(\frac{\partial H_\theta}{\partial \xi_r}(X)\right)E_t^{\alpha,j'}. \tag{37}$$

In Euclidean coordinates with Cartesian frames, all connection coefficients vanish ($\Gamma = 0$), and the orthonormal frame $E_t$ becomes static. As a result, the geometric correction term $V^j$ also disappears, recovering the standard stochastic Hamiltonian flow.

To summarize and simplify the geometric formulation, we now express the entire Hamiltonian diffusion in matrix notation. Let us define the following tensorial representations:

$$\{m, H_\theta\} \in \mathbb{R}^{1\times 2d}, \quad \mathrm{Id}\otimes\{m, H_\theta\}^\top \in \mathbb{R}^{d\times 2d}, \quad \Gamma^\flat \in \mathbb{R}^{2d\times d^2}, \quad \mathrm{vec}(E_t)\in\mathbb{R}^{d^2}.$$

Let $\mathbf{a} := \Gamma^\flat\cdot\mathrm{vec}(E_t)$ denote the geometric distortion vector. Then, the matrix form of the Hamiltonian diffusion reads:

$$\textbf{(Ito)}\quad d\mathbf{X}_t = \{m, H_\theta\}(\mathbf{X}_t)\,d\mathbf{B}_t + \frac{1}{2}\sum_{j=1}^{2d}\left(D^2\{m, H_\theta\}\cdot\mathbf{a}^j\right)\cdot\mathbf{a}^j\,dt, \tag{38}$$

where $\mathbf{a}^j$ denotes the $j$-th column of $\mathbf{a}$, and $D^2\{m, H_\theta\}\in\mathbb{R}^{2d\times 2d\times 2d}$ is a rank-3 tensor containing Hessian of Hamiltonian. This reformulation makes explicit the second-order geometry-aware correction arising from horizontal noise transport in local coordinates. With the form of Stratonovich's diffusion, one can recover the original definition used in Eq (5) as follows:

$$\textbf{(Stratonovich)}\quad d\mathbf{X}_t = \left[\{m, H_\theta\}(\mathbf{X}_t) - \left(\mathrm{Id}\otimes\{m, H_\theta\}^\top\right)\Gamma^\flat\,\mathrm{vec}(E_t)\right]\circ d\mathbf{B}_t. \tag{39}$$

The first part of the proof is complete by rewriting the above dynamics presented as Stratonovich SDEs.

As a next step, we aim to establish the theoretical validity of our geometric construction by verifying whether the proposed vector fields $G_\theta^{\mathrm{Hor}}$ indeed lie in the horizontal distribution of the orthonormal frame bundle $\mathcal{O}(\mathcal{M})$, where $\mathcal{M} := T^*\mathcal{Q}$ is the cotangent bundle equipped with a Sasaki-type metric.

We introduce canonical coordinates on $\mathcal{M}$ as

$$x^\alpha = (q^i, p_i), \quad \alpha\in\{1,\ldots,2d\}. \tag{40}$$

With block index conventions where $q$-indices are $i, j, k\in\{1,\ldots,d\}$ and fibre indices are $\bar{\imath} := d+i$, the Sasaki-type metric on $\mathcal{M}$ is given by

$$g^{\mathcal{M}} = g \oplus g^{-1}, \tag{41}$$

where $g$ is the base Riemannian metric on $\mathcal{Q}$. The Christoffel symbols of the Levi-Civita connection on $\mathcal{M}$, denoted $\Gamma^{\mathcal{M}}{}^{\alpha}_{\beta\gamma}$, have the following block structure (with $\partial_i := \partial/\partial q^i$):

$$\Gamma^{\mathcal{M}}{}^{\alpha}_{\beta\gamma} = \begin{cases} \Gamma_{jk}^i, & \alpha = i,\ \beta = j,\ \gamma = k, \\ \frac{1}{2}(\partial_j g^{ik} + \partial_k g^{ij}), & \alpha = i,\ \beta = j,\ \gamma = \bar{k}, \\ -\frac{1}{2}\partial_\ell g_{jk}, & \alpha = \bar{\ell},\ \beta = j,\ \gamma = k, \\ 0, & \text{otherwise.} \end{cases} \tag{42}$$

Let $U = (x, E) \in \mathcal{O}(\mathcal{M})$ be a point on the orthonormal frame bundle, where $E = (E^{\alpha}{}_a) \in \mathbb{R}^{2d \times 2d}$ is an orthonormal frame at $x$. We define the vectorized frame by

$$\text{vec}(E) \in \mathbb{R}^{(2d)^2}, \tag{43}$$

which stacks the columns of $E$. We also define the index-lowered connection tensor

$$[\Gamma^{\mathcal{M}}]^{\flat} \in \mathbb{R}^{(2d) \times (2d)^2} \tag{44}$$

via the transformation

$$([\Gamma^{\mathcal{M}}]^{\flat}\text{vec}(E))^{\alpha} := \Gamma^{\mathcal{M}\,\alpha}_{\beta\gamma} E^{\beta}{}_a. \tag{45}$$

Then for any $v \in \mathbb{R}^{2d}$, we have the key identity:

$$\text{mat}\left([I_{2d} \otimes v^{\top}][\Gamma^{\mathcal{M}}]^{\flat}\text{vec}(E)\right) = \Gamma^{\mathcal{M}}(v)E, \tag{46}$$

where $\Gamma^{\mathcal{M}}(v)^{\alpha}{}_{\beta} := \Gamma^{\mathcal{M}\,\alpha}_{\beta\gamma}v^{\gamma}$, and $\text{mat}(\cdot)$ reshapes a vector of length $(2d)^2$ into a $2d \times 2d$ matrix.

We now set

$$v := \{\mathbf{m}, \mathrm{H}_{\theta}\}(x) = (\partial_p \mathrm{H}_{\theta}, -\partial_q \mathrm{H}_{\theta}) \in \mathbb{R}^{2d}, \tag{47}$$

which defines the base vector field associated with the Hamiltonian dynamics. Then the lifted horizontal vector field on the frame bundle is given by

$$\mathrm{G}_{\theta}^{\text{Hor}} = v^{\alpha}\partial_{x^{\alpha}} - \left[I_{2d} \otimes v^{\top}\right][\Gamma^{\mathcal{M}}]^{\flat}\text{vec}(E) \cdot \partial_E. \tag{48}$$

To confirm horizontality, we define the temporal derivative of the frame:

$$\dot{E} := -[I \otimes v^{\top}][\Gamma^{\mathcal{M}}]^{\flat}\text{vec}(E), \tag{49}$$

which is equivalent to

$$\dot{E} = -\text{vec}(\Gamma^{\mathcal{M}}(v)E). \tag{50}$$

The Levi-Civita connection 1-form evaluated at a vector field $V$ is given by

$$\omega^a{}_b(V) = E^a{}_{\alpha}\left(\dot{E}^{\alpha}{}_b + \Gamma^{\mathcal{M}\,\alpha}_{\beta\gamma}v^{\gamma}E^{\beta}{}_b\right). \tag{51}$$

Substituting the expression for $\dot{E}^{\alpha}{}_b$, we obtain

$$\dot{E}^{\alpha}{}_b = -\Gamma^{\mathcal{M}\,\alpha}_{\beta\gamma}v^{\gamma}E^{\beta}{}_b, \tag{52}$$

which implies

$$\omega^a{}_b(\mathrm{G}_{\theta}^{\text{Hor}}) = E^a{}_{\alpha}\left(-\Gamma^{\mathcal{M}\,\alpha}_{\beta\gamma}v^{\gamma}E^{\beta}{}_b + \Gamma^{\mathcal{M}\,\alpha}_{\beta\gamma}v^{\gamma}E^{\beta}{}_b\right) = 0. \tag{53}$$

Therefore, the connection 1-form vanishes:

$$\omega(\mathrm{G}_{\theta}^{\text{Hor}}) = 0, \tag{54}$$

which confirms that the vector field lies in the horizontal distribution:

$$\mathrm{G}_{\theta}^{\text{Hor}} \in H\mathcal{O}(\mathcal{M}). \tag{55}$$

In conclusion, the proposed Stratonovich SDE

$$d\mathbf{U}_t = \mathrm{G}_{\theta}^{\text{Hor}}(\mathbf{U}_t) \circ d\mathbf{B}_t \tag{56}$$

is the *horizontal lift* of the Hamiltonian diffusion process on the cotangent bundle $T^*\mathcal{Q}$, ensuring geometric consistency with the underlying connection on $\mathcal{M}$.

$\square$

## D.2 Proof of Proposition 3.1

**Proposition D.2** (Time-uniform Generalization Bound of Hamiltonian Diffusion). *Let* $\mathbb{P}_t(\theta^\star) :=$ **Law**$(\gamma_t(\theta^\star))$ *be an associated probability measure of model trajectory, and assume that the proposed neural networks lie in Sobolev ball i.e.,* $\|\theta\|_{W^{2,s}} \leq R, \forall \theta \in \Theta$, *and first and second derivatives of Hamiltonian are Lipschitzian.*

$$\mathbb{P}\left[\sup_{t \in [0,T]} \sup_{\theta^\star \in \Theta} \mathcal{W}\left(\mathbb{P}_t(\theta^\star), \mathbb{P}_{t,data}\right) \leq \delta\right] \lesssim \exp\left(-\Omega \cdot \delta^{1/2}\|\mathbf{\Gamma}\|_\infty^{3/2} R^{1/2}(\log R)^{1/4}\right), \quad (57)$$

*While the first bound captures the uniform deviation of the learned trajectory distribution from the target data measure across time, we next provide a concentration result that controls the deviation between the empirical Wasserstein distance and its population expectation.*

$$\mathbb{P}\left[\sup_{t \in [0,T]} \sup_{\theta \in \Theta}\left|\mathcal{W}\left(\frac{1}{n}\sum_i^n \delta_{\gamma(\theta_i)}, \tilde{\mathbb{P}}_{data}\right) - \mathbb{E}_\theta \mathcal{W}\left(\gamma(\theta), \tilde{\mathbb{P}}\right)\right| \leq \delta\right]$$

$$\lesssim \exp\left(-\Omega \cdot \delta^{1/2}\|\mathbf{\Gamma}\|_\infty^{3/2} \frac{R^{1/2}(\log R)^{1/4}}{n^{1/4}}\right), \quad (58)$$

*where* $\mathcal{W} := \mathcal{W}^{2,2}_{T(T^*Q)}$ *stands for the squared Wasserstein distance on physical data space* $T(T^*Q)$, $\Omega := \Omega(\sigma, \lambda_{max}, L_H, L_{\nabla H}, d, s)$, $d > 2s$ *is a constant depending on metric tensor* $g$ *and the smoothness, Lipschitz constant of Hamiltonian.*

*Proof.* While the proposed stochastic system is semi-martingale, the chain rule with respect to Poisson bracket (*i.e.*, Eq.(2.8) Lázaro-Camí and Ortega [2008]) direct gives the following result:

$$\pi_\mathbf{q}^i \circ \pi(\mathbf{U}_t) - \pi_\mathbf{q}^i \circ \pi(\mathbf{U}_0) = \sum_{j=1}^d \int_0^t \{\pi_\mathbf{q}^i, H_\theta\}(\mathbf{X}_s) \circ d\mathbf{B}_s^j$$

$$= \sum_{j=1}^d \int_0^t \sum_{k=1}^d \left(\frac{\partial \pi_\mathbf{q}^i}{\partial q_k}\frac{\partial H_\theta^j}{\partial p_k} - \frac{\partial \pi_\mathbf{q}^i}{\partial p_k}\frac{\partial H_\theta^j}{\partial q_k}\right)(\mathbf{X}_s) \circ d\mathbf{B}_s^j \quad (59)$$

$$= \sum_{k,j=1}^d \int_0^t \left(\delta_i^k \frac{\partial H_\theta^j}{\partial p_k}\right)(\mathbf{X}_s) \circ d\mathbf{B}_s^j = \sum_{j=1}^d \int_0^t \frac{\partial H_\theta^j}{\partial p_i}(\mathbf{X}_s) \circ d\mathbf{B}_s^j,$$

where $\delta_a^b$ denotes the Kronecker delta. This reveals that the stochastic evolution of the velocity field depends explicitly on the derivatives with respect to momentum coordinates in Eq. (59), highlighting the necessity of incorporating additional physical information. For further discussion, we first give a Sasaki-type fiber metric (*i.e.*, norm) on $T\mathcal{M} = T(T^*Q)$. Then, the squared distance between $\gamma^1$ and $\gamma^2$ on tangent bundle $T\mathcal{M}$ can be naturally defined as follows:

$$d^2_{T(T^*Q)}(\gamma^1, \gamma^2) = \|(q^1, p^1, \dot{q}^1, \dot{p}^1) - (q^2, p^2, \dot{q}^2, \dot{p}^2)\|^2_{T(T^*Q)}$$

$$= \|q^1 - q^2\|^2_{g_\mathcal{Q}} + \|p^1 - p^2\|^2_{g_\mathcal{Q}^{-1}} + \|\dot{q}^1 - \dot{q}^2\|^2_{g_\mathcal{Q}} + \|\dot{p}^1 - \dot{p}^2\|^2_{g_\mathcal{Q}^{-1}} \quad (60)$$

where $g_\mathcal{Q}^{-1}(\alpha^\#, \beta^\#) = g_\mathcal{Q}(\alpha, \beta)$ is a dual metric on configuration space. Following by the definition of the norm on tangent bundle of total manifold $T(T^*Q)$, the discrepancy between model trajectory $\gamma(\theta$ and data trajectory $\tilde{\gamma}$ can be calculated as follows:

$$d^2_{T(T^*Q)}(\gamma(\theta), \tilde{\gamma}) = \|(\mathbf{q}_t(\theta), \mathbf{p}_t(\theta), \dot{\mathbf{q}}_t(\theta), \dot{\mathbf{p}}_t(\theta)) - (\tilde{\mathbf{q}}_t, \tilde{\mathbf{p}}_t, \dot{\tilde{\mathbf{q}}}_t, \dot{\tilde{\mathbf{p}}}_t)\|^2_{T(T^*\mathcal{Q})}$$

$$= \|(\mathbf{0}_d, \mathbf{p}_t(\theta), \dot{\mathbf{q}}_t(\theta), \mathbf{0}_d) - (\mathbf{0}_d, \tilde{\mathbf{p}}_t, \dot{\tilde{\mathbf{q}}}_t, \mathbf{0}_d)\|^2_{T(T^*\mathcal{Q})}$$

$$\leq \|(\mathbf{0}_d, \mathbf{p}_t(\theta), \dot{\mathbf{q}}_t(\theta), \mathbf{0}_d) - (\mathbf{0}_d, \mathbf{p}_t(\theta_0), \mathbf{p}_t(\theta_0), \mathbf{0}_d)\|^2_{T(T^*\mathcal{Q})} \quad (61)$$

$$+ \|(\mathbf{0}_d, \mathbf{p}_t(\theta_0), \dot{\mathbf{q}}_t(\theta_0), \mathbf{0}_d) - (\mathbf{0}_d, \tilde{\mathbf{p}}_t, \dot{\tilde{\mathbf{q}}}_t, \mathbf{0}_d)\|^2_{T(T^*\mathcal{Q})}$$

Let assume that the test neural network $\theta$ satisfies perfectly matches particle trajectories almost surely *i.e.*, $\pi_\mathbf{q} \circ \pi(\mathbf{U}_t) = q_t(\theta) = \tilde{q}_t$, and assume both mapping $\nabla_\mathbf{p}H(\mathbf{q}, \cdot; \theta)$ and $\nabla_\mathbf{q}H(\mathbf{p}, \cdot; \theta)$ is

an injective mapping for each fixed $\mathbf{q}$ and $\mathbf{p}$. Consider another neural network $\theta_0$ both matches both particle trajectory and their corresponding momentum behavior *e.g.*, $\theta_0 := \arg\min_\theta \mathcal{L}(\theta)$.

If there exists a inverse Lipschitz constant $L_H^{-1}$ of second mapping, then the assumptions leads to the second and third inequality in Eq. (62):

$$
\begin{aligned}
\mathcal{W}^{2,2}_{T(T^*Q)} & (\mathbf{Law}(\gamma(\theta), \mathbf{Law}(\tilde{\gamma})) \\
& \leq \mathbb{E}d^2_{T(T^*Q)}(\gamma(\theta), \tilde{\gamma}) \\
& \leq \mathbb{E}\|\dot{q}_t(\theta) - \dot{q}_t(\theta_0)\|^2_{g_Q} + \mathbb{E}\|p(\theta) - p(\theta_0)\|^2_{g_Q^{-1}} \\
& \leq (1 + L^{-2}_{H_\theta})\mathbb{E}\|\dot{q}_t(\theta) - \dot{q}_t(\theta_0)\|^2_{g_Q}.
\end{aligned}
\tag{62}
$$

Given the fact that velocity field lies in the tangent of configuration space $\dot{q}_t \in T\mathcal{Q} \cong \mathbb{R}^d$, taking a supremum with respect neural networks in both side of inequality in Eq. (62) gives

$$
\begin{aligned}
\sup_{\theta \in \Theta} \mathcal{W}\left(\mathbf{Law}(\gamma(\theta)), \mathbf{Law}(\tilde{\gamma})\right) & \leq (1 + L^{-2}_{H_\theta}) \sup_{\theta \in \Theta} \mathbb{E}\left[\|\dot{q}_t(\theta) - \dot{q}_t(\theta_0)\|^2_{g_Q}\right] \\
& \leq (1 + L^{-2}_{H_\theta})\lambda^2_{\max}(g_Q) \sup_{\theta \in \Theta} \mathbb{E}\left[\|\dot{q}_t(\theta) - \dot{q}_t(\theta_0)\|^2_E\right].
\end{aligned}
\tag{63}
$$

For readability, we simplify the notation as $\mathcal{W} := \mathcal{W}^{2,2}_{T(T^*Q)}$. Note that the expectation in this context is taken from $\mu_{q_t} \in \mathcal{P}(T_{q_t}Q)$ for each $t \in [0,T]$. In first inequality, we normalize the Riemannian inner product with the Euclidean correspondence by using the property: $\|v\|^2_{g_Q} \leq \lambda^2_{\max}(g_Q)\|v\|^2_E$ for any vectors $v \in T_{p_t}\mathcal{Q}$. Next, our goal is to obtain the following type of decomposition

$$
\sup_{\theta \in \Theta} \mathbb{E}\left[\|\dot{q}_t(\theta) - \dot{q}_t(\theta_0)\|^2_E\right] \leq \mathfrak{f} \cdot \sup_{\theta^{(1)}_t \in \Theta} \|\theta^{(1)}_t - \theta^{(2)}_t\|^2_\Theta.
\tag{64}
$$

where the time-dependent constant $\mathfrak{f} := \mathfrak{f}(t, \mathbf{\Gamma}, \partial\mathbf{\Gamma}, \partial^I H_\theta)$ depends on the connection form $\mathbf{\Gamma}$ and their derivative $\partial\mathbf{\Gamma}$, and the Lipschitz constants of higher-order derivatives $\mathbf{Lip}(\partial^I H_\theta), \forall I \leq 2$. To this end, we first define four auxiliary processes as follows:

$$
\begin{aligned}
\mathrm{D}_t(\theta) & := \partial_{p_i} H_\theta, & \mathrm{A}_{j,\alpha}(q_t) & = \sum_{k=1}^d \Gamma^k_{j\alpha}(q_t)E^{\alpha,k}_t, \\
\mathrm{B}^i_{j,\alpha}(\theta) & = (\partial_{p_i} H_\theta)E^{\alpha,i}_t, & \mathrm{Z}_t & = \sum_{j,\alpha} \mathrm{A}_{j,\alpha}(q_t)\delta\mathrm{B}_{j,\alpha}(t),
\end{aligned}
\tag{65}
$$

where the mean-squared norm of processes $\|\mathrm{A}\|^2_E, \|\delta\mathrm{B}\|^2_E, \|\mathrm{Z}\|^2_E, \|\mathrm{D}\|^2_E, \|\nabla\mathrm{Z}\|^2_E$ is bounded above with some constants $C_A, C_B, C_Z, C_D, C_{\nabla Z}$. Having the definition in hands, the proposed velocity vector fields for arbitrary network $\theta$ can be simplified with the following form:

$$
\dot{q}_t(\theta) = \int_0^t \mathrm{D}_s(\theta) - \sum_{j,\alpha} \mathrm{A}_{j,\alpha}(q_s)\mathrm{B}_{j,\alpha}(\theta) \circ d\mathbf{B}_t.
\tag{66}
$$

With the definition $\delta\dot{q}_t := \dot{q}_t(\theta) - \dot{q}_t(\theta_0)$ for deviation between two velocity vector fields, direct calculation leads to have norm-squared expectation as follows:

$$
\begin{aligned}
\mathbb{E}\left[\|\delta\dot{q}_s\|^2_E\right] & = \mathbb{E}\left[\left\|\int_0^t \delta\mathrm{D}_s - \sum_{j,\alpha} \mathrm{A}_{j,\alpha}(q_s)\,\delta\mathrm{B}_{j,\alpha}(s) \circ dB_s\right\|^2_E\right] \\
& \leq 2\mathbb{E}\left[\left\|\int_0^t \delta D_s ds\right\|^2\right] + 2\mathbb{E}\left[\left\|\int_0^t \underbrace{\sum_{j,\alpha} \mathrm{A}_{j,\alpha}(q_s)\,\delta\mathrm{B}_{j,\alpha}(s)}_{:=Z_s} \circ dB_s\right\|^2_E\right]
\end{aligned}
\tag{67}
$$

Following by the conversion of Stratonovich SDE into Ito's SDE in Lemma C.4, the second term of right-hand in last inequality can be upper-bounded with the following form:

$$
\mathbb{E}\left[\left\|\int_0^t Z_s \circ dB_s\right\|^2\right] \leq C^2_Z\left(1 + \frac{1}{4}t^2 C^2_{\nabla Z} + t^{\frac{3}{2}}C_{\nabla Z}\right)
\tag{68}
$$

This bound reflects the second-moment structure of the Stratonovich integral, where the dominant contribution arises from the squared noise norm $C_Z^2$, and the correction terms involve both the norm and gradient of the stochastic vector field $Z_t$.s

$$\|\mathrm{A}\|_E := C_A \le \sqrt{d}\lambda_{\max}^{-1/2}\|\mathbf{\Gamma}\|_\infty, \quad \|\mathbf{\Gamma}\|_\infty = \sup_{q \in \mathcal{Q}} \sup_{j,\alpha,k} |\Gamma_{j\alpha}^k(q)| \tag{69}$$

$$\|\delta\mathrm{B}\|_E := C_B \le L_\mathrm{H}\lambda_{\max}^{-1/2}\|\theta - \theta_0\|, \tag{70}$$

$$\|Z\|_E := C_Z \le d^2 C_A C_B = d^{5/2}\lambda_{\max}^{-1}\|\mathbf{\Gamma}\|_\infty L_\mathrm{H}\|\theta - \theta_0\|. \tag{71}$$

This shows that the magnitude of the noise vector $Z_t$ grows quadratically with the model distance $\|\theta - \theta_0\|$, and is modulated by the geometric curvature $\|\Gamma\|_\infty$ and the Hamiltonian smoothness $L_H$.

$$\nabla Z_s = \sum_{j,\alpha} \nabla \mathrm{A}_{j,\alpha}(q_s) \cdot \delta B_{j,\alpha}(s) + \mathrm{A}_{j,\alpha}(q_s) \cdot \nabla \delta B_{j,\alpha}(s). \tag{72}$$

This decomposition separates the gradient of the stochastic vector field $Z_t$ into two terms: one involving the spatial derivative of the geometry-aware coefficient $A_{j,\alpha}$, and the other involving the gradient of the perturbation $\delta B_{j,\alpha}$, both of which are influenced by the manifold structure and the Hamiltonian model.

$$\|\nabla \mathrm{A}(q_s)\|_E := C_{\nabla A} \le \sqrt{d}\|\partial\mathbf{\Gamma}\|_\infty, \quad \|\mathbf{\Gamma}\|_\infty = \sup_{q \in \mathcal{Q}} \sup_{j,\alpha,k} |\partial\Gamma_{j\alpha}^k(q)|^2 \tag{73}$$

$$\|\nabla\delta\mathrm{B}\|_E := C_{\nabla B} \le (L_\mathrm{H} + L_{\nabla H}\|\nabla_q E_s^{\alpha,i}\|_E^2)\|\theta - \theta_0\|, \tag{74}$$

Since the orthonormal frame is locally updated by parallel transport, one can obtain

$$\partial_{q_j} E_s^{\alpha,i}(q_s) = -\sum_i \Gamma_{ji}^k(q_s)E_s^{\alpha,i}, \quad \|\nabla_q E_s^{\alpha,i}\|_E \le d\lambda_{\max}^{-1/2}\|\mathbf{\Gamma}\|_\infty \tag{75}$$

Therefore, the spatial variation of each frame component $E_t^{\alpha,i}$ is entirely determined by the Christoffel symbol and remains uniformly bounded under smooth parallel transport.s

$$\begin{aligned}\|\nabla Z_s\|_E := C_{\nabla Z} &\le d^2(C_{\nabla A}C_B + C_A C_{\nabla B}) \\ &\le d^{5/2}\left[\|\partial\mathbf{\Gamma}_\infty\|L_H + \|\mathbf{\Gamma}\|_\infty(L_{\nabla H} + L_H\|\mathbf{\Gamma}\|_\infty)\right]\|\theta - \theta_0\|.\end{aligned} \tag{76}$$

Hence, the total spatial gradient $\nabla Z_t$ scales linearly with the parameter deviation $\|\theta - \theta_0\|$, and is tightly controlled by the geometry through $\|\Gamma\|_\infty, \|\partial\Gamma\|_\infty$ and the Lipschitz constants $L_H, L_{\nabla H}$.

$$\mathbb{E}\left[\left\|\int_0^t \delta\dot{q}_s ds\right\|_E^2\right] \le \mathfrak{f}(t,\mathbf{\Gamma},\partial\mathbf{\Gamma}) \cdot \mathbb{E}\left[\|\delta\theta\|_E^2\right] \tag{77}$$

For $L = L_H \vee L_{\nabla H}$, $\mathfrak{f}$ is defined as follows:

$$\mathfrak{f}(t,\mathbf{\Gamma},\partial\mathbf{\Gamma}) := L \vee d^5\lambda_{\max}^{-1}L^2\|\mathbf{\Gamma}\|_\infty^2\left(1 + t^3 d^5 L^2\left[\|\partial\mathbf{\Gamma}\|_\infty + \|\mathbf{\Gamma}\|_\infty(1 + \|\mathbf{\Gamma}\|_\infty)\right]^2\right). \tag{78}$$

Now, our goal is to simplify the inequality, making $\mathfrak{f}$ is related to the curvature skewness. Using Lemma C.1, the curvature tensor provides a lower bound on $\|\partial\Gamma\|_\infty$, which allows us to eliminate the explicit derivative dependence and reparameterize $R_1$ in terms of $\|\mathcal{R}\|_\infty$ and $\|\Gamma\|_\infty$.

$$\|\mathcal{R}\|_\infty \le 2\|\partial\Gamma\|_\infty + 2\|\Gamma\|_\infty^2 \quad \Rightarrow \quad \|\partial\Gamma\|_\infty \ge \tfrac{1}{2}\|\mathcal{R}\|_\infty - \|\Gamma\|_\infty^2 \tag{79}$$

This inequality gives a curvature-dependent upper bound on the Riemann tensor norm in terms of the supremum of the partial derivatives and $\|\Gamma\|_\infty$, which allows us to replace $\|\mathcal{R}\|_\infty$ by $\|\Gamma\|_\infty$ in subsequent expressions. By combining the previous inequality with the triangle inequality, we obtain a uniform bound on $\|\partial\Gamma\|_\infty + \|\Gamma\|_\infty(1 + \|\Gamma\|_\infty)$ that is linear in $\|\mathcal{R}\|_\infty$ and $\|\Gamma\|_\infty$, facilitating simplification of higher-order terms.

$$\|\partial\Gamma\|_\infty + \|\Gamma\|_\infty(1 + \|\Gamma\|_\infty) \le \tfrac{1}{2}\|\mathcal{R}\|_\infty - \|\Gamma\|_\infty^2 + \|\Gamma\|_\infty(1 + \|\Gamma\|_\infty) = \tfrac{1}{2}\|\mathcal{R}\|_\infty + \|\Gamma\|_\infty \tag{80}$$

Thus, the curvature dependence can be simplified to a function of $\|R\|_\infty$ and $\|\Gamma\|_\infty$ only.

$$(\|\partial\Gamma\|_\infty + \|\Gamma\|_\infty(1 + \|\Gamma\|_\infty))^2 \le 2\left(\tfrac{1}{2}\|\mathcal{R}\|_\infty\right)^2 + 2\|\Gamma\|_\infty^2 = \tfrac{1}{2}\|\mathcal{R}\|_\infty^2 + 2\|\Gamma\|_\infty^2 \tag{81}$$

Squaring both sides, we derive a bound for the squared norm $(\|\partial\Gamma\|_\infty + \|\Gamma\|_\infty(1 + \|\Gamma\|_\infty))^2$, which ensures that second-order curvature contributions can be expressed as a function of $\|\mathcal{R}\|_\infty$ and $\|\Gamma\|_\infty^2$ alone.

$$\mathfrak{f}(t, \mathbf{\Gamma}, \mathcal{R}) := L \vee d^5 \lambda_{\max}^{-1} L^2 \left(1 + t^3 d^5 L^2\right) \|\mathbf{\Gamma}\|_\infty^2 \left(\frac{1}{2}\|\mathcal{R}\|_\infty + 2\|\Gamma\|_\infty\right) \propto \|\mathbf{\Gamma}\|_\infty^3. \quad (82)$$

As a result, the original function $\mathfrak{f}(t, \Gamma, \mathcal{R})$ can now be written in terms of $\|\Gamma\|_\infty$ only, up to a multiplicative constant, removing explicit curvature dependence from the generalization bound.

We now consider the set of neural networks $\theta_0$ constrained within a metric ball defined by a Sobolev-type functional distance. Let $\Theta$ denote the set of such neural networks whose Sobolev norm and supremum norm are simultaneously bounded by a constant $R > 0$. We then define the associated function class with respect to the $L^2$-norm, and consider a probability space $(\Theta, \Sigma_\mu, \tilde{\mathbb{P}}_\mu)$ supported on $\Theta$.

$$\Theta := \{\theta \in W^{s,2} \cap L^\infty(\mu_q); \quad \|\theta\|_{W^{s,2}(\mu_q)} \leq r, \quad \|\theta\|_\infty \leq R\}, \quad (83)$$

$$\mathcal{F}_\theta^r = \left\{F(\theta) := \|\theta_0 - \theta\|_{L^2(\mu_q)}^2; \quad \theta, \theta_0 \in \Theta\right\}. \quad (84)$$

For the random variable $\theta(\omega) \in \Sigma_\mu$, let us define auxiliary processes as

$$\mathfrak{Z} = \sup_{t \in [0,T]} \sup_{F \in \mathcal{F}_\Theta^r} F(\theta_t), \quad \mathfrak{X} = \sup_{t \in [0,T]} \sup_{F \in \mathcal{F}_\Theta^r} \left|\frac{1}{n}\sum_i^n F(\theta_t^i) - \mathbb{E}F(\theta_t)\right|. \quad (85)$$

For the metric $d_\mathcal{F}((t, F), (s, G)) = \|F(\theta_t) - G(\theta_s)\|_{L^2(\mu_q)}$ where $s \leq t \in [0, T]$, the $\varepsilon$-covering number on product space $[0, T] \times \mathcal{F}_\theta^r$ can be interpreted as a product of two sub-coverings in separate spaces:

$$N(\epsilon, [0, T] \times \mathcal{F}_\Theta^r, d_\mathcal{F}) \leq N(\varepsilon_1, \mathcal{F}_\Theta^r, \|\cdot\|_{L_2}) \cdot N(\varepsilon_2, [0, T], |\cdot|) \quad (86)$$

This shows that the metric entropy can be decomposed as summation of two sub-terms:

$$\log N(\epsilon, [0, T] \times \mathcal{F}_\Theta^r, d_\mathcal{F}) \leq \log N(\varepsilon_1, \mathcal{F}_\Theta^r, \|\cdot\|_{L_2}) + \log N(\varepsilon_2, [0, T], |\cdot|), \quad (87)$$

where the algebraic constraint on $\varepsilon_1$ and $\varepsilon_2$ is given as

$$\varepsilon_1^2 + \varepsilon_2^2 = (\varepsilon\beta_\varepsilon)^2 + (\varepsilon\sqrt{1 - \beta_\varepsilon})^2 = \varepsilon, \quad \beta_\varepsilon \in (0, 1). \quad (88)$$

As a next step, we derive the upper bound of expectation for the variable $\mathfrak{Z}$, assuming that $F(\theta_t)$ has controlled by Gaussian-like long-tail property, Proposition 1.2.1 Talagrand [2005]). Specifically, we apply Dudley's entropy integral bound to have the following result:

$$\begin{aligned}
\tilde{\mathbb{E}}_\theta[\mathfrak{Z}] &:= \tilde{\mathbb{E}}_\theta\left[\sup_{t \in [0,T]} \sup_{F \in \mathcal{F}_\Theta^r} F(\theta)\right] \\
&\lesssim \limsup_{r \to \infty} \int_0^{\mathrm{Diam}(\Theta)} \sqrt{\log \mathcal{N}(\varepsilon, [0, T] \times \mathcal{F}_\theta^r, d_\mathcal{F})} d\varepsilon \\
&\lesssim \limsup_{r \to \infty} \int_0^{\mathrm{Diam}(\Theta)} \sqrt{\log \mathcal{N}(\varepsilon_1, \mathcal{F}_\theta^r, \|\cdot\|_{L^2}) + \log N(\varepsilon_2, [0, T], |\cdot|)} d\varepsilon \\
&\lesssim \limsup_{r \to \infty} \int_0^{\mathrm{Diam}(\Theta)} \sqrt{\log \mathcal{N}(\varepsilon_1, \Theta, \|\cdot\|_{L^2}) + \log N(\varepsilon_2, [0, T], |\cdot|)} d\varepsilon \\
&\leq \limsup_{r \to \infty} \int_0^{\mathrm{Diam}(\Theta)} \sqrt{\log \mathcal{N}(\varepsilon\beta_\epsilon, B_{W^{s,2}}, \|\cdot\|_{W^{s,2}}) + \log N(\varepsilon\sqrt{1 - \beta_\varepsilon}, [0, T], |\cdot|)} d\varepsilon \\
&\lesssim \int_0^r \sqrt{\left(\frac{r}{\varepsilon\beta_\varepsilon}\right)^{\frac{d}{s}} + \log\left(\frac{T}{\varepsilon\sqrt{1 - \beta_\varepsilon}} + 1\right)} d\varepsilon, \quad d > s,
\end{aligned} \quad (89)$$

where the expectation $\tilde{\mathbb{E}}_\theta$ is taken with the probability measure $\tilde{\mathbb{P}}_\mu$. The third inequality naturally follows from the embedding $W^{s,2} \hookrightarrow L^2$ and the fact that $\varphi(a) = \|\theta - a\|_{L^2}^2$ is 1-Lipschitz for some $a \in B_{W^{s,2}}(r)$ where we consider the function composition $\mathcal{F}_\theta = \varphi \circ B_{W^{s,2}}(r)$. In the fourth

inequality, we follow the entropy number in metric ball with radius $r$ in Sobolev space $W^{s,2}$ Wellner et al. [2013], and taking supremum under the constraint $\|\cdot\|_{W^{2,s}} \leq R$.

Since the final expression is non-integrable in general case, we only provide their approximation bound with Taylor expansion in the case when the first term in square root nominates the other term.

$$\tilde{\mathbb{E}}_\theta[\mathfrak{Z}] \lesssim \frac{(2r)^{d/2s}R^{1-d/2s}}{1-d/2s} + \frac{1}{2^{1+d/2s}r^{d/2s}}\frac{R^{d/2s+1}}{d/2s+1}\left[\log(R) - \frac{1}{d/2s+1}\right] + \mathcal{O}(R^{-2}), \quad (90)$$

where $d > 2s$, and we simply set the variable $\beta_\varepsilon = 0.5$. After optimizing with respect to the radius $r$ the right-hand side, we finally have the time-uniform upper-bound of empirical estimates.

$$\tilde{\mathbb{E}}_\theta[\mathfrak{Z}] \sim \mathcal{O}(R\sqrt{\log R}). \quad (91)$$

Let us now turn our attention to the empirical concentration behavior of the Wasserstein distance, initiating our analysis with the standard symmetrization lemma.

$$\tilde{\mathbb{E}}_\theta[\mathfrak{X}] \leq 2\mathfrak{R}_n := 2\mathbb{E}_{\sigma_{rad}}\left[\sup_{t\in[0,T]}\sup_{F\in\mathcal{F}_\Theta^r}\frac{1}{n}\sum_i^n a_i F(\theta_t^i)\right]$$

$$\leq \limsup_{r\to\infty}\frac{12}{\sqrt{n}}\int_0^{\text{Diam}(\Theta)}\sqrt{\log\mathcal{N}(\varepsilon,[0,T]\times\mathcal{F}_\theta^r,d_{\mathcal{F}}^n)}d\varepsilon \quad (92)$$

$$\leq \limsup_{r\to\infty}\frac{12}{\sqrt{n}}\int_0^{\text{Diam}(\Theta)}\sqrt{\log\mathcal{N}(\varepsilon_1,\mathcal{F}_\theta^r,\|\cdot\|_{L^2}^n) + \log N(\varepsilon_2,[0,T],|\cdot|)}d\varepsilon$$

where the empirical version of metrics can be rewritten as following form:

$$\|F\|_{L^2}^n := \frac{1}{n}\sum_{i=1}^n\|F(\theta_t^i)\|_{L^2}, \quad d_{\mathcal{F}}^n((t,F),(s,G)) = \frac{1}{n}\sum_{i=1}^n|F(\theta_t^i) - G(\theta_t^i)|. \quad (93)$$

Here, $a_i \sim \mathbf{U}[\{\pm 1\}]$ and $\mathfrak{R}_n$ are denote both Rademacher variables and their corresponding empirical Rademacher complexity. As with the similar calculation conducted in Eq. (89), the expectation of $\mathfrak{X}$ admits an upper bound that involves both the function class radius $R$ and sample complexity $n$.

$$\tilde{\mathbb{E}}_\theta(\mathfrak{X}) \sim \mathcal{O}\left(R\sqrt{\frac{\log R}{n}}\right). \quad (94)$$

The result was obtained by using the identical metric entropy calculated in Eq (89) to derive the upper bound. Note that the additional assumption of Gaussian-like property is not considered here as opposite to first inequality in Eq. (89). Combining the result in Eq. (63), Eq. (64) and the definition of auxiliary processes in Eq. (85), we obtain two inequalities

$$(\mathbf{R1}) := \sup_{t\in[0,T]}\sup_{\theta\in\Theta}\mathcal{W}\left(\mathbf{Law}(\gamma(\theta)), \tilde{\mathbb{P}}_{\text{data}}\right) \leq \mathfrak{A}\cdot\mathfrak{Z}, \quad (95)$$

$$(\mathbf{R2}) := \sup_{t\in[0,T]}\sup_{\theta\in\Theta}\left|\mathcal{W}\left(\frac{1}{n}\sum_i^n\delta_{\gamma(\theta_i)}, \tilde{\mathbb{P}}_{\text{data}}\right) - \mathbb{E}_\theta\mathcal{W}\left(\gamma(\theta), \tilde{\mathbb{P}}\right)\right| \leq \mathfrak{A}\cdot\mathfrak{X}, \quad (96)$$

$$\mathfrak{A} := (1 + L_{\mathrm{H}_\theta}^{-2})\cdot\lambda_{\max}^2\cdot\mathfrak{f}. \quad (97)$$

Next, we show the exponential probability inequality associated with $(L^2(\mu_q), \Sigma_\mu, \tilde{\mathbb{P}}_\mu)$ by introducing the classical result from Theorem 3 Massart [2000]. Here, the probability space is considered as one specific choice of generic metric space.

$$\tilde{\mathbb{P}}_\mu\left[\begin{bmatrix}\mathfrak{Z}\\\mathfrak{X}\end{bmatrix} \leq (1+\epsilon)\begin{bmatrix}\tilde{\mathbb{E}}_\theta[\mathfrak{Z}]\\\tilde{\mathbb{E}}_\theta[\mathfrak{X}]\end{bmatrix} - \sigma\sqrt{\frac{54}{5}x} - \left(\frac{5}{2} + \frac{216}{5}\varepsilon^{-1}\right)Rx\right] \leq e^{-x}, \quad (98)$$

where $\sigma^2 = \sup_{F\in\mathcal{F}_\theta^r}\text{Var}(F(\theta))$ is a maximal variance of function class, and two positive constants $\varepsilon, x > 0$ are arbitrary. Collecting the result obtained from Eq. (89), (95), (96), (98), we have

$$\tilde{\mathbb{P}}_\mu\left[\frac{1}{\mathfrak{A}}\begin{bmatrix}(\mathbf{R1})\\(\mathbf{R2})\end{bmatrix} \leq \begin{bmatrix}\tilde{\mathbb{E}}_\theta[\mathfrak{Z}]\\\tilde{\mathbb{E}}_\theta[\mathfrak{X}]\end{bmatrix}(1+\epsilon) - \sigma\sqrt{\frac{54}{5}x} - \left(\frac{5}{2} + \frac{216}{5\varepsilon}\right)Rx\right] \leq e^{-x} \quad (99)$$

Our goal is to find the optimal constant $D > 0$ such that the quadratic expression in right-hand side serves as an upper bound for the left-hand side polynomial of linear order in Eq. (100).

$$A - B\sqrt{x} - Cx \leq \mathfrak{A}Dx^2, \tag{100}$$

Here, the constants $A$, $B$, and $C$ capture the contributions from the empirical moments, noise level, and smoothness complexity, and are given by:

$$A = \alpha(1 + \epsilon) := \left[\frac{\tilde{\mathbb{E}}_\theta[\mathfrak{Z}]}{\tilde{\mathbb{E}}_\theta[\mathfrak{X}]}\right](1 + \epsilon), \quad B = \sigma(54/5)^{\frac{1}{2}}, \quad C = \frac{5}{2} + \frac{216}{5\varepsilon}. \tag{101}$$

By minimizing the right-hand side with respect to the free variable $x$, we obtain the optimal constant $D$ that balances the quadratic and linear terms:

$$D = \frac{CE^2 + BE - A}{\mathfrak{A}E^4}, \quad E = \frac{-3B + \sqrt{9B^2 + 32AC}}{4C}. \tag{102}$$

Substituting the identity $Dx^2 = \delta$ into the exponential tail inequality, we arrive at the following probabilistic bound for $(\mathbf{R1})$:

$$\tilde{\mathbb{P}}_\mu[(\mathbf{R1}) \leq \delta] \leq \exp\left(-\sqrt{\frac{\delta}{D}}\right). \tag{103}$$

Observing that $\sqrt{9B^2 + 32AC} \geq \sqrt{32AC}$ and setting $\epsilon = 1$, we obtain the following upper bound for $D(\alpha, \mathfrak{A})$ by simplifying the denominator expression:

$$D(\alpha, \mathfrak{A}) \leq \frac{1}{\mathfrak{A}}\left[-\frac{C^2}{\alpha} + \frac{BC^{3/2}}{\alpha^{3/2}}\right] = \frac{1}{\mathfrak{A}}\left[\frac{208849}{100\alpha} + \frac{3\sigma\sqrt{30} \cdot 457^{3/2}}{10^{3/2} \cdot 5\alpha^{3/2}}\right]. \tag{104}$$

By substituting the upper bound of $D(\alpha, \mathfrak{A})$ obtained in Eq. (104) into the general exponential inequality in Eq. (103), we derive the following explicit bound on the probability of the event $(\mathbf{R1})$, which reflects the asymptotic decay behavior in terms of $R$ and the structural parameters:

$$\tilde{\mathbb{P}}_\mu[(\mathbf{R1}) \leq \delta]$$
$$\lesssim \exp\left(-\sqrt{\delta\mathfrak{A}}\left[10R^{3/4}(\log R)^{3/8}\sqrt{\frac{1}{208849R^{1/2}(\log R)^{1/4} - 18536.41\sqrt{30}\sigma}}\right]\right). \tag{105}$$

We now further simplify the expression by isolating the leading-order terms and observing that the dominant contribution arises from the linear dependence on $R^{1/2}(\log R)^{1/4}$ in the denominator. This leads to a more interpretable asymptotic bound expressed in terms of the supremum norm of the function class.

$$\tilde{\mathbb{P}}_\mu[(\mathbf{R1}) \leq \delta] \lesssim \exp\left(-\Omega \cdot \delta^{1/2}\|\mathbf{\Gamma}\|_\infty^{3/2}R^{1/2}(\log R)^{1/4}\right). \tag{106}$$

Similarly, we consider the case where the effective complexity $\alpha$ scales with the number of samples $n$ as $\alpha = R\sqrt{\log R/n}$. This reflects a regime where the resolution increases with sample size, leading to the following generalization bound:

$$\tilde{\mathbb{P}}_\mu[(\mathbf{R2}) \leq \delta]$$
$$\lesssim \exp\left(-\sqrt{\delta\mathfrak{A}}\left[\frac{10R^{3/4}(\log R)^{3/8}}{n^{1/4}}\sqrt{\frac{1}{208849\sqrt{R}(\log R)^{1/4} - 18536.41\sqrt{30}n^{1/4}\sigma}}\right]\right). \tag{107}$$

As before, we simplify the expression by extracting the leading dependence on $R$, $\log R$, and $n$ to arrive at an asymptotic bound that reveals the effect of sample size scaling on the generalization rate:

$$\tilde{\mathbb{P}}_\mu[(\mathbf{R2}) \leq \delta] \lesssim \exp\left(-\Omega \cdot \delta^{1/2}\|\mathbf{\Gamma}\|_\infty^{3/2}\frac{R^{1/2}(\log R)^{1/4}}{n^{1/4}}\right), \tag{108}$$

where $\Omega := \Omega(\sigma, \lambda_{\max}, L_H, L_{\nabla H}, d, s)$, $d > 2s$. This final bound highlights that under the sample-size-aware complexity scaling $\alpha = R\sqrt{\log R/n}$, the generalization error decays exponentially in the effective resolution scale $R$ and the logarithmic complexity $\log R$, with an additional improvement in rate proportional to $n^{1/4}$. The result reveals how incorporating geometric inductive bias and adaptive complexity can yield sharper generalization guarantees in high-dimensional structured models. $\quad\square$

## D.3 Proof of Proposition 3.2

**Proposition D.3** (Worst-case geodesic deviation under frame rotations). *Let $\gamma : [0,T] \to T^*Q$ be a reference physical data represented as a geodesic. For any Hamiltonian $H_\theta$ define the frame–rotated trajectory $\mathbf{X}_t(h)$ by $\mathbf{X}_t(h) := (q_t, hp_t)$ for $h \in O(d)$. Write $\kappa_\theta := (L_\theta + \|R\|_\infty C_\theta)D$, where $L_\theta$ and $C_\theta$ are the Lipschitz bounds of $\nabla^2\{m, H_\theta\}$ and $\{m, H_\theta\}$, $D$ is the diameter of some compact domain $\mathcal{K}$, and $R$ is the Riemann tensor. Then, for every $t \in [0,T]$,*

$$\sup_{h \in O(d)} d^2\big(\mathbf{X}_t^{\mathrm{eq}}(h), \gamma(t)\big) \leq \sup_{h \in O(d)} d^2\big(\mathbf{X}_t(h), \gamma(t)\big) \leq d^2\big(X_0, \gamma(0)\big) e^{\kappa_\theta t},$$

*where $\mathbf{X}_t^{\mathrm{eq}}$ is generated by a $\Phi$-gauge–equivariant Hamiltonian $H_\theta$ satisfying the same bound with smaller constants $L_\theta^\Phi \leq L_\theta$, $C_\theta^\Phi \leq C_\theta$. Thus gauge equivariance minimizes the worst-case geodesic deviation over all frame actions $h$.*

*Proof.* Let us assume that the physical data trajectory $\gamma : [0,T] \to T^*Q$ forms a geodesic, which satisfies the vanishing connection:

$$\nabla_{\dot\gamma(t)}\dot\gamma(t) = 0. \tag{109}$$

This identity ensures that the acceleration of $\gamma$ with respect to the connection vanishes, meaning that $\gamma$ locally minimizes path length and follows the intrinsic geometry of the manifold. In order to quantify small deviations from the reference geodesic $\gamma$, we define a deviation vector field $J_t$ as the logarithmic map from $\gamma(t)$ to a nearby perturbed point $\mathbf{X}_t$:

$$J_t = \exp_{\gamma(t)}^{-1}(\mathbf{X}_t) = \log_{\gamma(t)}(\mathbf{X}_t) \in T_{\gamma(t)}(T^*Q). \tag{110}$$

This construction allows us to express perturbations within a common tangent space at $\gamma(t)$, facilitating differential analysis. To derive the stochastic differential equation governing $J_t$, we apply the chain rule for Stratonovich differentials adapted to manifold settings. This yields

$$\begin{aligned} dJ_t &= \frac{dF}{dt}(t, \mathbf{X}_t)dt + DF(t, \mathbf{X}_t)[d\mathbf{X}_t] + \frac{1}{2}\sum_k \nabla^2_{E_k, E_k}F(t, \mathbf{X}_t)dt \\ &= \nabla_{\dot\gamma(t)}F_t(\mathbf{X}_t)dt + DF(t, \mathbf{X}_t)[d\mathbf{X}_t] \\ &= \nabla_{\dot\gamma(t)}J_t dt + \nabla_{J_t}\{\mathbf{m}, H_\theta\}(\gamma(t)) \circ d\mathbf{B}_t \end{aligned} \tag{111}$$

where $F(t, x) = \exp_{\gamma(t)}^{-1}(x)$ and $\{E_k\}$ denotes an orthonormal basis of the tangent space. Expanding each term individually, we observe that the time derivative of the logarithmic map corresponds to the covariant derivative along the base curve $\gamma(t)$, giving

$$\frac{dF}{dt} = \frac{\partial}{\partial t}\exp_{\gamma(t)}^{-1} = \nabla_{\dot\gamma(t)}F. \tag{112}$$

Additionally, the differentials involving $d\mathbf{X}_t$ are computed via the pullback under $F_t$, while second-order corrections involving $\nabla^2 F$ are responsible for curvature effects, although these higher-order terms will vanish to leading order under our assumptions.

To measure the growth of deviations quantitatively, we introduce an energy functional $E(t)$ defined by the Riemannian norm of the differential of $J_t$:

$$dE(t) = \frac{1}{2}g\left(dJ_t, dJ_t\right). \tag{113}$$

Here, $g$ denotes the Riemannian metric lifted to the tangent bundle, such as the Sasaki metric if necessary. For notational clarity, we introduce two auxiliary processes:

$$K_t = \nabla_{\dot\gamma(t)}J_t, \quad L_t = \nabla_{J_t}\{\mathbf{m}, H_\theta\}(\gamma(t)). \tag{114}$$

The term $K_t$ represents the covariant derivative of $J_t$ along the trajectory $\gamma(t)$, while $L_t$ captures how the Hamiltonian vector field varies along the perturbation direction $J_t$. Following by standard estimation of SDEs (*i.e.*, $dt^2 = 0$, $dt \circ d\mathbf{B}_t = 0$, $(d\mathbf{B}_t)^2 = dt$), we have

$$\begin{aligned} dE(t) &= \frac{1}{2}g(dJ_t, dJ_t) = \frac{1}{2}g(K_t, L_t)(dt)^2 + g(K_t, L_t)dt \circ d\mathbf{B}_t + \frac{1}{2}g(L_t, L_t)(d\mathbf{B}_t)^2 \\ &= \frac{1}{2}g\left(\nabla_{J_t}\{\mathbf{m}, H_\theta\}(\gamma(t)), \nabla_{J_t}\{\mathbf{m}, H_\theta\}(\gamma(t))\right)dt = \frac{1}{2}\|\nabla_{J_t}\{\mathbf{m}, H_\theta\}(\gamma(t))\|_g. \end{aligned} \tag{115}$$

Finally, differentiating once more with respect to time and expanding the covariant derivatives using standard curvature identities yields

$$
\begin{aligned}
d\ddot{E}(t) &= \frac{d}{dt}\left(\frac{1}{2}g\left(\nabla_{J_t}\{\mathbf{m}, \mathrm{H}_\theta\}(\gamma(t)), \nabla_{J_t}\{\mathbf{m}, \mathrm{H}_\theta\}(\gamma(t))\right)\right) \\
&= g\left(\nabla_{\dot{\gamma}(t)}\nabla_{J_t}\{\mathbf{m}, \mathrm{H}_\theta\}(\gamma(t)), \nabla_{J_t}\{\mathbf{m}, \mathrm{H}_\theta\}(\gamma(t))\right) \\
&= g\left(\nabla_{J_t}\nabla_{\dot{\gamma}(t)}\{\mathbf{m}, \mathrm{H}_\theta\}(\gamma(t)) + R(\dot{\gamma}(t), J_t)\{\mathbf{m}, \mathrm{H}_\theta\}, \nabla_{J_t}\{\mathbf{m}, \mathrm{H}_\theta\}(\gamma(t))\right).
\end{aligned}
\tag{116}
$$

where $R$ denotes the Riemannian curvature tensor. This equation connects the second derivative of the energy with the curvature of the manifold and the structure of the Hamiltonian flow.

While the lifted Riemannian metric $g$ is compatible with the Levi-Civita connection (i.e., $\nabla g = 0$), we can relate the time derivative of the metric pairing along the trajectory as follows:

$$
\frac{d}{dt}g(\mathbf{U}, V) = g(\nabla_{\dot{\gamma}(t)}\mathbf{U}, V) + \mathbf{U}, g(\nabla_{\dot{\gamma}(t)}V),
\tag{117}
$$

where $\mathbf{U} = V = \{m, \mathrm{H}_\theta\}$ denotes the Hamiltonian vector field evaluated along the curve $\gamma(t)$. This relation reflects the fundamental property of metric compatibility and provides a way to track how the energy associated with the Hamiltonian flow evolves along the trajectory. To simplify further, we notice that the covariant derivative of a vector field composed with $\gamma(t)$ can be expanded by the product rule of covariant derivatives along curves:

$$
\nabla_{\dot{\gamma}(t)}\nabla_{J_t}W = \nabla_{J_t}\nabla_{\dot{\gamma}(t)}W + R(\dot{\gamma}(t), J_t)W,
\tag{118}
$$

where $W = \{m, \mathrm{H}_\theta\}(\gamma(t))$, and $R$ denotes the Riemannian curvature tensor. This decomposition separates the effects of directional covariant changes along the flow from the intrinsic curvature-induced distortions arising from the manifold's geometry.

Given the above expansion, we can derive an upper bound for the second derivative of the energy functional $\ddot{E}(t)$ in terms of the norms of relevant geometric quantities:

$$
d\ddot{E}(t) \le \left(\underbrace{\|\nabla_{J_t}\nabla_{\dot{\gamma}(t)}\{\mathbf{m}, \mathrm{H}_\theta\}(\gamma(t))\|_g}_{(A)} + \underbrace{\|R(\dot{\gamma}(t), J_t)\{\mathbf{m}, \mathrm{H}_\theta\}\|_g}_{(B)}\right)\underbrace{\|\nabla_{J_t}\{\mathbf{m}, \mathrm{H}_\theta\}(\gamma(t))\|_g}_{=dE(t)}.
\tag{119}
$$

Here, term (A) corresponds to the covariant second derivative contribution, while term (B) captures the effect of curvature-induced deviations along the geodesic trajectory. These two contributions govern the overall behavior of the energy growth along the stochastic Hamiltonian flow.

Applying Grönwall's inequality to this differential inequality, we obtain an exponential upper bound on the evolution of the energy deviation:

$$
d^2(\mathbf{X}_t, \gamma(t)) = dE(t) \le dE(0)e^{(\sup_s(A)|_s + (B)|_s)t}
\tag{120}
$$

where $(A)$ and $(B)$ represent the supremum bounds of the two terms over the time interval of interest.

This estimate provides a key control over the divergence between the stochastic trajectory $\mathbf{X}_t$ and the reference geodesic $\gamma(t)$ under Hamiltonian diffusion dynamics. We now aim to show that imposing an equivariance constraint on the Hamiltonian function intrinsically reduces the upper bound on the growth of geodesic distance. Specifically, the two key terms in the upper bound (92) can be estimated separately as follows:

$$
(A) := \|\nabla_{J_t}\nabla_{\dot{\gamma}(t)}\{\mathbf{m}, \mathrm{H}_\theta\}(\gamma(t))\|_g \le \|\mathbf{J}_g\| \cdot \|\nabla^2\{\mathbf{m}, \mathrm{H}_\theta\}\|_{op} \cdot \|J_t\|_g \cdot \|\dot{\gamma}(t)\|_g
\tag{121}
$$

$$
(B) := \|R(\dot{\gamma}(t), J_t)\{\mathbf{m}, \mathrm{H}_\theta\}\|_g \le \|R\|_\infty \cdot \|J_t\|_g \cdot \|\dot{\gamma}(t)\| \cdot \|\{\mathbf{m}, \mathrm{H}_\theta\}\|_g.
\tag{122}
$$

where $\|R\|_\infty$ denotes the sup-norm of the Riemannian curvature tensor over the manifold. Next, consider the action of the orthogonal group $\mathrm{O}(d)$ on the configuration and momentum coordinates $(q, p)$. For any $h \in \mathrm{O}(d)$, the equivariant Hamiltonian satisfies

$$
\mathrm{H}_\theta(q, p, \mathbf{U} \cdot h) = \mathrm{H}_\theta(q, \rho(h^{-1})p, \mathbf{U}) \qquad \forall h \in \mathrm{O}(d).
\tag{123}
$$

where $\rho(h) = \begin{bmatrix} h & 0 \\ 0 & I_d \end{bmatrix} \in GL(2d)$ acts on $(q, p)$ coordinates. Differentiating both sides with respect to $(q, p)$ and using the chain rule, we obtain

$$\nabla_{(q,p)} H(q, p, \mathbf{U} \cdot h) = \rho(h) \, \nabla_{(q,p)} H_\theta(q, \rho(h^{-1})p, \mathbf{U}), \tag{124}$$

where we have used the fact that $\rho(h)$ is orthogonal, and that differentiation of $\rho(h^{-1})$ introduces a right multiplication by $\rho(h)$. Differentiating once more yields the second derivative relation:

$$\nabla^2_{(q,p)} H_\theta(q, p, \mathbf{U} \cdot h) = \rho(h) \, \nabla^2_{(q,p)} H_\theta(q, \rho(h^{-1})p, \mathbf{U}) \, \rho(h^{-1}). \tag{125}$$

Since $\rho(h)$ is orthogonal, it preserves the operator norm of tensors. Therefore, for any matrix $A$,

$$\|\rho(h)A\| = \|A\|, \qquad \|\rho(h)A\rho(h^{-1})\| = \|A\| \quad \text{for all } A. \tag{126}$$

This property ensures that the norm of the differential operators remains invariant under frame transformations. Using these observations, we can describe the differential operators acting on the Hamiltonian as

$$\nabla\{\mathbf{m}, H_\theta\} := \nabla_{(q,p)}\{\mathbf{m}, H_\theta\}, \quad \nabla^2\{\mathbf{m}, H_\theta\} := \nabla^2_{(q,p)}\{\mathbf{m}, H_\theta(q, p, \mathbf{U})\}. \tag{127}$$

From the orthogonality and isometry of $\rho(h)$, it follows that

$$\begin{aligned} \sup_{h \in O(d))} \|\nabla_{(q,p)} H_\theta(q, p, \mathbf{U} \cdot h)\| &= \sup_{h \in O(d)} \|\nabla_{(q,p)} H_\theta(q, \rho(h^{-1})p, \mathbf{U})\| \\ &= \sup_{h \in O(d)} \|\nabla_{(q,p)} H_\theta(q, p, \mathbf{U})\| \\ &= \|\nabla_{(q,p)} H_\theta(q, p, \mathbf{U})\|, \end{aligned} \tag{128}$$

and similarly for the second derivative,

$$\sup_{h \in O(d)} \|\nabla^2_{(q,p)} H(q, p, \mathbf{U} \cdot h)\| = \|\nabla^2_{(q,p)} H(q, p, \mathbf{U})\|. \tag{129}$$

Let $\Phi \leq O(d)$ be the gauge subgroup that leaves the Hamiltonian $H_\theta$ invariant, that is,

$$H_\theta(q, p, \mathbf{U} \cdot h) = H_\theta(q, \rho(h^{-1})p, \mathbf{U}), \quad \forall h \in \Phi, \ \mathbf{U} \in O(d),$$

where $\rho : \Phi \to GL(2d)$ is the canonical block embedding $\rho(h) = \mathrm{diag}(h, \mathbf{1}_d)$ acting on $(q, p)$, and the right action $\mathbf{U} \mapsto \mathbf{U} \cdot h$ is free and proper. Consequently, $O(d) \twoheadrightarrow O(d)/\Phi$ is a principal $\Phi$-bundle with a smooth projection $\pi_\Phi : O(d) \to O(d)/\Phi$, and the base $O(d)/\Phi$ is a smooth homogeneous manifold homeomorphic to the coset space $O(d)/\Phi \simeq G/K$ for some closed subgroup $K \simeq \Phi$.

Endow the bundle with the canonical Ehresmann connection induced by the Levi–Civita connection of the configuration manifold. The tangent space at $\mathbf{U} \in O(d)$ splits as $T_U O(d) = \mathcal{H}_U \oplus \mathcal{V}_U$, where $\mathcal{V}_U$ is the vertical subspace associated with the $\Phi$-action. Since $\Phi$ acts by isometries, horizontal lifts preserve the Sasaki metric and the Itô–Stratonovich structure; Brownian noise injected along $\mathcal{H}_U$ descends canonically to the base $O(d)/\Phi$. Let $[\mathbf{U}] \in O(d)/\Phi$ denote a frame class and fix a smooth section $\sigma : O(d)/\Phi \to O(d)$. Define the lifted stochastic flow by

$$\mathbf{X}_t^{[\mathbf{U}]} := \mathbf{X}_t(\sigma([\mathbf{U}])),$$

where $\mathbf{X}_t(\cdot)$ solves the stochastic Hamiltonian system $d\mathbf{X}_t = \{m, H_\theta\}(\mathbf{X}_t) \circ dB_t$. Because $H_\theta$ is $\Phi$-equivariant, the Hamiltonian vector fields satisfy

$$\{m, H_\theta\}(q, p, \sigma([\mathbf{U}]) \cdot h) = \rho(h^{-1})\{m, H_\theta\}(q, \rho(h)p, \sigma([\mathbf{U}])), \quad \forall h \in \Phi,$$

where the lift $\rho$ respects the orthogonal structure. Consequently, both the first and second derivatives obey

$$\nabla_{(q,p)} H_\theta(\sigma([\mathbf{U}]) \cdot h) = \rho(h^{-1})\nabla_{(q,p)} H_\theta(\sigma([\mathbf{U}])),$$
$$\nabla^2_{(q,p)} H_\theta(\sigma([\mathbf{U}]) \cdot h) = \rho(h^{-1})\nabla^2_{(q,p)} H_\theta(\sigma([\mathbf{U}]))\rho(h),$$

which implies the norm invariance relations

$$\|\nabla_{(q,p)} H_\theta(\sigma([\mathbf{U}]))\| = \|\nabla_{(q,p)} H_\theta(\sigma([\mathrm{Id}]))\|, \quad \|\nabla^2_{(q,p)} H_\theta(\sigma([\mathbf{U}]))\| = \|\nabla^2_{(q,p)} H_\theta(\sigma([\mathrm{Id}]))\|.$$

Let $J_t^{[\mathbf{U}]} := \exp_{\gamma(t)}^{-1}(\mathbf{X}_t^{[\mathbf{U}]})$ be the deviation Jacobi field and define the Sasaki energy

$$E_{[\mathbf{U}]}(t) := \frac{1}{2} g\left(J_t^{[\mathbf{U}]}, J_t^{[\mathbf{U}]}\right).$$

Using the stochastic Jacobi equation and applying the curvature identity

$$\nabla_{\dot{\gamma}(t)} \nabla_{J_t} = \nabla_{J_t} \nabla_{\dot{\gamma}(t)} + R(\dot{\gamma}(t), J_t),$$

we obtain a differential inequality controlling the energy:

$$\dot{E}_{[\mathbf{U}]}(t) \leq (A_{[\mathbf{U}]} + B_{[\mathbf{U}]}) E_{[\mathbf{U}]}(t),$$

where $A_{[\mathbf{U}]} := \|\nabla^2 \{m, \mathrm{H}_\theta\}\|_{\mathrm{op}}$ and $B_{[\mathbf{U}]} := \|R\|_\infty \|\{m, \mathrm{H}_\theta\}\|_g$ evaluated along the flow. Due to (96), the terms $A_{[\mathbf{U}]}$ and $B_{[\mathbf{U}]}$ are invariant across $[\mathbf{U}] \in \mathrm{O}(d)/\Phi$. Thus, applying Grönwall's inequality yields the uniform bound

$$E_{[\mathbf{U}]}(t) \leq E_{[\mathbf{U}]}(0) e^{(A+B)t}, \quad d^2(\mathbf{X}_t^{[\mathbf{U}]}, \gamma(t)) \leq dE(0) e^{(A+B)t}, \quad \forall [\mathbf{U}] \in \mathrm{O}(d)/\Phi.$$

In contrast, for the non-equivariant model (without $\Phi$ symmetry), the best bound achievable is

$$d^2(\mathbf{X}_t(h), \gamma(t)) \leq dE(0) e^{(A_{\max}+B_{\max})t}, \quad A_{\max} := \sup_{h \in \mathrm{O}(d)} A(h), \quad B_{\max} := \sup_{h \in \mathrm{O}(d)} B(h),$$

with $A_{\max} \geq A$ and $B_{\max} \geq B$ in general. Hence, we conclude that

$$\sup_{[\mathbf{U}] \in \mathrm{O}(d)/\Phi} d^2(\mathbf{X}_t^{[\mathbf{U}]}, \gamma(t)) = d^2(\mathbf{X}_t^{[\mathrm{Id}]}, \gamma(t)) \leq dE(0) e^{(A+B)t} \leq \sup_{h \in \mathrm{O}(d)} d^2(\mathbf{X}_t(h), \gamma(t)).$$

Since $\mathrm{O}(d)/\Phi$ is compact (being the quotient of the compact Lie group $\mathrm{O}(d)$ by a closed subgroup), the uniform estimate (97) implies exponential $\mathrm{W}_2$-stability of the equivariant diffusion. In contrast, the non-equivariant dynamics suffer from a generally larger exponential factor $(A_{\max}+B_{\max}) > (A+B)$. Therefore, gauge-equivariant Hamiltonian learning provides strictly better control over stochastic deviation and ensures tighter uniform generalization error bounds on curved configuration spaces.

We now further refine the upper bounds for the error terms $(A)$ and $(B)$ by exploiting the geometric structure of the Hamiltonian system. First, we recall that the canonical symplectic matrix $\mathbf{J}_g$ satisfies $\|\mathbf{J}_g\| = 1$, as it acts isometrically on $T^*(\mathcal{Q})$ and preserves the standard Riemannian norm induced by the Sasaki metric. Moreover, since the reference trajectory $\gamma(t)$ is parametrized by arc-length, we have

$$\|\dot{\gamma}(t)\|_g = 1 \quad \text{for all} \quad t \in [0, T]. \tag{130}$$

Furthermore, assuming that both the stochastic trajectory $\mathbf{X}_t$ and the geodesic $\gamma(t)$ remain within a compact subset $\mathcal{K} \subset T^*\mathcal{Q}$, we have

$$\|J_t\|_g = d(\mathbf{X}_t, \gamma(t)) \leq D, \tag{131}$$

where $D := \mathrm{diam}(\mathcal{K})$ denotes the geodesic diameter of $\mathcal{K}$. Under these simplifications, the two contributions $(A)$ and $(B)$ can be bounded more explicitly. Using the fact that $\|\mathbf{J}_g\| = 1$ and $\|\dot{\gamma}(t)\|_g = 1$, the covariant second derivative term satisfies

$$(A) = \left\|\nabla_{J_t} \nabla_{\dot{\gamma}(t)} \{\mathbf{m}, \mathrm{H}_\theta\}(\gamma(t))\right\|_g \leq L_\theta \cdot D, \tag{132}$$

where $L_\theta := \sup_{x \in T^*\mathcal{Q}} \|\nabla^2 \{\mathbf{m}, \mathrm{H}_\theta\}(x)\|_{\mathrm{op}}$ denotes the global Lipschitz bound on the Hessian of the Hamiltonian vector field. Similarly, the curvature-induced deviation term satisfies

$$(B) = \|R(\dot{\gamma}(t), J_t)\{\mathbf{m}, \mathrm{H}_\theta\}(\gamma(t))\|_g \leq \|R\|_\infty \cdot C_\theta \cdot D, \tag{133}$$

where $\|R\|_\infty := \sup_{x \in \mathcal{Q}} \sup_{\|u\|_g = \|v\|_g = 1} \|R_x(u, v)\|_g$ is the global sup-norm of the Riemannian curvature tensor, and $C_\theta := \sup_{x \in T^*\mathcal{Q}} \|\{\mathbf{m}, \mathrm{H}_\theta\}(x)\|_g$ is the global growth bound on the Hamiltonian vector field. Therefore, the sum $(A) + (B)$ admits the uniform estimate

$$(A) + (B) \leq (L_\theta + \|R\|_\infty C_\theta) D, \tag{134}$$

which depends linearly on the diameter $D$ of the compact domain $\mathcal{K}$ and the regularity constants $L_\theta$ and $C_\theta$. If the Hamiltonian $\mathrm{H}_\theta$ is $\Phi$-equivariant under a subgroup $\Phi \subset \mathrm{O}(d)$, the constants $L_\theta$

and $C_\theta$ improve to $L_\theta^\Phi$ and $C_\theta^\Phi$ respectively, reflecting the additional regularity induced by gauge symmetry. In this case, we obtain the sharper bounds

$$(A)_{\text{eq}} \leq L_\theta^\Phi \cdot D, \quad (B)_{\text{eq}} \leq \|R\|_\infty \cdot C_\theta^\Phi \cdot D, \tag{135}$$

and the total deviation is controlled by

$$(A)_{\text{eq}} + (B)_{\text{eq}} \leq (L_\theta^\Phi + \|R\|_\infty C_\theta^\Phi)D, \tag{136}$$

where $L_\theta^\Phi < L_\theta$ and $C_\theta^\Phi < C_\theta$ due to the symmetry reduction. If the space is assumed to have bounded scalar curvature $\kappa_{max}$, one can improve the by following Lemma C.3 as follows:

$$(A)_{\text{eq}} + (B)_{\text{eq}} \leq (L_\theta^\Phi + \frac{\kappa_{max}}{d(d-1)}C_\theta^\Phi)D. \tag{137}$$

Substituting these improved bounds into the Grönwall estimate derived earlier, we conclude that the equivariant stochastic Hamiltonian dynamics exhibits exponentially tighter control of the geodesic deviation compared to the general non-equivariant case, with an explicit exponent that scales linearly with the curvature bounds, Hamiltonian regularity, and the diameter of the compact reachable set. $\quad\square$

**Corollary D.4** (Worst-case $W_2$ deviation). *Under the setting of Proposition D.4 assume, in addition, that the initial state is deterministic, $X_0 = \gamma(0)$. For each frame rotation $h \in \mathrm{O}(d)$, let us set*

$$\mathbb{P}_t(h) := \mathbf{Law}\big[\mathbf{X}_t(h)\big], \qquad \mathbb{P}_t^{\text{eq}}(h) := \mathbf{Law}\big[\mathbf{X}_t^{\text{eq}}(h)\big].$$

*Then, for every $t \in [0, T]$,*

$$\sup_{h \in \mathrm{O}(d)} \mathcal{W}\big(\mathbb{P}_t^{\text{eq}}(h), \delta_{\gamma(t)}\big) \leq \sup_{h \in \mathrm{O}(d)} \mathcal{W}\big(\mathbb{P}_t(h), \delta_{\gamma(t)}\big) \leq d^2\big(\mathbf{X}_0, \gamma(0)\big)e^{\kappa_\theta t}$$

*with the same $\kappa_\theta$ as in Proposition D.4. Hence gauge equivariance minimises the worst-case 2-Wasserstein divergence from the reference geodesic over all frame actions $h$.*

*Proof.* Fix a rotation $h \in \mathrm{O}(d)$ and let $\mathbb{P}_t(h) = \mathbf{Law}[\mathbf{X}_t(h)]$. Because the reference point $\gamma(t)$ is deterministic, the unique optimal coupling between $\mathbb{P}_t(h)$ and $\delta_{\gamma(t)}$ is the map $\mathbf{X}_t(h) \mapsto \gamma(t)$. Hence

$$\mathcal{W}\big(\mathbb{P}_t(h), \delta_{\gamma(t)}\big) = \mathbb{E}\, d^2\big(\mathbf{X}_t(h), \gamma(t)\big). \tag{138}$$

Proposition C.3 provides the uniform pathwise bound $d^2\big(\mathbf{X}_t(h), \gamma(t)\big) \leq d^2(X_0, \gamma(0))\, e^{\kappa_\theta t}$ for every $h$. Taking expectations preserves the same right–hand side yields

$$\mathcal{W}\big(\mathbb{P}_t(h), \delta_{\gamma(t)}\big) \leq d^2\big(X_0, \gamma(0)\big)\, e^{\kappa_\theta t} \quad \forall h \in \mathrm{O}(d). \tag{139}$$

The same reasoning with the gauge–equivariant trajectory $\mathbf{X}_t^{\text{eq}}(h)$ gives an analogous inequality with the smaller constants $L_\theta^\Phi, C_\theta^\Phi$; by Proposition C.3 this already realizes the infimum over $h$. By taking supremum operator over $h \in \mathrm{O}(d)$ in both, it completes the chain of inequalities stated in the corollary. $\quad\square$

