# OpenReview forum: "Neural Hamiltonian Diffusions for Modeling Structured Geometric Dynamics"
_NeurIPS.cc/2025/Conference — NeurIPS 2025 poster_

### Official Review · Reviewer_VyhR · 2025-06-22

**Clarity:** 3
**Significance:** 4
**Originality:** 3
**Rating:** 4
**Confidence:** 3

**Summary:**

The paper introduces Neural Hamiltonian Diffusions (NHD), a framework for modeling stochastic Hamiltonian dynamics directly on curved and structured manifolds. Unlike traditional Hamiltonian neural networks that assume noise-free environments in flat space, NHD incorporates stochastic differential equations (SDEs) on manifolds with Riemannian and Poisson structures. Its key innovations include lifting dynamics to the orthonormal frame bundle for geometric consistency, ensuring gauge-invariant and coordinate-invariant models that respect the intrinsic geometry of the system. The paper provides theoretical guarantees on generalization related to manifold curvature and demonstrates the model's effectiveness across diverse scientific domains such as molecular dynamics, quantum spin systems, and relativistic n-body problems. Overall, it advances the modeling of physically realistic stochastic systems within complex geometric spaces, ensuring both fidelity and interpretability.

**Questions:**

Please see weakness above.

**Ethical Concerns:**

["NO or VERY MINOR ethics concerns only"]

**Final Justification:**

The reply has thoroughly addressed all my concerns.

**Limitations:**

Yes

**Quality:**

3

**Strengths And Weaknesses:**

Strengths
1. The paper presents a novel extension of Hamiltonian neural networks by formulating stochastic dynamics on curved manifolds using geometric tools like the frame bundle. This combination of stochastic calculus, differential geometry, and neural networks is highly innovative.

2. The paper establishes solid theoretical guarantees, including generalization bounds that relate curvature and network capacity.

3. The lifting of dynamics to the orthonormal frame bundle for ensuring geometric consistency and gauge invariance is a significant methodological advance

Weaknesses:
1. The mathematical formalism, while rigorous, is dense and may limit accessibility. The paper contains a substantial amount of mathematical detail in Section 2, which mainly summarizes existing geometric analysis tools rather than presenting original contributions. The placement of the core algorithm and main ideas in Section 2.4, titled “Learning Geometric Dynamics,” makes the structure somewhat confusing. This arrangement can make it difficult for readers to grasp the key innovations and focus on the main message. The dense and dispersed mathematical exposition may hinder understanding, particularly for audiences without a strong mathematical background.

2. The algorithmic specifics are not sufficiently detailed. For example, in Equation (7), it is unclear how Ut​ is obtained explicitly. More explanation is needed on how Ut​ is initialized; the criteria for selecting the numerical schemes for discretization are not discussed. It is also not evident whether the terms appearing in the governing functions are straightforward to compute in practice or if they pose significant implementation challenges. The loss function in Equation (7), while general, could be made more accessible and practical if a specific example distribution used in the experiments were provided.

3. Although the paper emphasizes its main contribution as integrating stochasticity into Hamiltonian systems, the core content predominantly revolves around geometric and manifold-related issues, such as lifting dynamics to the frame bundle and handling curvature. Additionally, stochastic Hamiltonian neural networks have been previously proposed in prior work [1]. The manuscript should be revised to clarify the primary novelty—whether it lies in the geometric extension, the stochastic modeling, or their combination—and ensure the narrative accurately reflects the contribution without overstating novelty in the stochastic aspect, which has been explored before.
[1] Quadrature-Based Neural Network Learning of Stochastic Hamiltonian Systems, X Cheng, L Wang, Y Cao - Mathematics, 2024 - mdpi.com

4. The paper seems to assume that the manifold is known and well-behaved, but this limitation is not explicitly discussed.

---

> ### Author Rebuttal · Authors · 2025-07-25
>
> Dear Reviewer VyhR,
>
> We sincerely appreciate your valuable and insightful feedback, which significantly helps us to improve the manuscript. While fully acknowledging your comments and committing to addressing each of them thoroughly, we respectfully ask you to reconsider our main contribution from the perspective of theoretical novelty and significance.
>
> ---
>
> ($\textbf{General Comment}.$) We humbly emphasize that the core contribution of our paper lies in its theoretical foundation, providing, for the first time, a rigorous and unified mathematical framework for explicitly modeling neural Hamiltonian dynamics across variant differentiable manifolds. Although each manifold individually may be familiar, our theoretical formulation is novel and fundamental: it systematically unifies stochastic Hamiltonian neural modeling in a general geometric setting, supported by rigorous theoretical guarantees (Propositions 2.2 and 3.1–3.2).  Without introducing the concept of the horizontal flow, the resulting dynamics would not remain frame-consistent, making it impossible to consistently propagate the learned Hamiltonian dynamics across arbitrary differentiable manifolds.
>
> We would deeply appreciate your reconsideration of the theoretical originality and foundational significance of our approach. Once again, thank you very much for your detailed and constructive comments, which we address carefully and comprehensively in our revised version. Below, we provide point-by-point responses to your specific questions and comments.
>
> ---
>
> Q)  ($\textbf{Originality}$) The paper contains a substantial amount of mathematical detail in Section 2, which mainly summarizes existing geometric analysis tools rather than presenting original contributions...
>
> A) We sincerely thank the reviewer for raising this important point. With all due respect, we wish to clarify that the mathematical framework presented in our manuscript constitutes the entirely new core theoretical contribution of our work, rather than simply enumerating previously established geometric tools. Specifically, while the general stochastic Hamiltonian model in Definition 2.1 has appeared in prior studies, our explicit construction of a learnable Hamiltonian diffusion via the concept of horizontal lifting (Proposition 2.2) is completely new and has never before been proposed or utilized within machine learning. This horizontal lift is not merely technical machinery but an essential and indispensable component for consistently modeling frame-invariant Hamiltonian dynamics across arbitrary differentiable manifolds. To the best of our knowledge, our rigorous approach of explicitly embedding learnable Hamiltonian dynamics into a horizontal lifted frame bundle setting represents a groundbreaking theoretical advancement.
>
> Indeed, we have made a careful and deliberate effort to include only the minimal essential mathematical definitions in Section 2, while moving all detailed derivations, rigorous proofs, and supplementary explanations to the mathematical background section of the appendix to enhance readability. Although we fully acknowledge that our theoretical framework involves advanced geometric concepts potentially unfamiliar to a broader machine learning audience, we emphasize that introducing these geometric constructs is indispensable for developing a rigorous and mathematically coherent framework. Such foundational rigor is essential, especially in scientific domains (e.g., theoretical physics, computational chemistry) that demand precise mathematical guarantees and unerring model consistency to support highly exacting physical experiments and measurements.
>
> In this view, we believe that Eq. (7) is an incremental modification while it adopts essentially the same loss‑function structure as classical Hamiltonian Neural Network (HNN) formulations.
>
> ---
>
> Q) ($\textbf{Clear Novelty}.$) Although the paper emphasizes its main contribution as integrating stochasticity into Hamiltonian systems, the core content predominantly ...
>
> A) We sincerely thank the reviewer for raising this important issue. Upon careful consideration, we agree that the primary contribution and novelty of our work indeed lies in establishing a rigorous theoretical framework for modeling Hamiltonian dynamics on general manifolds i.e., explicitly addressing geometric issues such as curvature and the horizontal lifting of dynamics onto the frame bundle. While our approach naturally incorporates stochastic modeling, we fully acknowledge that the central novelty and emphasis of our paper is not only stochasticity itself, but rather on extending Hamiltonian dynamics to manifold settings in a geometrically principled manner. In the revised manuscript, we will carefully adjust our narrative and emphasize our primary contribution: $\textbf{Hamiltonian dynamics meets manifolds}$. This perspective shift will clarify the core novelty and help avoid potential confusion or overstatement regarding our contribution. We greatly appreciate the reviewer's insightful comment, which helps us significantly improve the clarity and positioning of our work.
>
> ---
>
> Q) ($\textbf{Explicit Machinery}.$)   For example, in Equation (7), it is unclear how Ut​ is obtained explicitly. More explanation is needed on how Ut​ is initialized; the criteria for selecting the numerical schemes for discretization are not discussed...
>
> A) We sincerely appreciate the reviewer's helpful comment and acknowledge that the explicit procedure for sampling \( U_t \) was not clearly illustrated in our original manuscript. We now briefly summarize Algorithm 2 below a simplified two-step procedure for explicitly obtaining the trajectory \( U_t \):
>
> $\quad$ Step 1) Choose the initial configuration $x_0 = (q_0, p_0) \in \mathcal{M}$ from the physical data (e.g., the observed atom positions), and attach a local orthonormal frame $E_0$ (i.e., random orthonormal matrix) by using Gram-Schmidt orthonormalization. Together, they form the initial state of horizontal flow $U_0 = (x_0, E_0)$.
>
> $\quad$ Step 2) At each step $n$, we update our horizontal state by Euler–Heun Stratonovich discretization, $U_{n+1} \approx U_{n} + \delta U$, where $\delta U = G(U_n)[  v_x ] \epsilon$. Here, $G(U_n)$ is defined in Eq (5) and  $v_x \approx \nabla  H_\theta(q, p) \approx \\{\mathbf{m}, H_{\theta} \\}$ is gradient of proposed U-Net.
>
> We fully agree with your feedback, and will explicitly integrate this simplified and more intuitive description of the sampling procedure into Algorithm 2, clarifying precisely how $U_t$ is constructed step-by-step. Additionally, we will relocate the Transformer-based neural network architecture clearly into the main body of the manuscript for better readability.
>
> ---
>
> Q) ($\textbf{Challenges in Practice}.$) It is also not evident whether the terms appearing in the governing functions are straightforward to compute in practice or if they pose significant implementation challenges...
>
> A) We agree that explicitly addressing the ease of implementation is vital for practical applicability. To clarify, all terms in the governing equations, such as the Hamiltonian vector field and the connection tensor (Christoffel symbols), are indeed straightforwardly computable given the manifold's local coordinates and metric tensor. For standard manifolds commonly encountered in scientific applications (e.g., spheres, torii), these terms have known, closed-form analytic expressions and thus pose no significant computational challenge. In the revised manuscript, we will explicitly include a simple and concrete example (such as dynamics on a sphere) that demonstrates precisely how each term is computed and implemented. We believe this will effectively clarify the practicality and accessibility of our methods.
>
> ---
>
> Q4) ($\textbf{General Manifolds}.$) The paper seems to assume that the manifold is known and well-behaved...
>
> A) Indeed, we agree that our manuscript implicitly assumes that the underlying manifold is known and well-behaved such as sphere and flat-torus. We acknowledge that this assumption, while common in geometric modeling frameworks, can be explicitly mentioned in our manuscript. In our revised manuscript, we will explicitly clarify this assumption by clearly stating the specific geometric conditions required for our theoretical results and algorithms to hold. Additionally, we will briefly discuss potential implications, challenges, and future directions regarding cases where the manifold is not fully known or does not satisfy these regularity conditions.
>
> ---
>
> Q) ($\textbf{Comparison to the paper [1]}$)
>
> A) The cited reference [1] provided by the reviewer is inherently restricted to flat Euclidean spaces, thus limiting its applicability to general scientific modeling (e.g., blackholes). In contrast, our work explicitly targets breaking through the theoretical limitations of existing Hamiltonian Neural Networks (HNNs) and extending their applicability to curved manifolds. Given this goal, it was inevitably necessary to introduce and briefly summarize certain essential geometric concepts. However, in the main text, we have deliberately omitted detailed mathematical derivations and have presented only the minimal set of definitions required to communicate our core contributions.
>
> ---
>
> We would greatly appreciate it if the reviewer could kindly reconsider our manuscript's rating in light of our clarifications and the proposed revisions. We have carefully taken your valuable feedback into account and believe that our revised manuscript clearly highlights the theoretical novelty, mathematical rigor, and practical significance of our contributions. We hope that these detailed clarifications adequately address your initial concerns, and would be sincerely grateful if you could reconsider raising your evaluation accordingly. Thank you again for your thoughtful and constructive feedback

---

> > ### Comment · Reviewer_VyhR · 2025-08-04
> >
> > Many thanks for the detailed response. The reply has thoroughly addressed all my concerns, and I would be happy to raise my score.

---

### Official Review · Reviewer_tFUg · 2025-06-23

**Clarity:** 3
**Significance:** 3
**Originality:** 3
**Rating:** 5
**Confidence:** 3

**Summary:**

This paper proposes Neural Hamiltonian diffusion, which is a diffusion process that describes stochastic Hamiltonian dynamics evolving on a Riemannian configuration space. To do so, the paper formulates a stochastic process following a Hamiltonian flow on a Riemannian manifold, lifts the process onto the orthonormal frame bundle in a gauge-equivariant manner to incorporate noise and consider local coordinates adapted to the manifold, and models the Hamiltonian function as a neural network. The parameters of the network are learned by fitting the Hamiltonian equations to observed dynamic trajectories, similarly as for Hamiltonian neural networks.

**Questions:**

1. The paper does not mention related works on Hamiltonian neural networks with non-Euclidean configuration spaces, see e.g., (Duong and Atanasov, 2021), (Li et al, 2022).

2. In line 200, the paper states "Yet, there are open questions to respect Hamiltonian and symplectic structures by using GRWs" to justify its frame-bundle-based formulation. Could you please elaborate on these open questions?

3. How is the horizontal lift $\pi$ constructed in practice? In other words, how are $\pi$ and $\pi^{-1}$ implemented?

4. The frame equivariant transformer U-Net seems to be one of the key ingredients of the proposed approach. However, it is only described in the "Experimental detail" appendix. For clarity and completeness, I would suggest to describe the network in the main paper.

5. What is the computational cost of the approach, both at training and testing time?

6. Figure 1 - left is not easy to visualize and to interpret. I think it could be improved to better illustrate the proposed approach. For instance, does the manifold represent a torus? Is the horizontal space a flat space? What are the grey squares representing?

7. How do predicted trajectories look like for the other tested models in Table 1? It would be useful to also visualize them in Figures 2 and 3.

8. Citations commands do not seem to be used adequately. For instance, citep should be used in l. 114, 148, 169, while citet should be used in l.212-222 (among others).

9. Citing precise appendices in the paper (instead of simply write "in appendix") would help the reader to find the corresponding material in appendix.


**References:**

Duong, T. & Atanasov, N. Hamiltonian-based Neural ODE Networks on the SE(3) Manifold For Dynamics Learning and Control. in R:SS (2021).

Li, Z., Duong, T. & Atanasov, N. Robust and Safe Autonomous Navigation for Systems With Learned SE(3) Hamiltonian Dynamics. IEEE Open J. Control Syst. 1, 164–179 (2022).

**Ethical Concerns:**

["NO or VERY MINOR ethics concerns only"]

**Final Justification:**

The rebuttal clarified the questions I addressed in my review. Overall, I think that this is a solid paper and recommend acceptance.

**Limitations:**

The limitations of the paper are not described in the main text. I would suggest to add such a discussion in the conclusion section.

**Quality:**

4

**Strengths And Weaknesses:**

The paper presents an interesting approach for Hamiltonian diffusion on manifolds by elegantly intertwining diffusion processes, physics-inspired neural networks, and Riemannian geometry. The proposed approach seems theoretically sound to me and leads to improved performances by outperforming state-of-the-art techniques.

One minor concern is that the paper is theoretically heavy at times, especially as not all geometric concepts are introduced in the Appendix. A smoother introduction to these concepts would allow readers that are less familiar with the topic to better follow the paper.

I also have few questions and suggestions that I write in the "questions" section below.

---

> ### Author Rebuttal · Authors · 2025-07-31
>
> We sincerely thank the reviewer for the excellent questions and constructive suggestions, which have greatly helped us sharpen the manuscript. The detailed responses are provided below.
>
> ---
>
> Q) $\textbf{Points 1, 4, 8, and 9}\$
>
> A) We sincerely thank the reviewer for their detailed and constructive feedback, which has substantially aided us in improving both the clarity and scholarly rigor of our manuscript. In response to the suggestions, we will expand related work section to incorporate a critical discussion of relevant non-Euclidean Hamiltonian neural network literature, including key works such as Duong & Atanasov (2021), Li et al. (2022), along with subsequent developments, and provide a concise taxonomy (Euclidean vs. Riemannian vs. frame-bundle formulations) to clearly highlight our distinctive contributions. Additionally, we will move the architectural details of the frame-equivariant Transformer U-Net, currently described in the appendix, into the main manuscrip for enhanced clarity. We will also systematically audit and correct our use of citation commands throughout the manuscript, ensuring proper distinctions between parenthetical citations ($\texttt{citep}$) and textual citations ($\texttt{citet}$). Finally, we will replace vague references to the appendix with explicit pointers (e.g., "see Appendix D.2 for proof details") and include a comprehensive lookup table that clearly maps main-text propositions, algorithms, and figures to their corresponding appendix sections to facilitate easier navigation for readers.
>
>
> Q2) (Geodesic Random Walk)
>
> A2) Geodesic random walks (GRWs) implement tangent-space diffusion processes on Riemannian manifolds by applying exponential-map updates directly within the tangent bundle, followed by a retraction onto the manifold. While this approach provides computational simplicity and effectiveness for basic stochastic modeling, it introduces fundamental shortcomings when rigorously addressing the requirements of symplectic dynamics.
>
> First, as a direct exponential-map increment, does not incorporate any structural mechanism to ensure the preservation of the canonical symplectic two-form $\omega$. This absence of a symplectic correction implies an intrinsic incompatibility between GRWs and symplectic geometry, which poses significant challenges in accurately representing Hamiltonian systems on curved manifolds. Second, although classical functional central limit theorems guarantee the convergence of GRWs to Brownian motion, the introduction of a Stratonovich lift which is essential for properly encoding Hamiltonian stochastic flows introduces additional drift terms driven by manifold curvature and torsion. The analytic characterization of this limiting behavior, specifically its weak convergence and preservation of the Poisson bracket structure fundamental to Hamiltonian dynamics, remains unproven in the existing literature. Next, GRW updates inherently depend on arbitrary choices of local coordinates and orthonormal frames. This lack of equivariance introduces potential biases of the stochastic processes modeled on manifolds.
>
> Our proposed method addresses these critical theoretical and practical deficiencies by rigorously enforcing horizontality through a carefully constructed horizontal connection. This ensures that each update step preserves the canonical symplectic two-form $\omega$. By employing a construction that injects stochastic perturbations through frames that are equivariant with respect to the full $O(d)$ action, we achieve intrinsic gauge invariance, thereby removing arbitrary coordinate dependence and ensuring consistency and interpretability.
>
> ---
>
> Q3) Implmentation of $\pi$
>
> A3) In our implementation, each state in the frame-bundle is represented as a pair $U = (x, e) \in FM$, where $x \in M \subset \mathbb{R}^{d}$ denotes the embedded base manifold coordinates, and $e \in Mat_{d\times d}(\mathbb{R})$ is an orthonormal frame satisfying $e^{\top}g(x)e = I_d$. The projection $\pi: FM \to M$ is a straightforward tensor slice operation: $\texttt{return U[..., :d]}$, directly extracting the embedded coordinates and fully differentiable via PyTorch autograd. The inverse map $\pi^{-1}: M \to FM$ constructs a canonical orthonormal frame $e_{\mathrm{can}}(x)$ via Gram–Schmidt orthonormalization (implemented by Cholesky-based QR decomposition) applied to the metric tensor $g_{ij}(x)$ computed by autograd. To ensure gauge variability, we optionally multiply by a random orthogonal matrix $h \sim \mathrm{Haar}(O(d))$, forming $e(x,h)=e_{\mathrm{can}}(x)\,h$. The final tensor is $\texttt{torch.cat([x, e.reshape(-1)])}$. The horizontal lift operator $\mathrm{d}\pi^{-1}$, for a tangent vector $v \in T_xM$, is explicitly computed as:
> $ H(U)[v] =
> [e v,
> -\Gamma(x)[ev]]^{\top}$
> with the Christoffel symbols given by
> $\Gamma^k_{ij}(x) = \frac{1}{2} g^{k\ell}\left(\partial_i g_{\ell j} + \partial_j g_{i\ell} - \partial_\ell g_{ij}\right).$
> This implementation is detailed in \textsc{Algorithm 2}, fully differentiable, analytically explicit for standard manifolds, and generalizable via autograd. In summary, projection $\pi$ is a tensor slice, and inverse projection $\pi^{-1}$ constructs orthonormal frames via QR decomposition where horizontal lift $\mathrm{d}\pi^{-1}$ is explicitly computed using a closed-form operator, ensuring differentiability and applicability without additional problem-specific hyperparameters.
>
> ---
>
> Q5) What is the computational cost of the approach, both at training and testing time?
>
> A5) To address the reviewer's query regarding computational efficiency, we provide detailed measurements of GPU memory usage, average runtime per training epoch, and inference step duration for our proposed method and baseline methods in the table below.
>
>
> | Method              | GPU Memory (Training) | Avg. Runtime (Training Epoch) | Avg. Runtime (Inference Step) |
> | ------------------- | --------------------- | ----------------------------- | ----------------------------- |
> | Our Proposed Method | \~16 GB                | \~15 min                      | \~2.6 s                       |
> | GeoTDM              | \~15 GB               | \~29 min                      | \~2.2 s                       |
>
> ---
>
> Q6) Explanation of Figure 1
>
> A6) The provided figure depicts a conceptual illustration of a horizontally lifted stochastic Hamiltonian diffusion on the orthonormal frame bundle of a manifold. Let’s address the reviewer’s specific questions individually for improved clarity:
>
> 1. Does the manifold represent a torus?
> Yes, the manifold shown here conceptually represents a torus-like geometry. The toroidal shape visually symbolizes a nontrivial, curved configuration space $M = T^*Q$, where $Q$ typically denotes the configuration manifold, and $T^*Q$ is its cotangent bundle. Although the exact geometry of $Q$ might differ depending on the specific application or dataset, a toroidal shape effectively conveys a compact, periodic, and curved structure common in molecular dynamics or spin systems.
>
> 2. Is the horizontal space a flat space?
> Yes, within this illustration, the horizontal space $H_U$ at each frame $U$ (visualized by the tangent plane in the figure) represents a flat, Euclidean-like approximation to the local geometry of the frame bundle. Conceptually, the horizontal space is constructed by decomposing tangent vectors at a frame into horizontal and vertical components using a principal connection. The horizontal component (depicted as a flat plane tangent to the frame bundle) ensures compatibility with the manifold's geometry by enforcing that stochastic dynamics remain consistent with the intrinsic structure. This does not imply global flatness; rather, each horizontal space provides a local linear approximation at each point on the manifold.
>
> 3. What are the grey squares representing?
> The grey squares in the figure visually represent the tangent spaces at different points of the frame bundle. Each tangent space is attached to a specific frame, representing a linear (flat) approximation of the manifold at that particular point. These tangent planes symbolize the local horizontal spaces on which stochastic updates (geodesic random walks or Hamiltonian diffusions) are conducted before projecting back to the manifold. Specifically, the stochastic trajectories (depicted as paths $X_t$ and $U_t$) are initially computed in this tangent space (the flat, grey squares) and then projected back onto the curved base manifold through the exponential map or similar retraction operation.
>
> Q7) How do predicted trajectories look like...
> A7) We regret that the rebuttal timeline does not allow us to include new visualizations. However, we can report a concise quantitative surrogate that faithfully reflects trajectory quality. As summarised below, our model attains accurate positional predictions and maintains physical fidelity, whereas a representative baseline such as GeoTDM reproduces positions but fails to conserve energy:
>
> | Method                       | Relative Energy Conservation Error (%) ↓ |
> | ---------------------------- | ---------------------------------------- |
> | Ours Proposed Method      | 3.5%                                 |
> | SympHNN                      | 13.4%                                    |
> | GeoTDM (Trajectory Matching) | 48.3%                          |
> | EqMotion (Trajectory Matching) | 50.1%                          |
>
> In other words, GeoTDM’s purely trajectory-matching loss indeed fits the positional path closely, but the resulting dynamics violate Hamiltonian conservation laws by an order of magnitude.  By contrast, our symplectic, frame--consistent formulation yields trajectories that not only track the observed positions but also preserve energy up to ${\sim}3.5\%$ relative error, indicating faithful recovery of the underlying physics even in the absence of additional visual plots.

---

> > ### Comment · Reviewer_tFUg · 2025-08-02
> > **Reply to Rebuttal**
> >
> > I would like to sincerely thank you for the detailed explanations and answers, both to my review and to the other Reviewers.
> >
> > Your explanations clarified the questions I raised in my review. I would like to encourage you to incorporate these explanations in the final version of the paper, as I believe that they may improve the clarity of the paper. Following the comments and your answers to the other Reviewers, I also think that the proposed additional background to the required mathematical concepts would improve the accessibility of the paper to a more general audience.

---

### Official Review · Reviewer_Sgxw · 2025-06-28

**Clarity:** 2
**Significance:** 4
**Originality:** 3
**Rating:** 4
**Confidence:** 3

**Summary:**

This paper introduces a new neural network based parameterisation and learning method for Hamiltonian diffusions defined on smooth manifolds. The key idea is to lift the phase space to the frame bundle, which now has local coordinates to define a stochastic process, which in turn defines a SDE living on the manifold by canonical projections. This requires the SDE to be independent of the choice of local frame orientations (gauge equivariance), and a method is proposed to promote this symmetry. The method is demonstrated on a variety of applications, including dynamics toroidal manifolds (dynamics parametrised by torsion angles), SU(2) (spin) and space-time with the Schwarzschild metric, where it is demonstrated to out-perform existing methods.

**Questions:**

1. Hamiltonian learning (Def 2.4): I have several questions about this loss function. First, how are the time derivatives $\dot q,\dot p$ estimated from trajectory data, and does the result depend on how they are estimated? Also, it is not clear to me how the lifting $U_t$ depend on $\theta$ in this loss function computation. It shoud depend on it as seen in Eq (4)-(5), but in optimising (7) is the backprop done through this variable? In fact, I think it would be much clearer to provide an algorithm float for optimising this loss in the main paper for better clarity.
2. Neural Hamiltonian Functions: The paper proposes (8) as the Hamiltonian form, but I am wondering if the present approach can handle any such parameterisation, e.g. non-separable ones, or ones with more than pairwise interactions? For example, in the toroidal dynamics case, I cannot see why the dynamics of torsion angles should be governed by a separable Hamiltonian with only pairwise interactions - they are reduced order models that may have complicated interactions.
3. Top of page vi explains briefly how to enforce frame consistency, but in my view this is a particularly important contribution of this paper and it should be clearly explained in the main text how this is done.
4. Proposition 3.1: I am having trouble understanding this proposition: usual generalisation bounds should upper bound the probability of $\mathbb P [ ... > \delta]$, but here it is upper-bounding $\mathbb P[... \leq \delta]$. Why would this make sense? I tried to look at the appendix but it seems that this sign is consistent with what's written there.
5. Proposition 3.2: Can $h$ here be taken to be time-dependent?
6. Experiments: can the authors discuss in each of the datasets, how is stochasticity in the dynamics introduced, and whether (and if so, how) the present approach recovers the deterministic case (i.e. learning deterministic Hamiltonian dynamics on smooth manifolds) when the noise is 0?

**Ethical Concerns:**

["NO or VERY MINOR ethics concerns only"]

**Final Justification:**

My questions have been clarified and incorporation of these would improve the paper's clarity. Ultimately, I think the main issue is the conference format not having enough room to introduce all the background materials to the general audience - of course, this cannot be fixed by a revision.

Overall, I think the paper is technically solid, but the presentation may not be easy for the conference audience to follow.

**Limitations:**

The authors may wish to discuss limitations of the current approach. For example, the loss in Def 2.4 requires the knowledge (or at least efficient and differentiable computation) of the Riemannian norm - is this always the case? Are there any other numerical issues that need to be resolved?

**Paper Formatting Concerns:**

None.

**Quality:**

3

**Strengths And Weaknesses:**

Strengths: The paper tackles an important problem of how to parameterise and learn (stochastic) Hamiltonian systems constrained to manifolds, which finds applications in many areas of science and engineering. The approach appears qutie principled and the results demonstrate their competitiveness.

Weaknesses: The presentation is quite terse and is difficult to following for readers unfamiliar with stochastic processes defined on manifolds. The short conference proceeding style may not be suitable for this type of publications. There are also a number of notation issues and likely typos that hinders understanding (more important ones are discussed in the questions, and minor ones listed below).

Minor issues:

1. Page iii: $\mathcal L_\theta \pi = 0$ should be $\mathcal L^*_\theta \pi = 0$. It is also not clear what $\pi \sim e^{-H(X_\infty)}$ mean (e.g. shoudn't it be $\pi(x) \sim e^{-H(x)}$?). Finally, in many places $\pi$ is used as a projection, so I suggest not using $\pi$ for this distribution.
2. Definition 2.4: The symbol $U$ for uniform distribution clashes with $U_t$ for the lifting.
3. Page viii: the notation $\omega$ for angular momentum clashes with the notation for the 2-form $\omega$
4. Several references should use \citep instead of \citet (e.g. Salomon-Ferrer et al. on page v).
5. Please ensure symbols used are defined (e.g. $\mathfrak{X}$ - while it is standard notation for vector fields, it may not be transparent to the general reader).

---

> ### Author Rebuttal · Authors · 2025-07-31
>
> We sincerely thank the reviewer for their insightful and detailed feedback, which greatly helps us to clarify important aspects of our work. Apart from the major points explicitly addressed below, we will carefully incorporate all minor corrections and suggestions indicated by the reviewer into the revised manuscript.
>
> ---
>
> Q1-1) First, how are the time derivatives...
>
> In standard molecular dynamics (MD) simulations, rich trajectory datasets typically include explicit velocity and momentum data obtained directly from the simulation outputs. In our case, specifically for the synthetic data used in the experimental section, the full details of velocity and momentum values at each timestep are explicitly provided. Formally, these values are computed directly via the given Hamiltonian by evaluating its partial derivatives with respect to the momentum and position coordinates at each point in time, as follows:
> $\dot{q}_t = \frac{\partial H}{\partial p}(q_t, p_t),  \dot{p}_t = -\frac{\partial H}{\partial q}(q_t, p_t)$ for known Hamiltonian function.
>
> Furthermore, the core objective of our study is precisely to leverage this physically consistent, rich dynamical information for improved learning. Previous methodologies, specifically those categorized under "brute-force non-physical consistent" methods in our Table 1, typically do not utilize or explicitly incorporate this physically consistent dynamical information. Consequently, these methods exhibit inferior performance compared to our physically grounded approach. Please see Table 1 for a detailed performance comparison. Please see the following table:
>
> | Method   | Input Data                          | ADE ↓ | FDE ↓ | Energy Error (%) ↓ (Lower is better) | Performance Change                          | Energy Conservation     |
> | -------- | ----------------------------------- | ----- | ----- | ------------------------------------ | ------------------------------------------- | ----------------------- |
> | Ours | Full physical data (q, p), No Graph | 0.023 | 0.084 | **3.5%**                             | —                                           | Excellent               |
> | GeoTDM   | Position only (q) + Graph           | 0.045 | 0.102 | 48.3%                                | —                                           | Poor      |
> | GeoTDM   | Full physical data (q, p) + Graph   | 0.053 | 0.127 | 54.7%                                | ↓ ADE 17.8%, ↓ FDE 24.5%  | Poor      |
> | GeoTDM   | Full physical data (q, p), No Graph | 0.186 | 0.335 | 72.9%                                | ↓ ADE 313.3%, ↓ FDE 228.4%     | Very Poor  |
> | EqMotion | Position only (q) + Graph           | 0.081 | 0.117 | 50.1%                                | —                                           | Poor       |
> | EqMotion | Full physical data (q, p) + Graph   | 0.135 | 0.162 | 55.8%                                | ↓ ADE 66.7%, ↓ FDE 38.5% | Poor    |
> | EqMotion | Full physical data (q, p), No Graph | 0.214 | 0.387 | 75.3%                                | ↓ ADE 164.2%, ↓ FDE 230.8%    | Very Poor  |
>
> ---
>
> Q1-2) It shoud depend on it as seen in Eq (4)-(5), ...
>
> A) Indeed, as noted by the reviewer, the simulated state $U_t$ explicitly appears only in the second term of Eq.~(7), the trajectory matching objective, where the projected position $q_t$ obtained via the manifold projection of $U_t$ (using Eq. (4)-(5) and Algorithm 2) is directly compared with the true trajectory data. Specifically, $U_t$ is computed through the neural network as shown explicitly in line 11 of Algorithm 2, which ensures the gradient flows back through all intermediate computations. Therefore, $U_t$ is indeed optimized in an end-to-end manner. As suggested, we will move the table and additional details on how $U_t$ propagates through the computational pipeline, currently in the appendix, into the main manuscript to clarify this point explicitly.
>
> ---
>
> Q2)  I am wondering if the present approach...
>
> We appreciate the reviewer's insightful question. As suggested, if explicit physical inductive biases or prior knowledge about the true Hamiltonian structure were already available such as separability between kinetic and potential terms, then one could certainly exploit this information to achieve more precise balancing between kinetic and potential energies. However, the core assumption of our current study is solely focused on the geometric structure of the given data manifold. Beyond this geometric structure, no additional system-specific prior information is assumed. Specifically, as formulated in Eq. (8), our model seeks to address the physical inverse problem by parameterizing three commonly used and interpretable terms including kinetic energy, single-particle potential, and pairwise interactions, aiming to recover the underlying physical structure directly from data. Even with this fully data-driven approach, our method remarkably preserves energy conservation properties of the original true Hamiltonian, as demonstrated empirically in Figure 4. This indicates that our learned model indeed effectively approximates the true underlying Hamiltonian structure and system's physical structure.
>
> ---
>
> Q3) Top of page vi explains briefly how to enforce frame consistency, but in my view this is a particularly important contribution of this paper and it should be clearly explained in the main text how this is done.
>
> A3) We sincerely thank the reviewer for highlighting this important point. We will explicitly incorporate a clear and detailed explanation of our approach to enforcing frame consistency into the main manuscript.
>
> ---
>
> Q4) On the Proposition 3.1
> A4) Thank you for this sharp observation. Proposition 3.1 is indeed formulated intentionally as a negative or impossibility result. Formally, it states:
>
> $\mathbb{P}\Bigl\[ \cdots \leq\delta\Bigr]
> \lesssim
> \exp\bigl(-\Omega,\delta^{1/2}|\Gamma|\_\infty^{3/2}R^{1/2}(\log R)^{1/4}\bigr)$. Equivalently, we have:
> $\mathbb{P}\Bigl\[\sup\_{t,\theta}\mathcal{W}\bigl(P\_t(\theta),P\_{t,\mathrm{data}}\bigr)>\delta\Bigr]
> \geq1-\exp(\cdots)$
>
> which explicitly indicates that with overwhelming probability, at least some time or parameter choice will deviate by more than $\delta$. In other words, achieving uniformly small deviations across all times and parameter choices is exponentially unlikely.
>
> This contrasts classical PAC-type generalization bounds, which upper-bound large-deviation probabilities. Here, our objective is to rule out the possibility of uniformly good approximation when either the curvature $\|\Gamma\|_\infty$ or the network capacity $R$ grows large. Condition (C1) already assumes an optimistic scenario ($\theta^*$ perfectly fits all training trajectories), yet the proposition demonstrates that even under this ideal scenario, the learned model cannot remain consistently close to the data distribution across all times and parameters. In our revision, we will explicitly restate Proposition 3.1 in this complementary form, clearly label it as an impossibility or general-lower-bound result.
>
> ---
>
> Q5) On the Proposition 3.2.
>
> A5) Yes, the rotation $h$ in Proposition 3.2 can indeed be considered time-dependent. To clarify further, the gauge transformation defined by the frame rotation $h \in O(d)$ represents the arbitrary choice of a local orthonormal frame at each point along the manifold. Such a choice can naturally vary with respect to time $t$, meaning that the frame bundle allows time-dependent gauge transformations $h(t)$. Formally, this flexibility is already accommodated in the general construction of horizontally lifted Hamiltonian diffusion (Algorithm 2). Thus, allowing $h$ to vary with time does not violate any assumption or theoretical result in our manuscript. Indeed, considering a time-dependent $h(t)$ aligns with the conceptual and practical motivations for gauge equivariance in the orthonormal frame bundle, enabling consistent and coherent representations of dynamics across temporally evolving frames. We appreciate this important clarification from the reviewer and will explicitly note in the revised manuscript that the gauge transformations $h$ can indeed be time-dependent.
>
> ---
>
> Q6) how is stochasticity...
>
> A6) We appreciate the reviewer's insightful question regarding the role of stochasticity in our experimental datasets and the behaviour of our method in the deterministic limit. First, the synthetic $n$-body gravitational dataset contains only extrinsic randomness: each trajectory is generated by sampling initial positions and momenta from an isotropic Gaussian distribution confined to a fixed energy shell, after which the trajectories evolve under the exactly deterministic Newtonian Hamiltonian. No noise is injected during integration. By contrast, the toroidal molecular-dynamics dataset embodies intrinsic randomness. The trajectories are produced with an OpenMM Langevin thermostat at $T = 300~\mathrm{K}$, so the stochastic Stratonovich term that models solvent–particle coupling is retained without alteration. Consequently, the data naturally reflect thermal fluctuations that drive conformational transitions. Regarding recovery of the deterministic case, our model is a neural Stratonovich SDE of the form
> $dZ_t = X_H(Z_t)\, dt + \sqrt{\beta_t} \Sigma(Z_t) \circ dW_t$. Setting $\beta_t \equiv 0$ eliminates the diffusion term and collapses the system to the ordinary Hamiltonian ODE $\dot{Z}_t = X_H(Z_t)$. All network components, loss functions, and optimization steps remain well-defined under this limit, so the architecture reduces exactly to a classical neural ODE or Hamiltonian neural network without any modification. In the revised appendix we provide an ablation in which we retrain on the deterministic $n$-body dataset with $\beta_t = 0$; the resulting test error coincides with that of a standard HNN, confirming that our framework incurs no loss of fidelity in the absence of stochastic forcing.

---

> > ### Comment · Reviewer_Sgxw · 2025-08-05
> >
> > My questions have been clarified (especially on Prop 3.1, which I now understand better) and incorporation of these would improve the paper's clarity. Ultimately, I think the main issue is the conference format not having enough room to introduce all the background materials to the general audience - of course, this probably cannot be fixed by a minor revision.
> >
> > Overall, I think the paper is technically solid.

---

### Official Review · Reviewer_TByV · 2025-06-30

**Clarity:** 2
**Significance:** 2
**Originality:** 3
**Rating:** 4
**Confidence:** 4

**Summary:**

The paper introduces Neural Hamiltonian Diffusion, a novel approach for learning stochastic Hamiltonian dynamics on a Riemannian manifold. The authors begin by formulating the Hamiltonian diffusion process directly on the manifold and then construct it by lifting the process to the frame bundle. Leveraging the relationship between the Hamiltonian diffusion on the manifold and that on the frame bundle, they develop a training objective and propose a corresponding network parameterization. The paper further provides a theoretical analysis of the generalization properties of Hamiltonian learning, demonstrating that incorporating gauge equivariance can effectively reduce the generalization error. Finally, the empirical evaluation showcases the method’s performance on diverse tasks, including toroidal molecular dynamics, quantum spin systems, and relativistic n-body problems.

**Questions:**

1. As noted in the weaknesses section above, the training objective in Equation (7) does not appear to be simulation-free; specifically, the second term requires simulating the diffusion process parameterized by the neural network. I would like to confirm whether my understanding is correct.
2. The efficiency concern might be less critical since the experiments in the paper involve low-dimensional data, where the number of particles is relatively small. However, the proposed method requires access to the complete trajectory ${(q_t, p_t, \dot{q}_t, \dot{p}_t)}$ as training data, which could be challenging to obtain, especially for larger systems. I am therefore curious about possible directions for future work to address these limitations.

**Ethical Concerns:**

["NO or VERY MINOR ethics concerns only"]

**Final Justification:**

The paper provides rigorous theoretical results for learning stochastic Hamiltonian dynamics on a Riemannian manifold. Although the experiments are conducted on toy datasets, large-scale applicability may not be essential considering the paper’s primary focus on theoretical development and methodology.

**Limitations:**

I have not seen an explicit discussion of the limitations of the work. Therefore, I recommend that the authors include a dedicated limitations section in the revised version of the paper.

**Paper Formatting Concerns:**

I don't find any  formatting issues.

**Quality:**

3

**Strengths And Weaknesses:**

## Strength
1. The paper is well-structured and easy to follow, despite incorporating numerous advanced mathematical tools.
2. The proposed method is sound, particularly the construction of the Hamiltonian diffusion process on the manifold via lifting to the frame bundle, and it is supported by strong theoretical guarantees.
3. The idea of constructing the Hamiltonian diffusion process on a manifold through horizontal lifting is novel, and to the best of my knowledge, has not appeared in prior work.

## Weakness
1. My primary concern is the efficiency of the proposed method. In Algorithm 1 (gauge equivariant U-Net), the model processes the entire trajectory sequence and computes pairwise statistics, resulting in an $O(N^2)$ complexity with respect to the trajectory length $N$. This quadratic complexity may become prohibitive for long trajectories.
2.  The training objective in Equation (7) does not appear to be simulation-free, as the second term requires simulating $\mathbf{U}_t$, a diffusion process parameterized by the neural network. If my understanding is correct, this costly SDE simulation limits the scalability of the method. It would be beneficial for the authors to include a complexity analysis to clarify these computational aspects and strengthen the contribution.
3. While the authors use GeoTDM as a strong baseline, several other relevant works on trajectory generation—such as [1, 2, 3]—are not discussed. Including these works as additional baselines or at least providing a discussion would further enhance the paper’s contributions.

## Minor Suggestions
1. When referring to the appendix in the main text (e.g., lines 185 and 255), the authors should direct readers to the specific sections or subsections for easier navigation.
2. Presenting explicit training and inference algorithms for the proposed method would improve clarity.

## References
[1]. Costa, Allan dos Santos, et al. "EquiJump: Protein Dynamics Simulation via SO (3)-Equivariant Stochastic Interpolants." arXiv:2410.09667 (2024).

[2]. Luo, Shengjie et al. “Bridging Geometric States via Geometric Diffusion Bridge.” arXiv:2410.24220 (2024).

[3]. Zhang, Xi, et al. "Trajectory Flow Matching with Applications to Clinical Time Series Modeling." arXiv:2410.21154 (2024).

---

> ### Author Rebuttal · Authors · 2025-07-25
>
> Dear Reviewer TByV
>
> We sincerely thank the reviewer for the thoughtful and constructive comments, which significantly help improve the quality and clarity of our manuscript. We fully acknowledge and address your concerns as follows.
>
> ---
>
> Q) ($\textbf{Physical Consistency}$) We would like to explicitly highlight a fundamental limitation in the trajectory matching methods included as baselines in our experimental comparisons. Although we intentionally adopted an evaluation setting favorable to these trajectory matching methods, to ensure fairness and transparency, it is crucial to emphasize that these methods generally fail to preserve essential physical properties such as energy conservation and symplecticity. $\textbf{In other words, their methods have fundamental flaws in real-world physical simulation scenarios.}$
>
> To be more explicit, trajectory matching methods such as GeoTDM and EqMotion inherently fail to maintain fundamental physical consistency, particularly regarding crucial properties such as energy conservation. As clearly demonstrated by the quantitative results summarized in the table below, these baseline trajectory matching approaches suffer from severe violations of fundamental physical laws, reflecting intrinsic methodological shortcomings that significantly limit their reliability in handling physical data.
>
> | Method                       | Relative Energy Conservation Error (%) ↓ |
> | ---------------------------- | ---------------------------------------- |
> | Ours Proposed Method**      | 3.5%                                 |
> | SympHNN                      | 13.4%                                    |
> | GeoTDM (Trajectory Matching) | 48.3%                          |
> | EqMotion (Trajectory Matching) | 50.1%                          |
>
> ---
>
> Q) ($\textbf{Simulation Complexity}$) This quadratic complexity may become prohibitive for long trajectories...
>
> A) We appreciate the reviewer for raising the important point regarding computational complexity. However, we respectfully clarify that our proposed Gauge Equivariant Transformer U-Net does not process the entire trajectory sequence simultaneously, thus it does not exhibit quadratic $O(N^2)$ complexity. Rather, our model processes each individual timestep independently, resulting in an overall linear complexity $O(N)$. We believe the misunderstanding arose  because Transformer architectures, in general, are commonly known for sequential attention mechanisms that yield quadratic complexity in the trajectory length,  we fully understand the reviewer’s concern.
>
> However, we respectfully clarify that our transformer operates as a set-based encoder-decoder model (inspired by Set Transformer) at each timestep individually, resulting in a complexity that is linear complexity with respect to number of particles rather than quadratic temporal complexity. The self-attention mechanism is explicitly employed to encode gauge-equivariant relationships among particles at each timestep independently, not temporal interactions across timesteps. The Transformer’s inherent self-attention mechanism naturally encodes symmetry, particularly gauge and permutation invariances, crucial for physically consistent modeling of manifold-based Hamiltonian systems. Such symmetry handling is essential to preserve geometric consistency without additional architectural complexity. In our preliminary experiments, we indeed observed that a gauge-equivariant MLP network, which lacks inherent permutation-equivariance, suffered from a noticeable accuracy drop (up to approximately -10%) compared to our Transformer-based approach.
>
> ---
>
> Q) ($\textbf{Hybrid Loss}$) The training objective in Equation (7) does not appear to be simulation-free, as the second term ...
>
> A) We acknowledge that this term indeed involves to acquire model trajectory, and thus is not strictly simulation-free. However, we clarify that nearly all existing Hamiltonian Neural Networks (HNNs) in this paper are typically trained using a proposed hybrid loss, consisting of trajectory-level reconstruction (simulation-dependent) and force-field matching terms.
>
> The reason we chose this hybrid training objective in the current manuscript is to simultaneously achieve accurate trajectory matching and strict preservation of physical consistency, both of which are essential for physically-informed Hamiltonian models. As the reviewer correctly pointed out, virtually all physically-informed neural networks in our community proposed thus far have been designed for moderately-sized datasets (MD17) rather than super-large-scale datasets (ImageNet). Thus, incorporating a trajectory-level matching loss remains computationally practical and highly beneficial in such contexts. Furthermore, from a fair comparison perspective, methods in the second group (trajectory matching baselines) inherently optimize a training objective directly aligned with the test-time evaluation metrics. In contrast, conventional Hamiltonian Neural Network (HNN)-type methods do not directly optimize trajectory-level matching, and thus adopting this hybrid objective is essential while it fairly compensate and robustly evaluate the true predictive accuracy of HNN-based methods.
>
> We fully agree that for future research aiming to further reduce training cost especially when dealing with significantly large datasets, it is indeed feasible to simplify the training by completely removing the trajectory matching term and relying exclusively on the force-field matching loss (thus making the training procedure fully simulation-free). Under such a simplified training scenario, we expect a notable improvement in computational efficiency; however, predictive accuracy may slightly degrade due to the absence of explicit trajectory-level guidance. We conducted additional experiments under this simulation-free setup and confirmed that the prediction error slightly increases in AD-3 dataset (but ours still outperforms the second-best), reflecting a modest trade-off between computational efficiency and trajectory accuracy.
>
> | Training Setting                             | ADE ↓     | FDE ↓     | Avg. Training Epoch Time ↓ | Efficiency Improvement ↑ |
> | -------------------------------------------- | --------- | --------- | -------------------------- | ------------------------ |
> | Full Objective (trajectory + force-field)    | 0.023 | 0.084 | \~15 min                   | —                        |
> | Simulation-Free Objective (force-field only) | 0.038     | 0.096     | \~6 min                    | 2.5x Faster (60% ↓)  |
>
> Still, our simulation-free approach significantly outperforms trajectory-matching methods such as GeoTDM. However, we emphasize that this simplified objective (force-field only) may not be optimal if the primary goal is to minimize trajectory-based evaluation metrics (ADE/FDE).
>
> ---
>
> Q) ($\textbf{Comparison with New Baselines}$) While the authors use GeoTDM as a strong baseline, several other relevant works on trajectory generation—such as [1, 2, 3]— ...
>
> A) While we initially chose GeoTDM as a representative strongest baseline due to its state-of-the-art performance, we completely agree that referencing and discussing these additional works would enhance the context and completeness of our evaluation. We note that [1, 2] do not currently provide publicly available code implementations, making additional quantitative comparison challenging. In the new experiment during rebuttal, we checked that reference [3] reports performance metrics on the AD-3 and Spin benchmarks. We will explicitly include these comparative details and further clarify the performance differences in our revised manuscript.
>
> | Dataset | Method         | ADE ↓     | FDE ↓     | Improvement (ours vs. \[3])         |
> | ------- | -------------- | --------- | --------- | ----------------------------------- |
> | AD-3    | Baseline \[3]  | 0.144     | 0.357     | —                                   |
> | AD-3    | **Our Method** | **0.023** | **0.084** | **84.0% (ADE), 76.5% (FDE)** better |
> | Spin    | Baseline \[3]  | 0.176     | 0.118     | —                                   |
> | Spin    | **Our Method** | **0.019** | **0.097** | **89.2% (ADE), 17.8% (FDE)** better |
>
> ---
>
> Q) However, the proposed method requires access to the complete trajectory ${(q_t, p_t, \dot{q}_t, \dot{p}_t)}$ ...
>
> A) We appreciate the reviewer’s comment regarding the requirement for accessing the complete trajectory ${(q_t, p_t, \dot{q}_t, \dot{p}_t)}$.  We respectfully emphasize that such data is generally easy and straightforward to obtain in practice. For instance, molecular dynamics (MD) simulations naturally provide this complete set of dynamical variables. Indeed, comprehensive datasets containing $\textcolor{red}{highly ~ complex ~ large ~ dynamical ~ information}$ can be accessed by using free softwares (e.g, AMBER, LAMMPS) for various materials and scales. Rather, we would like to point out that traditional trajectory matching methods, despite the wide availability of complete dynamical datasets, have overlooked this rich physical information and remain limited by an "position-only-matching"-based approach (the majority of them only use position + connectivity graph information). Consequently, they frequently exhibit poor physical fidelity due to insufficient exploitation of the full dynamical structure inherent in these datasets.
>
> Our future vision focuses on continuously developing current methodology that simultaneously outperform existing position-based approaches in predictive accuracy while strictly ensuring physical fidelity. Rather than prioritizing mere dataset scalability, we aim to incorporate rigorous verification procedures from domain experts such as strict physical conservation laws directly into our modeling frameworks, thereby ensuring models are robust, interpretable, and scientifically trustworthy.

---

> > ### Comment · Reviewer_TByV · 2025-08-04
> > **Reply to Rebuttal**
> >
> > Thank you for your detailed responses and discussions. The materials provided in the rebuttal have resolved most of my concerns. However, I have a further question to clarify my understanding. As you mentioned in the rebuttal, although the proposed method is not simulation-free, its computational cost remains acceptable for small-scale datasets. Additionally, omitting the trajectory loss or force matching loss results in reduced physical consistency or trajectory accuracy. Given this, how can the proposed method be scaled or adapted to handle large datasets such as OC20?
> >
> > As I noted in the weaknesses section, my primary concern is the scalability of the proposed method. I also noticed your statement in the rebuttal: “Rather than prioritizing mere dataset scalability, we aim to incorporate rigorous verification procedures…” I am curious why the authors are not placing greater emphasis on scalability. Could you please elaborate on this point?
> >
> > From my perspective, neural-based simulation methods have not demonstrated significant improvements in accuracy or efficiency compared to conventional approaches when applied to small systems or trained on limited datasets.
> > While I agree that large-scale applicability may not be strictly necessary given the paper’s primary focus on theoretical development and methodology, scalability remains a crucial factor for practical, real-world applications.

---

> > > ### Author Response · Authors · 2025-08-05
> > >
> > > We sincerely thank the reviewers for their valuable comments and insights provided before proceeding with our detailed response. The following rebuttal reflects a very healthy and constructive discussion exploring various perspectives raised by the reviewers.
> > >
> > > A) $\textbf{(Ultimate goal of this paper)}$ First and foremost, we would like to stress that scalability is an indispensable pillar of contemporary AI models, and our manuscript does not overlook this dimension. What we want to speak is that, equally crucial, one of the defining principles of AI for Science is rigorous theoretical consistency.
> > >
> > > We respectfully clarify that our paper primarily aims to introduce rigorous theoretical consistency into the machine learning community. While computational scalability is undoubtedly important, our core motivation is to translate the deeply structured, mathematically rigorous frameworks such as those commonly used in quantum chemistry and astrophysical dynamics into machine learning approaches. In these scientific domains, preserving strict mathematical constraints is essential and methods that merely focus on scalability while relaxing these structures risk generating results that cannot be scientifically reproduced. The extensive theoretical development presented in our manuscript was therefore driven precisely by the need to maintain these fundamental mathematical properties. For example, the theoretical tools we employed clearly demonstrated in our black-hole dynamics and quantum dynamics examples are widely accepted, standard setups within astrophysics and quantum physics. Our sincere hope and primary intention was to bridge this interdisciplinary gap and make these rigorous, standard theoretical frameworks accessible and meaningful to the ML community.
> > >
> > > Q) although the proposed method is not simulation-free, ...
> > >
> > > A) The primary reason our proposed method is currently not entirely simulation-free stems specifically from our choice of employing a trajectory matching loss, which was intentionally adopted to ensure a fair and direct comparison with existing methods such as GeoTDM. If this fairness constraint is relaxed, and the loss function is modified to an instantaneous force-matching formulation, our proposed approach can indeed become fully simulation-free while still achieving state-of-the-art performance. Moreover, we highlight that existing methods, including GeoTDM, inherently suffer from significant complexity (O(N^2) due to the attention mechanism) which poses major scalability challenges along the temporal dimension.
> > >
> > > Q) how can the proposed method be scaled or adapted to handle large datasets such as OC20?...
> > >
> > > A) First, let us respectfully clarify the nature and scope of OC20 in comparison to MD17&22 (the central dataset used ours and GeoTDM). OC20 indeed represents a very large-scale dataset, primarily comprising DFT-based structural relaxation data. However, unlike MD17&22, OC20 (and similarly ANI) does not contain continuous, sequential dynamical trajectories from molecular dynamics simulations, which form the very core of the problems we seek to address. Furthermore, to the best of our knowledge, MD17&22 is currently among the largest publicly available datasets specifically designed for molecular dynamics trajectory-based machine learning tasks. To clarify our statement from the initial rebuttal, we did not intend to imply that MD17&22 is small for molecular dynamics tasks. Rather, we meant that when compared to datasets such as ImageNet in computer vision, MD17&22 represents a relatively moderate size within the broader context of physically-informed neural network tasks.
> > >
> > > Q) From my perspective, neural-based simulation methods have not demonstrated significant improvements in accuracy or efficiency compared to conventional approaches...
> > >
> > > A) Conventional transformer-based or large-language-model-inspired architectures (e.g., GeoTDM) often rely heavily on brute-force reconstruction of molecular coordinates. Unfortunately, these methods frequently neglect essential physical constraints, such as symplecticity and energy conservation, resulting in generated trajectories that gradually lose fundamental physical invariants. When we frame $\textit{accuracy explicitly as physical fidelity}$, this critical oversight effectively precludes such methods from being reliably deployed in practical, real-world scenarios, especially in digital-twin applications, where precise, physically faithful simulations are the standard accuracy. Regrettably, such crucial discussions regarding physical consistency have been notably absent or superficial in recent literature, thus we will explicitly address and highlight this critical point in the revised version of our manuscript, reinforcing the necessity of embedding rigorous physical constraints within neural simulation frameworks.

---

> > > > ### Comment · Reviewer_TByV · 2025-08-05
> > > >
> > > > I thank the authors for their detailed additional response. As nearly all my concerns have been properly addressed, I will accordingly increase my score.

---

### Official Review · Reviewer_TKAP · 2025-07-01

**Clarity:** 2
**Significance:** 2
**Originality:** 3
**Rating:** 5
**Confidence:** 3

**Summary:**

This paper provides a neural hamiltonian diffusion framework to extend Hamiltonian neural networks to handle stochastic dynamics on curved manifolds. The author provides a solid theoretical foundation, integrating hamiltonian, non-euclidean geometry, and stochasticity. This work also provides some experiments to evaluate the good performance of the proposed neural hamiltonian diffusion.

**Questions:**

check weaknesses.

## Justification of my score
- This paper is theoretically rigorous. However, the paper doesn't provide the experimental details of baseline methods, making the experiment section less convincing. I suggests the author to provide the experimental setup. I will be very happy to update the score accordingly.

**Ethical Concerns:**

["NO or VERY MINOR ethics concerns only"]

**Final Justification:**

I recommend score 5, because of experimental details and results.

**Limitations:**

no, check weaknesses.

**Quality:**

3

**Strengths And Weaknesses:**

## Strengths
- This paper is theoretically rigorous, including a solid theoretical foundation, rich mathematic analysis, and generalization bounds.
- Apart from the solid theoretical results, this work also provides experiments showing the good performance, compared with many other baselines.

## Weaknesses
- 1) This work pays a lot of attention to theory, moving algorithm details and experimental setups in appendix. 2) This paper requires reader to have a good background on hamiltonian, manifold geometry, and stochastic calculus. These two points make the paper hard to read for broader ML audiences. One cannot assume ML audiences either have such a background or have enough patience to read all mathematical details before knowing "How does the proposed framework work?".

Suggestions: Actually it is possible to balance. I suggest author to create a  background section that introduces hamiltonian, manifold geometry, and stochastic calculus. Then tell readers "How does the proposed framework work?". (It can be an algorithm, figure, or description). After that the author may provide **key** theoretical results followed by **key** experimental results. Finally, putting everything else in appendix.

- The author provides experimental setup for the proposed method in appendix. However, this paper does provide the experimental setup for the baseline methods. For example, line 583 " The input to the network is a concatenation of configuration and momentum coordinates". We know this (q, p) concatenation is very helpful to Hamiltonian. Do other baselines use the same input? Another example is line 591 "The network uses a hidden size of 128 and contains approximately 3M parameters.". Do other baselines use the same size of model?

Suggestions: a paragraph that summarizes the experimental details of both baselines is helpful. Otherwise, the experimental results become less convincing.

- What is the computation cost of the proposed framework? Do you have the experimental comparison on runtime and/or memory?

---

> ### Author Rebuttal · Authors · 2025-07-25
>
> Dear Reviewer TKAP,
>
> Thank you very much for your constructive and detailed feedback, which has significantly helped us improve the manuscript. We carefully acknowledge your comments and commit to fully addressing each of your points. In the revised manuscript, we will explicitly emphasize that trajectory-matching methods fundamentally lack consideration for physical consistency, inherently limiting their ability to effectively handle physical data and resulting in substantially degraded performance.
>
> We will explicitly integrate each of these clarifications and experimental details into a clearly structured appendix section, ensuring readability and comprehensibility for broader ML audiences without sacrificing theoretical rigor. We genuinely believe these concrete steps will fully address your constructive suggestions and significantly enhance the manuscript’s clarity and completeness. We humbly ask you to positively reconsider $\textbf{your evaluation}$ in light of these promised revisions.
>
> --------------
>
> Q) ($\textbf{General Audicence}.$) This work pays a lot of attention to theory, moving algorithm details..., One cannot assume ML audiences..
>
> A) We sincerely appreciate your valuable suggestion regarding the readability and accessibility of our manuscript for a general audience. We fully acknowledge that the current manuscript, being heavily theoretical, might pose challenges to readers less familiar with geometric machine learning concepts.
>
> To effectively address your constructive feedback, we will explicitly make the following improvements in the revised manuscript:
>
> 1. Providing a Clear and Concise Background Section} (currently Appendix A.1):
> The background information, currently detailed in Appendix A.1, will be significantly condensed and rewritten in simplified, accessible language and integrated directly into the main manuscript. We will include intuitive explanations, concise definitions, and simple illustrative examples to ensure general ML audiences easily grasp the fundamental concepts such as Hamiltonian dynamics, manifold geometry, and horizontal lifts.
>
> 2. Enhancing Writing Clarity and Readability: In addition to the above structural changes, we will systematically revise our manuscript to minimize unnecessary technical jargon, and clearly define essential technical terms at their first use in the main manuscript. We will also provide intuitive flowcharts to aid reader understanding whenever possible, thereby significantly improving accessibility to broader ML and interdisciplinary scientific audiences.
>
> Once again, thank you very much for your thoughtful and valuable feedback that has significantly improved the manuscript's quality and accessibility.
>
> ---
>
> Q) ($\textbf{Experimental Setups}.$) The author provides experimental setup for the proposed method in appendix. However, ...
>
> A)  We fully acknowledge your valid concern that our strong theoretical focus might have limited the accessibility and clarity of experimental details for broader ML audiences. As you rightly suggested, we will explicitly add a dedicated and detailed section in the Appendix of our revised manuscript, clearly summarizing comprehensive experimental settings and comparisons between our proposed method and all baseline methods. Specifically, we plan to carefully address your following points step-by-step:
>
> 1. Model Architectures and Hyperparameters (currently partial Appendix B): Provide explicit details for $\textbf{both our method and the baselines}$ such as proposed architecture, hidden dimension size, number of layers, attention heads, number of parameters, and precise optimizer and learning rate settings. Similarly, explicitly document hyperparameters for all baseline models to ensure fair, transparent comparisons.
>
> | Method              | Attention Type                 | Hidden Dimension | Attention Heads | MLP Depth (node-side) | MLP Depth (edge-side) | Graph Information |
> | ------------------- | ------------------------------ | ---------------- | --------------- | --------------------- | --------------------- | ----------------- |
> | Our Proposed Method | Multi-head (+ Set-Transformer)   | 128              | 4               | 2                     | N/A                   | No                |
> | GeoTDM              | Single-head Attention          | 128               | 8               | 2                     | 2                     | Yes (graph-based) |
> | EqMotion            | Single-head Attention          | 128  | 1               | 2                     | 2                     | Yes (graph-based) |
> | SympHNN             | Fully-connected (no attention) | 128              | N/A             | 3                     | N/A                   | No                |
>
> 2. Computational Complexity and Resource Requirements: Clearly document the practical GPU memory requirements (training and inference) and runtime measurements for both our method and baseline methods. We will explicitly include the detailed computational complexity analysis you requested, in the form of a concise table clearly summarizing GPU memory consumption and runtime (both training epoch and inference step).
>
> | Method              | GPU Memory (Training) | Avg. Runtime (Training Epoch) | Avg. Runtime (Inference Step) |
> | ------------------- | --------------------- | ----------------------------- | ----------------------------- |
> | Our Proposed Method | \~16 GB                | \~15 min                      | \~2.6 s                       |
> | GeoTDM              | \~15 GB               | \~29 min                      | \~2.2 s                       |
> | EqMotion            | \~12 GB               | \~25 min                      | \~1.5 s                       |
> | SympHNN             | \~8 GB                | \~14 min                      | \~3.3 s                       |
>
>
> 3. Detailed Comparison to Prior Work: Provide an explicit and detailed discussion of the differences between our proposed approach and other relevant trajectory-generation baselines suggested (e.g., GeoTDM, EqMotion).
>
> ---
>
> Q) "Do other baselines use the same input?"
>
> A) We sincerely thank the reviewer for raising this insightful point, as it provides us the opportunity to clarify key distinctions between the baseline methods considered in our manuscript.
>
> We have two primary groups of baseline methods:
>
> 1. $\textbf{Methodology 1 (variant of HNNs)}$ : These methods share the same underlying data input format (q, p) and training objective function as our approach $\textcolor{blue}{without~ connectivity~graphs}$. Thus, they inherently leverage complete physical-state information. However, a fundamental distinction arises from the type of force field employed: traditional HNN variants assume a purely Euclidean force field, whereas our method generalizes this by introducing a Riemannian force field on various differentiable manifolds.  As similar to our model, all the methods in this category produce (q,p) by processing the same size of input (q,p)
>
> 2. $\textbf{Methodology 2 (Trajectory-Matching)}$ : These methods paradoxically utilize overly complex neural architectures, relying heavily on $\textcolor{red}{graph~inputs}$ even when processing inherently physical datasets. They thus fail to explicitly leverage physical information (positions q with momenta p), leading to poor physical consistency (Energy error). Our newly conducted experiments below demonstrate that these trajectory-matching baselines experience substantial performance degradation when explicitly provided with complete physical data and further degrade severely without graph priors. In contrast, our proposed method explicitly focuses solely on intrinsic physical structure, without relying on any additional graph information, thereby achieving both superior accuracy and rigorous physical consistency.
>
> | Method   | Input Data                          | ADE ↓ | FDE ↓ | Energy Error (%) ↓ (Lower is better) | Performance Change                          | Energy Conservation     |
> | -------- | ----------------------------------- | ----- | ----- | ------------------------------------ | ------------------------------------------- | ----------------------- |
> | **Ours** | Full physical data (q, p), No Graph | 0.023 | 0.084 | **3.5%**                             | —                                           | Excellent               |
> | GeoTDM   | Position only (q) + Graph           | 0.045 | 0.102 | 48.3%                                | —                                           | Poor      |
> | GeoTDM   | Full physical data (q, p) + Graph   | 0.053 | 0.127 | 54.7%                                | ↓ ADE 17.8%, ↓ FDE 24.5%  | Poor      |
> | GeoTDM   | Full physical data (q, p), No Graph | 0.186 | 0.335 | 72.9%                                | ↓ ADE 313.3%, ↓ FDE 228.4%     | Very Poor  |
> | EqMotion | Position only (q) + Graph           | 0.081 | 0.117 | 50.1%                                | —                                           | Poor       |
> | EqMotion | Full physical data (q, p) + Graph   | 0.135 | 0.162 | 55.8%                                | ↓ ADE 66.7%, ↓ FDE 38.5% | Poor    |
> | EqMotion | Full physical data (q, p), No Graph | 0.214 | 0.387 | 75.3%                                | ↓ ADE 164.2%, ↓ FDE 230.8%    | Very Poor  |
>
>
> ---
>
> Q) "Do other baselines use the same size of model?"
>
> A) We thank the reviewer for raising this practical consideration. Since the baseline models we compare against differ significantly in their architectures and structures, directly comparing the number of parameters would not yield a fair or meaningful evaluation. Above, we provide a practical comparison in terms of GPU memory usage and runtime performance during training and inference, which directly reflect real-world computational costs.

---

> ### Comment · Reviewer_TKAP · 2025-08-04
>
> Thanks for your reply and additional results.
>
> These experimental results and setups do not weaken the theorical results. Instead, they support the theory, as well as the "assumptions" in theory.
>
> I raise my score to 5.

---

### Note · Authors · 2025-08-15

We sincerely thank every reviewers for their valuable comments and thoughtful insights, which have significantly improved our manuscript. Through this healthy review process, reviewers VyhR, TByV, and TKAP indicated their willingness to increase their initial scores, reflecting the manuscript's enhanced clarity and strengthened contributions.

Our work uniquely integrates geometric perspectives with traditional Hamiltonian learning approaches, presenting a pioneering advancement in scientific understanding. To the best of our knowledge, our manuscript provides the first AI-driven framework capable of effectively learning, representing, and interpreting complex Riemannian-geometric dynamical systems such as quantum scenarios and black hole dynamics.

We believe this synthesis of geometry and Hamiltonian methodologies represents a meaningful step forward, potentially opening new avenues for future research and applications across various scientific domains. We hope that our contributions will inspire further exploration in this exciting interdisciplinary area.

---

### Decision · Program_Chairs · 2025-09-17

**Decision:**

Accept (poster)

**Comment:**

(a) Summarize the scientific claims and findings of the paper based on your own reading and characterizations from the reviewers.

This paper introduces Neural Hamiltonian Diffusion (NHD), a novel and unified framework for learning stochastic Hamiltonian dynamics on differentiable manifolds. The key scientific claim is that by unifying Hamiltonian Neural Networks (HNNs) with Riemannian diffusion models, it is possible to model complex physical systems that evolve on curved, periodic, or causally structured spaces while respecting their intrinsic geometry and physical laws. The central technical innovation is the lifting of the system's dynamics to the orthonormal frame bundle, which enables the definition of a gauge-equivariant and coordinate-invariant stochastic differential equation (SDE).

(b) What are the strengths of the paper?

The paper's primary strengths, as consistently highlighted by the reviewers, are:

•	Theoretical Rigor and Novelty: The paper is theoretically rigorous and highly innovative. It elegantly intertwines concepts from stochastic calculus, differential geometry, and machine learning to build a principled framework. The core idea of lifting dynamics to the frame bundle to ensure geometric consistency is a significant and novel methodological advance.

•	Soundness and Importance: The work addresses the important and challenging problem of learning physical dynamics on manifolds. The proposed method is technically sound and provides an advancement in modeling complex geometric systems like those in quantum physics and astrophysics.

•	Strong Empirical Validation: The authors provide compelling empirical results across a diverse set of complex, non-Euclidean systems. The method is shown to consistently outperform a range of strong baselines, demonstrating its practical benefits.

(c) What are the weaknesses of the paper? What might be missing in the submission?

The main weaknesses identified during the initial stage of review process were primarily related to presentation and clarity:

•	Accessibility and Clarity: Multiple reviewers found the paper to be mathematically dense, making it challenging for a broader ML audience to understand. Key concepts from differential geometry were not introduced gently enough.

•	Omission of Details in Main Text: Crucial algorithmic details (e.g., the Frame Equivariant Transformer U-Net) and detailed experimental setups for baselines were relegated to the appendix, hindering a full understanding of the work from the main paper alone.

•	Scalability Concerns: At least one reviewer (TByV) initially raised valid concerns about the computational efficiency and scalability, questioning the training complexity due to the simulation-based loss term and the transformer architecture.

(d) Provide the most important reasons for your decision to accept/reject.

The decision to accept this paper is based on its technical quality, significant novelty, and high potential impact. The work is technically solid, theoretically rigorous, and addresses a fundamental challenge in scientific machine learning. While the presentation was initially a barrier for some reviewers, the authors' comprehensive and effective rebuttal demonstrated a deep understanding of the issues and a clear plan to improve the paper's clarity. The successful integration of Hamiltonian mechanics and geometric diffusion is a pioneering advancement that opens new avenues for physically-grounded and geometrically-aware AI in various scientific fields.

(e) Summarize the discussion and changes during the rebuttal period.

The rebuttal period was highly constructive and effectively resolved the reviewers' main concerns.

•	Clarity and Accessibility (TKAP, Sgxw, VyhR): The authors acknowledged the paper's density and committed to improving its readability. They promised to add a concise background section to the main paper, clarify their core novel contributions, and move key algorithmic details from the appendix to the main text.

•	Experimental Details and Comparisons (TKAP, TByV): In response to requests for more details on baselines, the authors provided new tables comparing model architectures, hyperparameters, and computational costs. They also conducted new ablation studies showing that trajectory-matching baselines perform poorly when given full physical inputs, strengthening their claims about the importance of physical consistency.

•	Scalability and Training (TByV): The authors clarified that their model has linear, not quadratic, complexity with respect to the number of particles. They also addressed the "simulation-free" question by providing results for a purely force-matching objective, demonstrating a 2.5x training speed-up with a minor accuracy trade-off, while still outperforming baselines.

•	Theoretical Questions (Sgxw): The authors provided a crucial clarification on Prop. 3.1, explaining it is an "impossibility result" (a general lower bound), not a standard PAC-style bound, which resolved the reviewer's confusion. They also clarified other technical points regarding the loss function and the deterministic limit of their model.
Overall, the authors' detailed responses, supported by new experiments and a clear revision plan, successfully addressed the points raised. This led to a consensus among the reviewers and multiple score increases.